# Activating lattice oxygen in NiFe-based (oxy) hydroxide for water electrolysis

Zuyun He[1,5], Jun Zhang [2,5], Zhiheng Gong[1], Hang Lei[3], Deng Zhou[4], Nian Zhang [4], Wenjie Mai [3], Shijun Zhao [2✉] & Yan Chen [1✉]

Transition metal oxides or (oxy)hydroxides have been intensively investigated as promising electrocatalysts for energy and environmental applications. Oxygen in the lattice was reported recently to actively participate in surface reactions. Herein, we report a sacrificial template-directed approach to synthesize Mo-doped NiFe (oxy)hydroxide with modulated oxygen activity as an enhanced electrocatalyst towards oxygen evolution reaction (OER). The obtained MoNiFe (oxy)hydroxide displays a high mass activity of 1910 A/$g_{metal}$ at the overpotential of 300 mV. The combination of density functional theory calculations and advanced spectroscopy techniques suggests that the Mo dopant upshifts the O 2$p$ band and weakens the metal-oxygen bond of NiFe (oxy)hydroxide, facilitating oxygen vacancy formation and shifting the reaction pathway for OER. Our results provide critical insights into the role of lattice oxygen in determining the activity of (oxy)hydroxides and demonstrate tuning oxygen activity as a promising approach for constructing highly active electrocatalysts.

[1] School of Environment and Energy, State Key Laboratory of Pulp and Paper Engineering, South China University of Technology, Guangzhou, Guangdong 510006, China. [2] Department of Mechanical Engineering, City University of Hong Kong, Hong Kong, China. [3] Siyuan Laboratory, Guangdong Provincial Engineering Technology Research Centre of Vacuum Coating Technologies and New Energy Materials, Department of Physics, Jinan University, Guangzhou, Guangdong 510632, China. [4] State Key Laboratory of Functional Materials for Informatics, Shanghai Institute of Microsystem and Information Technology, Chinese Academy of Sciences, Shanghai 200050, China. [5]These authors contributed equally: Zuyun He, Jun Zhang. ✉email: shijzhao@cityu.edu.hk; escheny@scut.edu.cn

Transition metal oxides or (oxy)hydroxides have been intensively investigated as promising alternative electro-catalysts because of their high catalytic activity, low cost, and good stability. The surface metal sites in these materials are generally considered as the active sites for surface reactions[1,2]. Interestingly, recent studies have shown that oxygen in the lattice of metal oxides and (oxy)hydroxides can also participate in surface reactions and may play a critical role in regulating the catalyst activity[3–5]. For example, Grimaud[6] and Mefford[7] reported that lattice oxygen could participate in the oxygen evolution reaction (OER) on the $(La,Sr)CoO_3$ surface, which was later referred to as lattice oxygen mechanism (LOM). Similar LOM mechanisms were then discovered for the OER on other oxides and (oxy)hydroxides, such as $SrCo_{1-y}Si_yO_{3-\delta}$[8], $PrBa_{0.5}Sr_{0.5}Co_{1.5}Fe_{0.5}O_{5+\delta}$[9], $Sr(Co_{0.8}Fe_{0.2})_{0.7}B_{0.3}O_{3-\delta}$[10], CoZn (oxy)hydroxide[11], $CoAl_2O_4$[12], $Na_xMn_3O_7$[13] and NiFe (oxy)hydroxide[14]. In addition to the OER reaction at room temperature, Hwang et al.[15] reported that the NO oxidation reaction on $La_{1-x}Sr_xCoO_3$ oxides also strongly depended on the surface oxygen activity, which is defined as the oxygen $2p$-band center relative to the Fermi level. Chen et al.[16] tuned the oxygen activity in perovskite ferrite by Co doping, leading to a change in hydrogen oxidation reaction performance at elevated temperatures. All these results demonstrate that modulating lattice oxygen activity is an effective method for improving the activity of transition metal oxide or (oxy)hydroxide electrocatalysts.

OER is regarded as the main bottleneck in many electrochemical energy devices due to its sluggish reaction kinetics[9]. Because of the highly oxidative environment, OER catalysts suffer from effects such as spontaneous dissolution and surface reconstruction during operation, which strongly impact the stability of the devices[17,18]. These effects in certain cases were also reported to promote the OER activities[19–22]. For instance, several research groups reported highly active OER catalysts with perovskite oxide[8,23], nitride[24], or phoshide[25] as the core materials, and with self-reconstructed amorphous phase or (oxy)hydroxides as the active phase on the surface[18,19,26]. Jiang et al.[27] demonstrated that the leaching of lattice $Cl^-$ from cobalt oxychloride $(Co_2(OH)_3Cl)$ during the OER process could trigger the atomic-level unsaturated sites and efficiently boost catalytic activity. While all the pioneering works mentioned above have demonstrated the self-reconstruction or material leaching effects during operation as an effective way to achieve highly active catalysts, the impacts of pre-catalysts on the activity of final catalysts are still lack of investigation.

In this work, we report a sacrificial template-directed approach to synthesize ultrathin NiFe-based (oxy)hydroxide with modulated lattice oxygen activity as highly efficient and stable OER catalysts. $MoS_2$ nanosheets are used as sacrificial templates to adsorb metal cations to form self-assembled NiFe (oxy)hydroxide on the surface. After removing the $MoS_2$ sacrificial template by Mo leaching under the OER condition, ultra-thin NiFe (oxy) hydroxides with Mo doping (MoNiFe (oxy)hydroxide) are obtained. The MoNiFe (oxy)hydroxide displays enhanced OER performance, with a high mass activity of 1910 $A/g_{metal}$ at the overpotential of 300 mV, which is about 60 times higher than that of NiFe (oxy)hydroxide. The combination of synchrotron-based soft X-ray absorption spectroscopy (sXAS), X-ray photoelectron spectroscopy (XPS), in-situ Raman spectroscopy, and density functional theory (DFT) calculation results suggest that the lattice oxygen activity of NiFe (oxy)hydroxide is effectively modulated by Mo doping, resulting in the shift of reaction pathway and a significantly improved intrinsic OER activity. The approach used in this work can be easily adapted for synthesizing (oxy)hydroxide with controlled oxygen activity for other reactions such as $H_2O_2$

generation and biomass conversion in energy and environmental applications.

## Results

**Synthesis and characterization of catalysts.** Ultrathin NiFe (oxy) hydroxide with Mo doping was synthesized using a sacrificial template-directed synthesis approach, as shown in Fig. 1a. First, $MoS_2$ nanosheets were grown on carbon cloth substrates by a hydrothermal reaction (Fig. 1b), which were then used as the template to physically adsorb Ni and Fe ions from the solution. After drying the material in air, we obtained a hetero-structured pre-catalyst, which consists of an ultra-thin layer of NiFe layered double hydroxide (LDH) coated on 1 T phase $MoS_2$ nanosheets (Supplementary Figs. 1, 2). The ratio of Ni and Fe in the LDH can be easily controlled by varying the ion ratio in the solution. The as-synthesized $MoS_2$/NiFe LDH pre-catalysts were then subjected to cyclic voltammetry (CV) to remove the $MoS_2$ sacrificial template through the Mo leaching process.

The final structure we obtained were ultra-thin $(Ni_{1-x}Fe_x)$ (oxy) hydroxides with Mo doping (MoNiFe-x% (oxy)hydroxide, x = 0%, 5%, 27%, 50%, 85%, 100%) on the carbon fiber substrates. Fig. 1c–h show the characterization results for a representative MoNiFe-27% sample with Ni: Fe = 73: 27. The scanning electron microscopy (SEM) image (Fig. 1c) shows that the (oxy)hydroxide layer uniformly covers the carbon fiber. The atomic force microscopy (AFM) results of the (oxy)hydroxide flake prepared by ultrasonic treatment suggest that the active catalyst was ultrathin with an atomic thickness of 0.8 nm (mono-layer, denoted as 1 L) or 1.5 nm (double-layer, denoted as 2 L) (Fig. 1d). Using inductively coupled plasma-optical emission spectrometry (ICP-OES), the Ni, Fe, and Mo contents of the obtained catalyst were determined to be 48.7, 19.4, and 0.11 $\mu g/cm^2$, respectively, indicating a small concentration (0.1%) of Mo doping in MoNiFe-27% (oxy)hydroxide (Supplementary Fig. 3). The presence of Mo dopant in NiFe (oxy)hydroxide was further confirmed by the aberration-corrected high-angle annular dark-field scanning transmission electron microscope (HAADF-STEM) (Supplementary Fig. 4). Because of the low loading mass of MoNiFe (oxy)hydroxide, we could not determine the crystal structure of MoNiFe-27% (oxy)hydroxide using X-ray diffraction measurement (Supplementary Fig. 5). We relied on the transmission electron microscopy (TEM) measurement to confirm the formation of (oxy)hydroxide phase. Figure 1e, f show the TEM images of MoNiFe-27% (oxy)hydroxide flakes. The spacing between two adjacent lattice planes is quantified to be 0.2 nm (Fig. 1f), which is assigned to the (105) plane of oxyhydroxide. The selected area electron diffraction (SAED) pattern of the (oxy)hydroxide in Fig. 1g exhibits clear diffraction rings of (101), (110) plane for Ni-based hydroxide (PDF-#14-0117) and (105), (006) plane for Ni-based oxyhydroxide (PDF-#06-0075). The energy dispersive spectroscopy (EDS) mapping revealed the uniform distribution of O, Fe, Ni, and Mo elements in MoNiFe (oxy)hydroxide (Fig. 1h). MoNiFe-x% (oxy)hydroxide samples with other Fe contents exhibited very similar structure characteristics (Supplementary Fig. 6). All the MoNiFe-x% exhibited similar Mo content (Supplementary Fig. 7).

**Oxygen evolution reaction performance evaluation.** Having confirmed the successful synthesis of MoNiFe (oxy)hydroxide by using $MoS_2$ nanosheets as sacrificial templates and Mo sources, the OER activities of the obtained catalysts were further evaluated. To reveal the role of Mo dopant, pure NiFe (oxy)hydroxides with various Fe contents were synthesized for comparison, which were denoted as NiFe-x% (x = 0%, 5%, 27%, 50%, 85%, 100%).

We systematically evaluated the OER activity of MoNiFe (oxy) hydroxide and NiFe (oxy)hydroxide reference samples using a

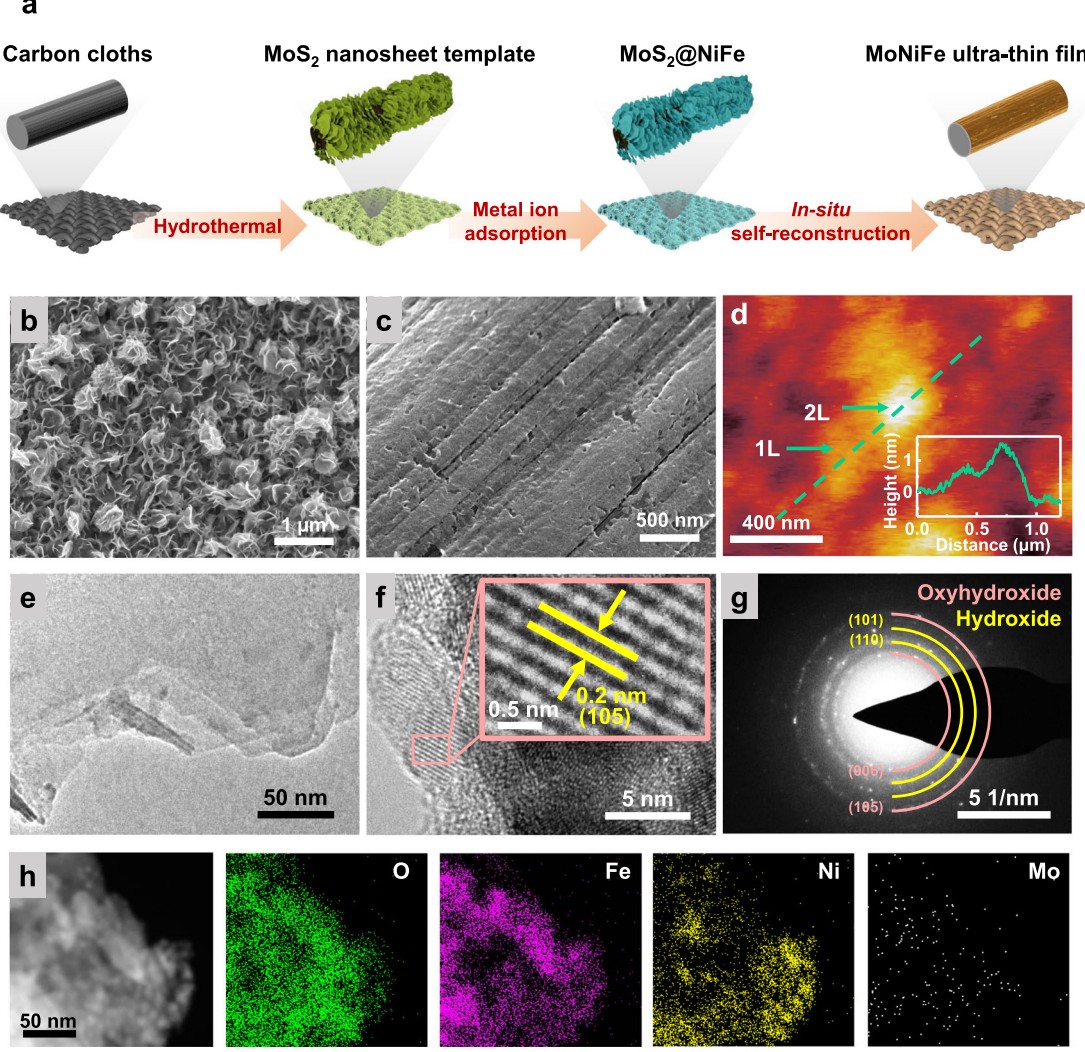

**Fig. 1 Preparation and characterizations of the MoNiFe (oxy)hydroxide. a** Schematic illustration of the preparation process of the MoNiFe (oxy) hydroxide. Scanning electron microscopy (SEM) images of **b** the MoS$_2$ nanosheet template and **c** MoNiFe (oxy)hydroxide. **d** Atomic force microscopy (AFM) image of the MoNiFe (oxy)hydroxide flakes. The inset figure is the corresponding line-trace height profile across a MoNiFe (oxy)hydroxide flake. High-resolution transmission electron microscopy (HRTEM) images with **e** low magnification and **f** high magnification, **g** selected area electron diffraction (SAED) pattern and **h** energy dispersive spectroscopy (EDS) mapping for the MoNiFe (oxy)hydroxide with Ni:Fe ratio of 73: 27.

typical three-electrode configuration in 1.0 M KOH at room temperature. The OER activity of MoNiFe (oxy)hydroxide and NiFe (oxy)hydroxide exhibited similar dependence on the Fe contents. As shown in Supplementary Fig. 8-9, the current density firstly increased with Fe contents, but declined after Fe contents further increased. The optimal Fe content was 27% for both MoNiFe and NiFe (oxy)hydroxide samples. For the samples with the same Fe content, the MoNiFe (oxy)hydroxide exhibited a noticeable higher OER activity than the NiFe (oxy)hydroxide reference samples. For clarity, we show the cyclic voltammetry (CV) polarization curves for the Ni, MoNi, Fe, MoFe, NiFe-27%, and MoNiFe-27% (oxy)hydroxides in Fig. 2a. The MoNiFe-27% (oxy)hydroxide delivered an overpotential of 242 mV at the current density of 10 mA/cm$^2$, which was much lower than the NiFe-27% did (306 mV). To reach a current density of 100 mA/cm$^2$, the MoNiFe-27% (oxy)hydroxide required only 290 mV overpotential. Electrochemical impedance spectra (EIS) and Tafel curves were also measured to evaluate the OER kinetics. The semicircle of EIS curves for the MoNiFe (oxy)hydroxide samples was much smaller than that of NiFe (oxy)hydroxide with the same Fe content (Supplementary Fig. 10), indicating a smaller

charge transfer resistance after Mo doping. The Tafel slope for the MoNiFe-27% (oxy)hydroxide was 23 mV/dec, which was the smallest among all samples (Fig. 2c), indicating its fastest OER kinetics.

We further quantified the mass activity of the electrocatalysts by normalizing the CV curves using loading mass obtained from ICP-OES results (Supplementary Fig. 11). Consistently, the MoNiFe (oxy)hydroxide showed significantly higher mass activity than that for the NiFe (oxy)hydroxide. Particularly, the MoNiFe-27% (oxy)hydroxide exhibited the highest mass activity among all samples, with a current density of 1910 A/g at the overpotential of 300 mV. Such high mass activity of MoNiFe-27% (oxy)hydroxide is about 60 times higher than that of NiFe-27% (oxy)hydroxide (Fig. 2b). In addition, MoNiFe-27% (oxy)hydroxide also delivered a noticeable lower overpotential and higher mass activity than the benchmark RuO$_2$ and IrO$_2$ catalysts (Supplementary Fig. 12, note 1). The high mass activity of MoNiFe (oxy)hydroxide is attributed to the following two reasons: first, the ultra-thin nature of the catalyst layer enables the full exposure of active sites and strongly facilitates the charge transfer process between the conductive substrate and the catalyst; secondly, as will be shown

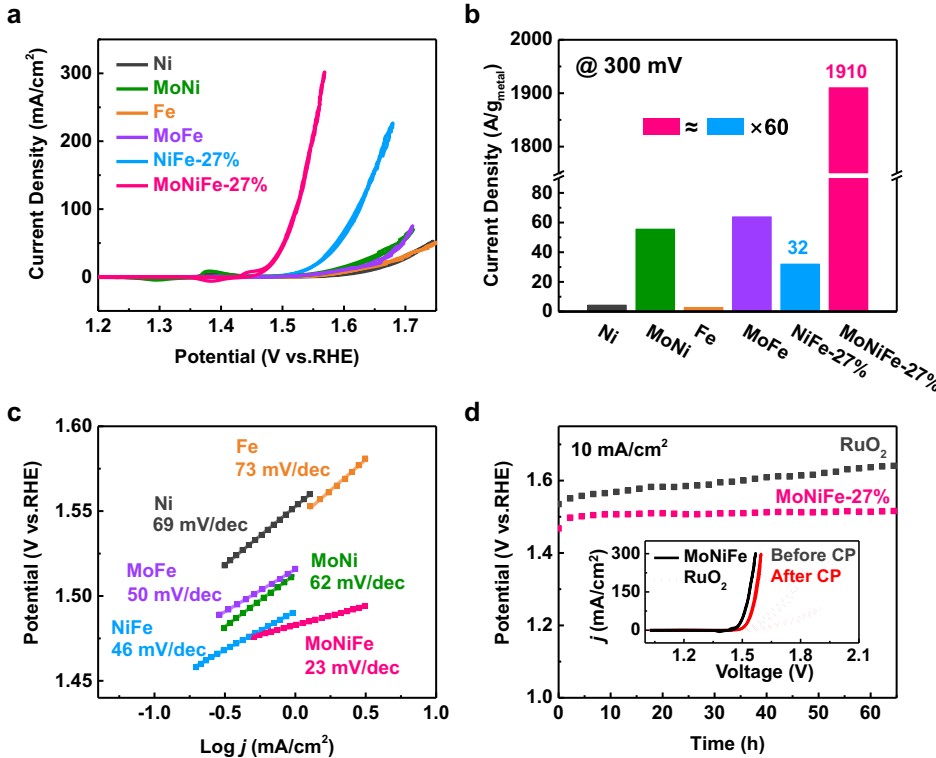

**Fig. 2 OER performance of the NiFe and MoNiFe (oxy)hydroxides. a** Cyclic voltammetry polarization curves, **b** mass activities and **c** Tafel curves of Ni, MoNi, Fe, MoFe, NiFe-27%, MoNiFe-27% (oxy)hydroxide. **d** Chronopotentiometry curves at the current density of 10 mA/cm² for MoNiFe-27% (oxy) hydroxide and commercial RuO₂. The inset figure is the corresponding cyclic voltammetry polarization curves before and after chronopotentiometry measurement.

in the later section, Mo doping can effectively modulate the lattice oxygen activity of NiFe (oxy)hydroxide, leading to the strongly enhanced intrinsic OER activity.

The stability of MoNiFe (oxy)hydroxide was evaluated by chronopotentiometry (CP) tests at the current density of 10 mA/cm². As shown in Fig. 2d, the MoNiFe-27% (oxy)hydroxide displayed significantly better stability than the commercial RuO₂ catalyst. The inset figure in Fig. 2d shows the CV curves of MoNiFe (oxy)hydroxide and RuO₂ before and after CP measurement. The decline of OER performance for MoNiFe-27% (oxy)hydroxide was much smaller than that of RuO₂. The structure and composition of MoNiFe (oxy)hydroxide catalyst remain unchanged after the long-time operation, as revealed by SEM, TEM, EDS, XPS, and ICP-OES characterizations (Supplementary Fig. 13-17, note 2). It is reported that the OER stability of (oxy)hydroxide is strongly dependent on its structural charcteristics[28,29]. Chen et al.[28] reported that the slow diffusion of proton acceptors within interlayer in NiFe hydroxide could lead to a local acidic environment. This can be one primary reason for the performance degradation of multilayer NiFe hydroxide due to the local etching process. The ultra-thin nature of our MoNiFe (oxy)hydroxide can effectively prevent such local etching, and therefore is beneficial for the catalyst to remain stable during operation in alkaline solution.

**Elucidation of the OER mechanism.** To reveal the mechanism of the high intrinsic OER activity of MoNiFe (oxy)hydroxides, isotope-labeling experiments and DFT calculations were carried out on NiFe-27% and MoNiFe-27% (oxy)hydroxide. For simplicity, the NiFe-27% and MoNiFe-27% (oxy)hydroxide will be referred to as NiFe and MoNiFe (oxy)hydroxide in the following context.

As mentioned above, the OER on NiFe-based (oxy)hydroxides was reported to follow the LOM, in which lattice oxygen directly participates in the OER reactions[14,30]. To validate the participation of lattice oxygen in OER for our material systems, the $^{18}O$ isotope-labeling experiments were carried out using the procedure described in the experimental section. In-situ differential electrochemical mass spectrometry (DEMS) measurements results on the $^{18}O$-labeled NiFe and MoNiFe (oxy)hydroxide showed the signals of m/z = 32, m/z = 34, and m/z = 36 (Supplementary Fig. 18), suggesting the presence of $^{16}O_2$, $^{16}O^{18}O$, and $^{18}O_2$ in the gas production[31–33]. This result implies that both NiFe and MoNiFe (oxy)hydroxide follow the LOM mechanism[14,30]. The mass spectrometric cyclic voltammograms (MSCVs) which plot the real-time gas product contents as a function of applied potential can provide direct comparison about the participation of lattice oxygen in OER process. The $^{18}O$-labeled MoNiFe (oxy)hydroxide is with noticeably higher contents of $^{16}O^{18}O$ and $^{18}O_2$ in the reaction product than the $^{18}O$-labeled NiFe (oxy)hydroxide (Fig. 3a–d and Supplementary Fig. 19-20), implying the lattice oxygen of MoNiFe (oxy) hydroxide participated more actively into the OER reaction than that of NiFe (oxy)hydroxide.

In addition to the DEMS measurement, the quasi in-situ Raman spectra were also used to confirm the participation of lattice oxygen in OER. The samples were first activated in electrolyte with $^{18}O$ to form $^{18}O$-NiOOH species, and then were subjected to a positive potential (1.65 V vs. RHE) in electrolyte with $H_2{}^{16}O$. The Raman peaks of the samples activated in electrolyte with $^{18}O$ (named as $^{18}O$-labelled sample) shifted to lower wavenumber comparing to that of the samples activated in electrolyte with $^{16}O$ (named as $^{16}O$-labelled sample), because of the impact of oxygen mass on the

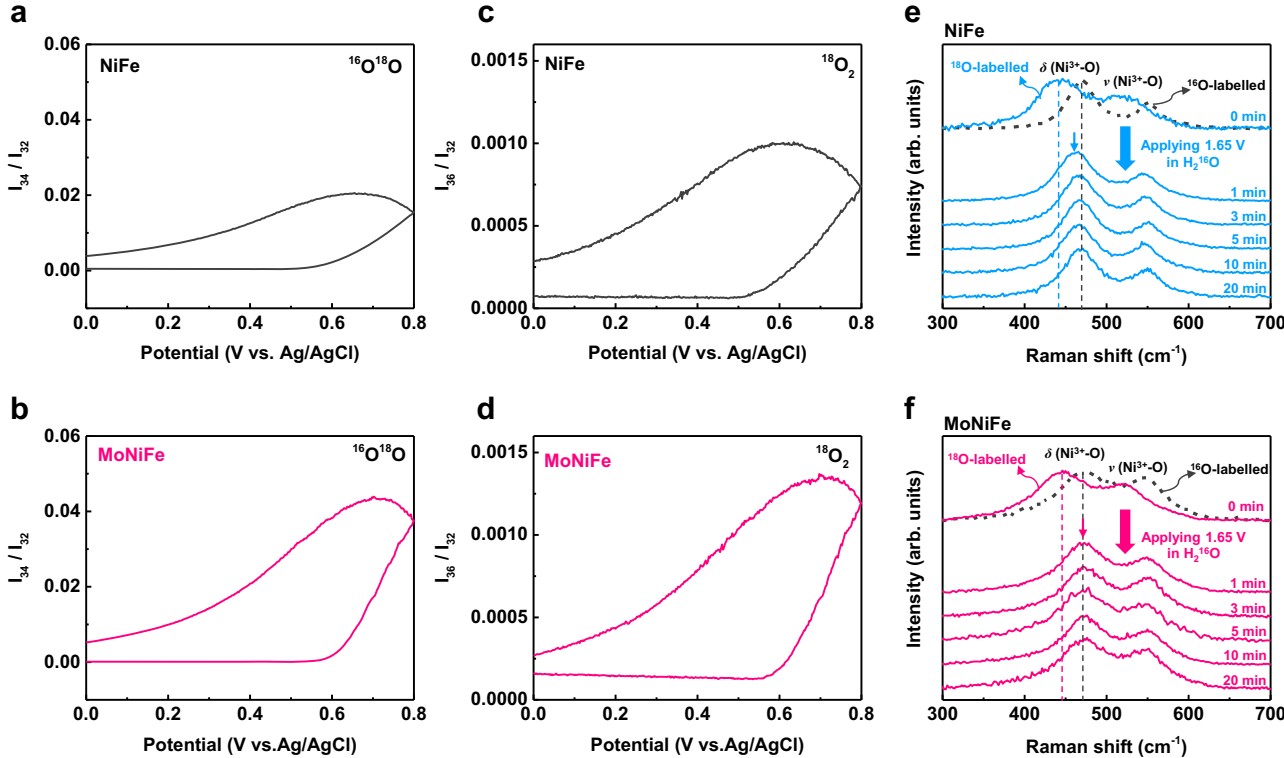

**Fig. 3 Evidence of lattice oxygen participating in OER provided by $^{18}O$ isotope-labeling experiments.** Mass spectrometric cyclic voltammograms results showing different gaseous products content of OER reaction as a function of applied potential for the $^{18}O$-labeled samples: $^{16}O^{18}O$ for **a** NiFe (oxy) hydroxide and **b** MoNiFe (oxy)hydroxide, $^{18}O_2$ for **c** NiFe (oxy)hydroxide and **d** MoNiFe (oxy)hydroxide. The contents of all the species were normalized by the amount of $^{16}O_2$ in the reaction products. Quasi in-situ Raman spectra of **e** $^{18}O$-labelled NiFe and **f** $^{18}O$-labelled MoNiFe (oxy)hydroxides after being applied a positive potential of 1.65 V (vs. RHE) in 1.0 M KOH with $H_2^{16}O$ for different time (1 min to 20 min). The Raman spectra of $^{16}O$-labelled samples were shown in black dash lines for comparison.

vibration mode[2,34](Fig. 3e, f). This result suggests that we successfully labelled both NiFe and MoNiFe samples with $^{18}O$. Then, the $^{18}O$-labelled (oxy)hydroxides were placed in electrolyte with $^{16}O$ and were treated by applying a positive potential of 1.65 V (vs. RHE) for different periods of time (1 min to 20 min). The Raman spectra of the obtained samples are shown in Fig. 3e, f. The Raman peak of $^{18}O$-labelled MoNiFe (oxy)hydroxide shifts back to the position for $^{16}O$-labelled MoNiFe (oxy)hydroxide within 1 min of treatment, which is much faster than that for the NiFe (oxy) hydroxide (20 min). This result suggests that while both samples follow the LOM mechanism, the MoNiFe (oxy)hydroxide exhibits much higher rate of oxygen exchange between lattice oxygen and electrolyte. The Raman spectra and DEMS results on the $^{18}O$-labeled samples consistently suggest that Mo doping in NiFe (oxy)hydroxide effectively promotes the lattice oxygen to participate in the OER reaction.

In additional to $^{18}O$ isotope-labeling experiments, DFT calculations were also carried out to identify the OER mechanism on NiFe and MoNiFe (oxy)hydroxide. Both adsorbate evolution mechanism (AEM) pathway (Supplementary Fig. 21a) and LOM pathway (Fig. 4a) of OER were considered. In the AEM pathway, the Fe sites were found to be the active sites with lower barriers than Ni sites (Supplementary Fig. 21b, c). The deprotonation of *OH in AEM pathway serves as the potential determining step (PDS) for both NiFe and MoNiFe (oxy)hydroxide, with a barrier of 1.05 eV and 0.76 eV, respectively. In the LOM pathway, the (oxy)hydroxides first go through the deprotonation process to form oxyhydroxide (step 1) (Fig. 4a). The exposed lattice oxygen then receives OH$^-$ via nucleophilic attack to form *OOH (step 2). After the deprotonation of *OOH (step 3), gaseous $O_2$ releases

from the lattice, and an oxygen vacancy is generated on the surface (step 4). The resulting oxygen vacancy sites are refilled by OH$^-$ and the surface is recovered (step 5). The calculated Gibbs free energy diagrams of OER on NiFe and MoNiFe (oxy) hydroxide are displayed in Fig. 4b. For the NiFe (oxy)hydroxide, the desorption of $O_2$, which was accompanied by the formation of oxygen vacancy, was found to be the PDS with a high energy barrier of 0.75 eV. In contrast, the barrier of oxygen vacancy formation became much smaller after Mo doping, which pushed the PDS on MoNiFe (oxy)hydroxide to the deprotonation of *OOH with a decreased energy barrier of 0.42 eV. It is noted that both the barriers of PDS for NiFe and MoNiFe (oxy)hydroxide in LOM pathway were much lower than that in AEM pathway, suggesting that both the NiFe and MoNiFe (oxy)hydroxide follow the LOM mechanism[14,30]. This result is consistent with the results of the $^{18}O$ isotope-labeling experiments. The changes in mechanism and PDS derived from Mo doping are quite different from other cation doping reported by previous works[35–37].

The DFT results suggest that Mo doping shifts the PDS from oxygen vacancy formation to the deprotonation of *OOH. Such a transition of PDS from the one involving only lattice oxygen to the one involving surface proton transfer might result in enhanced dependence of the OER activity on the proton activity in solution. Therefore, to confirm such a shift of PDS by Mo doping as revealed by calculations, we evaluated the dependence of OER activity of NiFe and MoNiFe(oxy)hydroxide on proton activity by carrying out pH dependence measurements and deuterium isotopic labeling experiments.

The OER activities of NiFe and MoNiFe (oxy)hydroxide were assessed at different pH conditions (pH = 11.78, 12.75, 12.91,

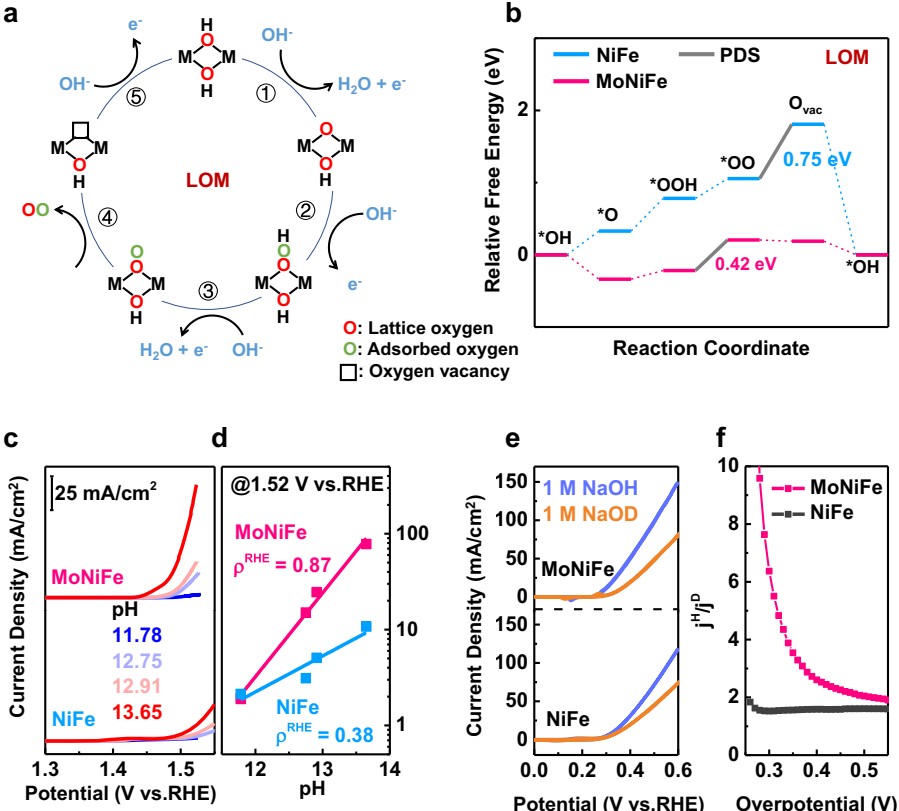

**Fig. 4 OER mechanism revealed by DFT calculation, pH dependence, and deuterium isotopic labeling experiments. a** Schematic illustration and **b** Gibbs free energy diagrams of the LOM pathway on NiFe and MoNiFe (oxy)hydroxide. **c** Linear sweep voltammetry (LSV) curves for NiFe and MoNiFe (oxy) hydroxide measured in KOH with pH = 11.78, 12.75, 12.91, and 13.65. **d** OER current density at 1.52 V versus RHE plotted in log scale as a function of pH, from which the proton reaction orders ($\rho^{RHE} = \partial \log i / \partial pH$) were calculated. **e** LSV curves for NiFe and MoNiFe (oxy)hydroxide measured in 1 M NaOH and 1 M NaOD solution. The LSV curves are without iR compensation. **f** The kinetic isotope effect of MoNiFe and NiFe (oxy)hydroxide. $j^H$ and $j^D$ are referred to the current density measured in NaOH and NaOD solution, respectively.

and 13.65) (Fig. 4c,d). Fig. 4d shows the OER current density at 1.52 V (vs. RHE) in log scale as a function of pH, from which the proton reaction orders on RHE scale ($\rho^{RHE} = \partial \log i / \partial pH$) are calculated to be 0.38 and 0.87 for NiFe and MoNiFe (oxy) hydroxide, respectively. The higher $\rho^{RHE}$ for MoNiFe (oxy) hydroxide implied a stronger pH-dependent OER activity, which might be due to the higher degree of decoupled proton-electron transfer during the PDS step, i.e., the deprotonation of *OOH. (Supplementary note 3, Supplementary Fig. 22-23)[8,9,38]. To further prove the impact of proton activity, the OER activity of NiFe and MoNiFe (oxy)hydroxide were also evaluated in the NaOD and NaOH solution. The LSV curves for NiFe and MoNiFe (oxy)hydroxide measured in 1 M NaOH (dissolved in $H_2O$) and NaOD (dissolved in $D_2O$) solution are shown in Fig. 4e. To show the kinetic isotope effect (KIE) for NiFe and MoNiFe (oxy)hydroxide clearly, the ratio of current density obtained in NaOH and in NaOD at the given potential[39,40] is plotted in Fig. 4f. MoNiFe (oxy)hydroxide exhibited a noticeably larger KIE value in comparison to NiFe (oxy)hydroxide, suggesting a severe degradation of OER activity in NaOD. This result suggests that proton transfer has a greater impact on the OER process on MoNiFe (oxy)hydroxide than that on NiFe (oxy)hydroxide. The deuterium isotopic experiments performed in NaOH/NaOD with a different concentration of 0.5 M provided consistent results (Supplementary Fig. 24). The large isotopic effect of MoNiFe (oxy)hydroxide suggests that the proton transfer is involved in the PDS. This conclusion is in accord with the DFT calculation results, which show that the

PDS step of OER on MoNiFe (oxy)hydroxide is the deprotonation of *OOH (Fig. 4b).

**Analysis of lattice oxygen activity**. Further insight into the underlying reason for the shift of the reaction pathway was deduced by analyzing the oxygen activity of the NiFe and MoNiFe (oxy)hydroxide based on DFT calculations of electronic structures.

The oxygen activity can be represented by the metal-oxygen bond strength, which was evaluated by calculating the crystal orbital Hamilton populations (COHP) by DFT (Fig. 5a and Supplementary Fig. 25)[26,41]. The negative and positive values of -COHP correspond to the anti-bonding (grey area) and bonding state (white area), respectively. The occupied anti-bonding states of Ni and Fe 3$d$ band appeared under Fermi level for both NiFe and MoNiFe (oxy)hydroxide (-COHP peak whose energy level is higher than −2.5 eV, Fig. 5a and Supplementary Fig. 25). Owing to the upshift of the O 2$p$ band, MoNiFe (oxy)hydroxide shows a more significant overlap between the O 2$p$ band and Ni 3$d$ and Fe 3$d$ anti-bonding state under Fermi level than NiFe (oxy) hydroxide. To quantify the metal-oxygen bond strength, the integral of -COHP up to the Fermi level (-IpCOHP$_{Fermi}$) of Ni-O (Fe-O) bonding were determined to be 2.41 (1.78) and 1.19 (1.34) for NiFe and MoNiFe (oxy)hydroxide, respectively (Fig. 5b). The lower value of -IpCOHP$_{Fermi}$ for the MoNiFe indicates that Mo doping results in more electrons filled into the anti-bonding orbitals, leading to the weaker metal-oxygen bond. Such a

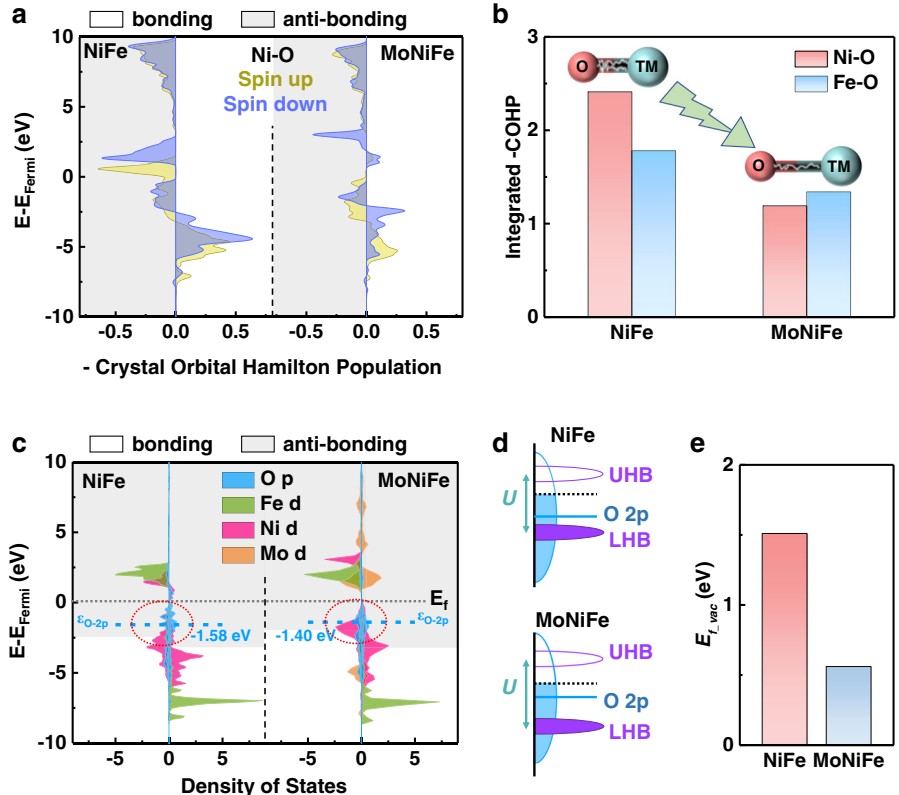

**Fig. 5 Lattice oxygen activity determined by density functional theory (DFT) calculations. a** Crystal orbital Hamilton populations (COHP) of the Ni-O bond in NiFe and MoNiFe (oxy)hydroxide. **b** The integrated -COHP up to Fermi level comparison of Ni-O and Fe-O in NiFe and MoNiFe (oxy)hydroxide. TM refers to transition metal. **c** Projected density of states of NiFe and MoNiFe (oxy)hydroxide. The anti-bonding states below the Fermi level were highlighted by dash circles. **d** Schematic band diagrams of NiFe and MoNiFe (oxy)hydroxide. The $d$-orbitals split into electron-filled lower Hubbard band (LHB) and empty upper Hubbard band (UHB) with an energy difference of $U$. **e** The oxygen vacancy formation energy ($E_{f\_vac}$) of NiFe and MoNiFe (oxy)hydroxide.

weakened metal-oxygen bond can facilitate oxygen vacancy formation.

Secondly, we calculated the O $2p$ band center ($\varepsilon_{O-2p}$) position for both NiFe and MoNiFe (oxy)hydroxide. The density of state (DOS) profile of NiFe and MoNiFe (oxy)hydroxide are shown in Fig. 5c. The O $2p$ band center is determined to be $-1.58$ eV and $-1.40$ eV for NiFe and MoNiFe (oxy)hydroxide, respectively (Fig.5c). The O $2p$ band noticeably shifts toward the Fermi level after Mo doping into NiFe (oxy)hydroxide. The distance between the O $2p$ band center to the Fermi level has been frequently employed as a descriptor for oxygen activity[15,42,43]. It is reported that O $2p$-band center is required to be high enough to guarantee the lattice oxygen to escape from the lattice[43]. The upshift of the O $2p$ band results in deeper penetration of Fermi level into the O $2p$ band, which further facilitates the electron flow away from oxygen sites when an anodic potential is applied, making the lattice oxygen release from the lattice more easily[3,12,43]. As a consequence, oxygen with high O $2p$ band position exhibits facilitated oxygen vacancy formation process and thus promotes the LOM mechanism[42].

In addition to the O $2p$ band position, the Mott-Hubbard splitting in $d$-orbitals was also investigated. For late transition metals, $d$-orbitals can further split into electron-filled lower Hubbard band (LHB) and empty upper Hubbard band (UHB) due to the strong $d$-$d$ Coulomb interaction[3,14]. The LHB/UHB center is determined by the total metal $3d$-orbital distribution below/above $E_{Fermi}$ in DOS diagrams. The specific positions of LHB and UHB were calculated to be $-4.36$ eV and $2.01$ eV for NiFe (oxy)hydroxide, and $-4.67$ eV and $2.90$ eV for MoNiFe

(oxy)hydroxide, respectively. The energy distance between the LHB and UHB band center ($U$) is also an important parameter governing the lattice oxygen activity[3,14]. The $U$ values of NiFe and MoNiFe (oxy)hydroxide were calculated to be 6.38 eV and 7.58 eV, respectively, indicating a stronger $d$-$d$ Coulomb interaction after Mo doping. Such an enlarged $U$ value gives rise to the downshift of LHB (Fig. 5d). As a result, as anodic potential is applied, the electron removal from oxygen sites is strongly facilitated[11,14]. It is noted that the LHB center is located beneath the O $2p$ band center. Therefore, the downshift of LHB center and upshift of O $2p$ band center for MoNiFe (oxy)hydroxide leads to a smaller overlap of metal $3d$-orbital and oxygen $2p$-orbital, which results in the weaker metal-oxygen bond. In addition, the density of states of metal $3d$-orbital, especially for Ni $3d$-orbital, upshift close to Fermi level. Although such upshift lead to an increased overlap between Ni $3d$-orbital and O $2p$-orbital in DOS diagrams, the overlap of O $2p$ - Ni $3d$ orbital occurs on the anti-bonding states below Fermi level as highlighted in dash circles in Fig. 5c and results in a weaker Ni-O bond, which is consistent with the COHP calculations (Fig. 5b).

The weakened metal-oxygen bond, the upshifted O $2p$ band relative to Fermi level, and the enlarged $U$ value in the MoNiFe (oxy)hydroxide in comparison with the NiFe (oxy)hydroxide indicated that Mo doping effectively activated the lattice oxygen, thereby promoting the oxygen vacancy formation process[15,42,43]. To further confirm such impact of Mo doping, we directly calculated the oxygen vacancy formation energy ($E_{f\_vac}$) using DFT. The $E_{f\_vac}$ of MoNiFe (oxy)hydroxide was determined to be 0.56 eV, which is much lower than the 1.51 eV for the NiFe (oxy)

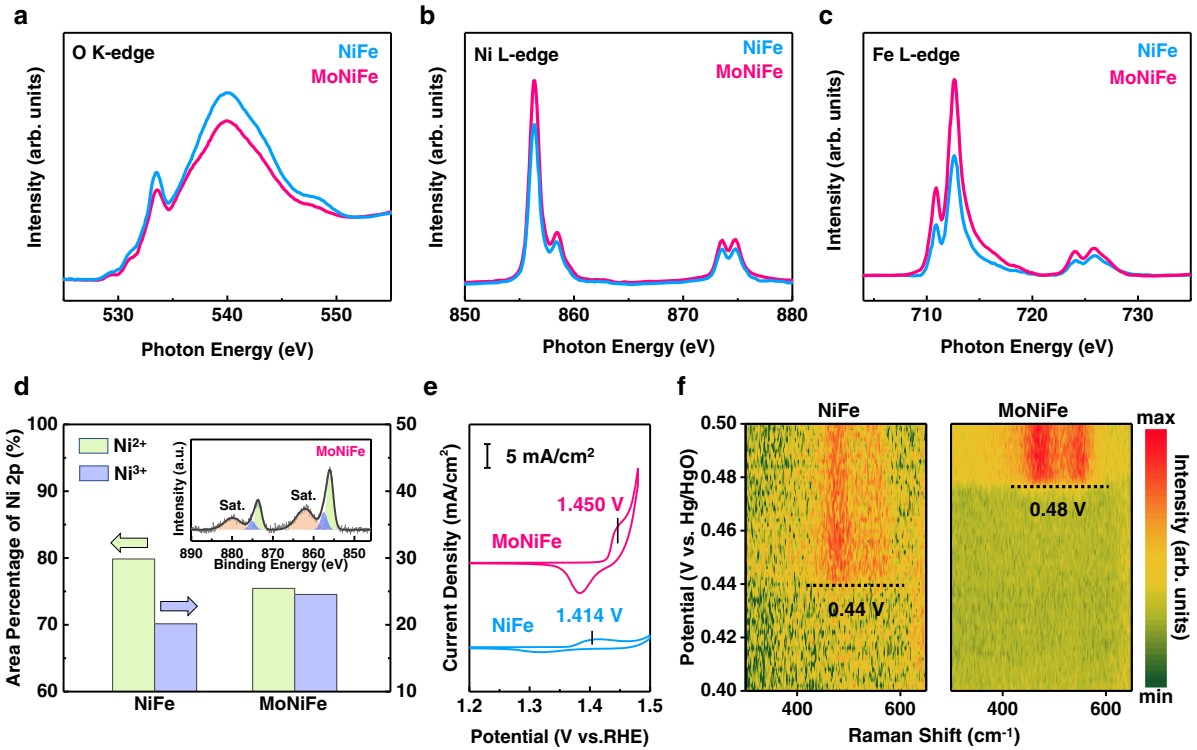

**Fig. 6 Lattice oxygen activity determined by advanced spectroscopy techniques. a** O K-edge, **b** Ni L-edge, and **c** Fe L-edge soft X-ray absorption spectroscopies (sXAS) of NiFe and MoNiFe (oxy)hydroxide. The step at the background of O K-edge spectra was normalized to be 1[46]. The background of Ni L-edge and Fe L-edge spectra were subtracted. The raw data and the background of sXAS spectra are shown in Supplementary Fig. 35-37. **d** Oxidation state of nickel in NiFe and MoNiFe (oxy)hydroxide determined by Ni 2p X-ray photoelectron spectroscopy (XPS). The inset figure is the Ni 2p XPS spectrum of MoNiFe (oxy)hydroxide. **e** Redox peaks of Ni in cyclic voltammetry (CV) curves of NiFe and MoNiFe (oxy)hydroxide. **f** In-situ Raman spectra map of NiFe and MoNiFe (oxy)hydroxide acquired during CV measurement, in which the dash lines mark the required potential for driving the transition from $Ni^{2+}$ to $Ni^{3+}$.

hydroxide, as shown in Fig. 5e. This result was also confirmed by the higher content of defective oxygen in MoNiFe (oxy)hydroxide by O 1 s XPS anlysis (Supplementary Fig. 26, note 4). Our DFT calculation further shows that the LOM pathway is still dominant for both NiFe and MoNiFe (oxy)hydroxide when there is oxygen vacancy presence on the surface (Supplementary Fig. 27-30, note 5).

The DFT results above demonstrated that Mo doping in NiFe (oxy)hydroxide effectively enhanced the oxygen activity. In the following section, we further compare the lattice oxygen activity of NiFe and MoNiFe (oxy)hydroxide experimentally by probing the local density of states around the oxygen ligands, the metal oxidation state, and cationic electrochemical redox process using advanced spectroscopy techniques, including synchrotron-based sXAS, XPS, in-situ Raman spectroscopy.

First, the variation of the local density of states around the oxygen ligands of NiFe and MoNiFe (oxy)hydroxide was detected by carrying out O K-edge sXAS measurement with total electron yield (TEY) mode. The O K-edge sXAS spectra consist of two characteristic peaks at ~533.5 eV and ~540 eV, which were assigned to the O 2p - metal 3d hybridization and the O 2p - metal 4sp hybridization[44,45]. As shown in Fig. 6a, the intensity of O K-edge decreases after Mo doping, indicating a decrease in unoccupied density of states[46] and a weakening of $3d/4sp-2p$ hybridization[44,47]. Such decreased intensity in O K-edge spectra, accompanying with the increased intensity of Ni L-edge and Fe L-edge peak for MoNiFe (oxy)hydroxide (Fig. 6b, c), suggests a higher electron density at the O site and a lower electron density at the Ni/Fe sites, a higher ionic metal-oxygen bond[46,48]. This

result is consistent with the weaker metal-oxygen bond after Mo doping, as revealed by the COHP calculation (Fig. 5b). In addition, the increased electron density on oxygen sites in MoNiFe (oxy)hydroxide might promote the donation of electrons from oxygen as an anodic potential was applied. The O K-edge sXAS result suggests that the Mo doping effectively increased the local density of states around the oxygen ligands, which can potentially give rise to the lattice oxygen activation.

There have been many previous works that demonstrated that the oxygen activity could be probed indirectly by characterizing the metal oxidation state, and a higher oxidation state of the transition metal was normally correlated to an increased oxygen activity[7,14,49]. For example, Grimaud et al.[49] reported that the hybridization of Co-O bonds in perovskites cobaltite increased with cobalt oxidation state, which was correlated to the upshift of O p-band center relative to Fermi level. Mefford et al.[7] also showed that the d-orbitals of cobalt have a greater overlap with the s, p orbitals of oxygen as the cobalt oxidation state increased, leading to the promoted lattice oxygen activity of perovskite cobaltite. Zhang et al.[14] reported that the formation of $Ni^{4+}$ species in Ni-based (oxy)hydroxide can drive holes into oxygen ligands to trigger lattice oxygen activation. In addition, an enlarged U value, a descriptor for enhanced oxygen activity, was reported to be related to the increased valence state of the metal[11,50]. Inspired by these pioneering works, we compared the change of oxidation states of Ni and Fe in (oxy)hydroxide after Mo doping.

The Ni L-edge sXAS spectra of NiFe and MoNiFe (oxy) hydroxide are shown in Fig. 6b. The MoNiFe (oxy)hydroxide

exhibited higher intensity than that of NiFe (oxy)hydroxide. In addition to intensity changes, the Ni L-edge spectra of MoNiFe (oxy)hydroxide shifted to higher photon energy relative to NiFe (oxy)hydroxide (Supplementary Fig. 31). Both the higher intensity and positive shift of Ni L-edge peak for MoNiFe (oxy) hydroxide than NiFe (oxy)hydroxide suggest an increased number of unoccupied density of states on Ni sites[51,52]. Similarly, the Fe L-edge spectra of MoNiFe (oxy)hydroxide exhibited higher intensity than that of the NiFe (oxy)hydroxide, indicating an increased unoccupied density of states on Fe sites (Fig. 6c). The changes in Ni/Fe L-edge spectra derived from the partial electron transfer from Ni/Fe sites to the Mo sites through bridging oxygen (μ-O) in Ni-O-Mo-O-Fe moiety in MoNiFe (oxy)hydroxide (Supplementary Fig. 32, note 6)[36], leading to the electron depletion in metal sites and the increment of metal oxidation state. In addition to the sXAS results, the XPS analysis (Fig. 6d, Supplementary Fig. 33-34) provided the same conclusion of higher metal oxidation state in the MoNiFe (oxy)hydroxide with detailed discussion in Supplementary note 7.

As mentioned above, the upshifted O 2p band and enlarged U value in MoNiFe (oxy)hydroxide would promote the lattice oxygen redox chemistry as an anodic potential is applied. Because of the competition of electron donation from oxygen anion and metal cations redox process, the enhanced oxygen reactivity should be reflected on the delayed cationic electrochemical redox process (Supplementary Fig. 38, note 8)[3]. As shown in Fig. 6e, the $Ni^{2+}/Ni^{3+}$ redox peak for MoNiFe (oxy)hydroxide (1.450 V vs. RHE) shifted positively compared to that of NiFe (oxy)hydroxide (1.414 V vs. RHE), indicating that Ni in (oxy)hydroxide required higher positive potential to oxidize after Mo doping. We further carried out in-situ Raman spectroscopy to confirm such a change of $Ni^{2+}/Ni^{3+}$ electrochemical redox during OER process (Fig. 6f). Two characteristic peaks of $Ni^{3+}$-O were found on the Raman spectra at 476 and 557 $cm^{-1}$ when a sufficiently high positive potential was applied. These two peaks corresponded to the $E_g$ bending vibration ($\delta$(Ni-O)) and $A_{1g}$ stretching vibration ($\nu$(Ni-O)) mode in $\gamma$-NiOOH, respectively[2]. The emergence of the Raman peaks of $Ni^{3+}$-O occurred at 0.44 V and 0.48 V (vs. Hg/HgO) for NiFe and MoNiFe (oxy)hydroxide, respectively. These results suggest that the nickel redox process gets delayed after Mo doping due to the facilitated lattice oxygen oxidation, which is consistent with DFT calculation results.

All the DFT calculations and experimental results above consistently suggest that the lattice oxygen activity was strongly enhanced by Mo doping, leading to the facilitated oxygen vacancies formation. Consistently, we observed the reaction barrier of oxygen vacancy formation for the MoNiFe (oxy) hydroxide to be much smaller than that for the NiFe (oxy) hydroxide (Fig. 4b). Consequently, the PDS transforms from oxygen vacancy formation for the NiFe (oxy)hydroxide to the *OOH deprotonation for the MoNiFe (oxy)hydroxide. These results provide critical insight into the role of lattice oxygen in determining the electrocatalytic activity of transition metal (oxy) hydroxide.

**Overall water splitting performance**. To demonstrate the practical application of MoNiFe (oxy)hydroxide for electrochemical production of hydrogen, a two-electrode electrolytic cell was constructed using $MoS_2$/NiFe LDH pre-catalyst as both anode and cathode for overall water splitting. During the water splitting process, the anode was transformed from the $MoS_2$/NiFe LDH pre-catalyst into MoNiFe (oxy)hydroxide, while the $MoS_2$/NiFe LDH cathode remained unchanged (Supplementary Fig. 39). The final cell structure was denoted as $MoS_2$/NiFe LDH | MoNiFe in the following context. A reference cell with commercial noble-

metal-based catalysts Pt/C and $RuO_2$ as cathode and anode, respectively, was also tested for comparison (denoted as Pt/C | $RuO_2$). The polarization curves of the cell with $MoS_2$/NiFe LDH | MoNiFe coupled electrodes and the reference cell with Pt/C | $RuO_2$ coupled electrodes for overall water splitting in 1 M KOH electrolyte were shown in Fig. 7a. The electrolytic cell with $MoS_2$/NiFe LDH | MoNiFe coupled electrodes presented higher overpotential at low current density than the cell with Pt/C | $RuO_2$ coupled electrodes. Nevertheless, the cell with $MoS_2$/NiFe LDH | MoNiFe coupled electrodes exhibited a noticeable better performance at high current density. To reach a current density of 100 mA/$cm^2$, the cell with $MoS_2$/NiFe LDH | MoNiFe coupled electrodes only required a voltage of 1.728 V, which was significantly lower than the reference cell with Pt/C | $RuO_2$ coupled electrodes (1.755 V).

To achieve sufficient hydrogen production rate, the electrolytic cell needs to operate at a high current density. Therefore, the stability of the electrolytic cell was evaluated at a current density of 10 mA/$cm^2$ and 100 mA/$cm^2$. At a current density of 10 mA/$cm^2$, both the cell with $MoS_2$/NiFe LDH | MoNiFe coupled electrodes and the one with Pt/C | $RuO_2$ coupled electrodes displayed excellent stability (Supplementary Fig. 40). At a current density of 100 mA/$cm^2$, the cell with $MoS_2$/NiFe LDH | MoNiFe coupled electrodes remained stable during operation, while the cell with Pt/C | $RuO_2$ coupled electrodes degraded rapidly (Fig. 7b).

Finally, we compared the cell voltage for $MoS_2$/NiFe LDH | MoNiFe at a high current density of 100 mA/$cm^2$ with recently reported noble-metal-free electrocatalysts for overall water splitting (Fig. 7c, Supplementary Table 1). As shown in Fig. 7c, our cell performance at high current density (1.728 V at 100 mA/$cm^2$) is competitive among the noble-metal-free electrocatalysts reported in the literature.

## Discussion

In this work, a sacrificial template-directed approach was reported to synthesize ultra-thin NiFe-based (oxy)hydroxide with Mo doping as highly efficient and stable OER catalysts. $MoS_2$ nanosheets grown by hydrothermal approach were used as templates to adsorb metal cations to form self-assembly NiFe (oxy) hydroxide and served as Mo sources for doping. The obtained MoNiFe (oxy)hydroxide exhibited a high mass activity of 1910 A/$g_{metal}$ at the overpotential of 300 mV, which is 60 times higher than that of the NiFe (oxy)hydroxide. The electrolytic cell with $MoS_2$/NiFe LDH | MoNiFe coupled electrodes exhibited good activity and stability for the overall water splitting, which required a low voltage of 1.728 V to achieve a current density of 100 mA/$cm^2$. DFT calculation suggested that MoNiFe (oxy)hydroxide exhibited higher lattice oxygen activity, which was represented by the weakened metal-oxygen bond, upshifted O 2p center relative to Fermi level, enlarged U values, and lower oxygen vacancy formation energy. Consistently, synchrotron-based sXAS, XPS, and in-situ Raman measurements demonstrated that the MoNiFe (oxy)hydroxide was with higher local density of states around the oxygen ligands, a higher metal oxidation state, and a delayed cationic electrochemical redox process in comparison with the NiFe (oxy)hydroxide. Such activation of lattice oxygen shifted the potential determining step from oxygen vacancy formation for the NiFe (oxy)hydroxide to the *OOH deprotonation for the MoNiFe (oxy)hydroxide, resulting in strongly enhanced intrinsic OER activity. The methodology used in this work can be easily adapted for constructing other transition metal (oxy)hydroxide with modulated oxygen activity for catalyzing other reactions such as biomass electrooxidation reactions. The mechanistic understanding of the role of lattice oxygen in determining surface

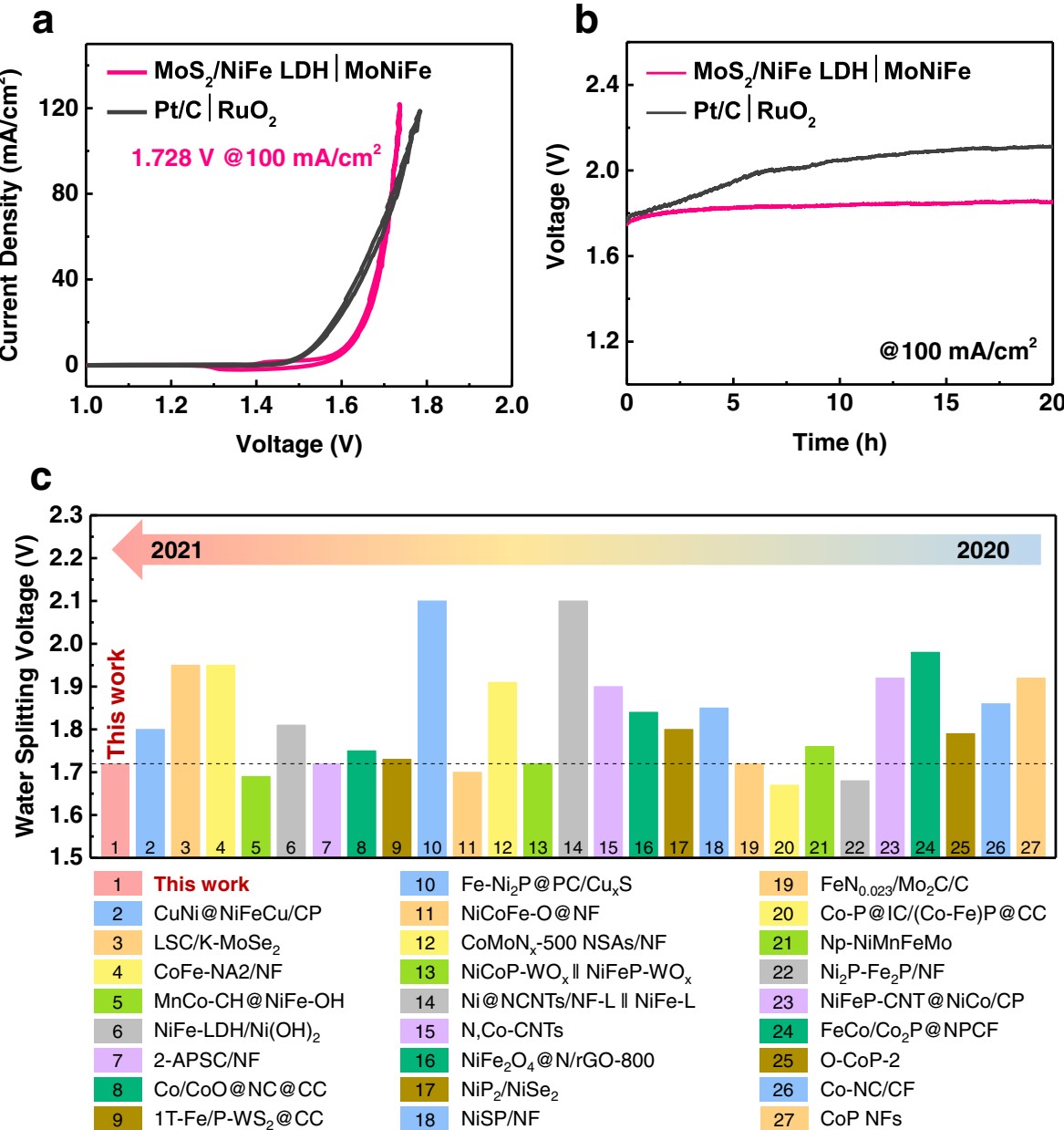

**Fig. 7 The overall water splitting performance. a** Polarization curves of the electrolytic cell with MoS$_2$/NiFe LDH│MoNiFe coupled electrodes and the reference cell with Pt/C│RuO$_2$ coupled electrodes for overall water splitting. **b** Chronopotentiometry curves at the current density of 100 mA/cm$^2$ of the electrolytic cell with MoS$_2$/NiFe LDH│MoNiFe coupled electrodes and the reference cell with Pt/C│RuO$_2$ coupled electrodes. **c** The comparison of overall water splitting performance at the current density of 100 mA/cm$^2$ for MoS$_2$/NiFe LDH│MoNiFe and other noble-metal-free electrocatalysts in recently reported literature, such as MnCo-CH@NiFe-OH (1.69 V)[61], NiFe-LDH/Ni(OH)$_2$ (1.81 V)[62], NiCoFe-O@NF (1.7 V)[63], NiP$_2$/NiSe$_2$ (1.8 V)[64], Ni2P-Fe2P/NF (1.68 V)[65], NiFeP-CNT@NiCo/CP (1.92 V)[66], FeCo/Co$_2$P@NPCF (1.98 V)[67], Co-NC/CP (1.86 V)[68] and CoP NFs (1.92 V)[69].

reactions can guide the rational design of high-performance photo-, thermal-, or electro-catalysts.

## Methods

**Synthesis of MoNiFe (oxy)hydroxide**. MoS$_2$ nanosheets were grown on carbon cloths by a hydrothermal method with ammonium molybdate tetrahydrate [(NH$_4$)$_6$Mo$_7$O$_{24}$·4H$_2$O] and thiourea (CH$_4$N$_2$S) as the precursors. The obtained MoS$_2$ nanosheets were immersed into a mixed solution of nickel acetate and ferrous sulfate to adsorb Fe and Ni ions onto the surface. After drying in air, the MoS$_2$/NiFe LDH pre-catalysts were constructed. The MoS$_2$/NiFe LDH pre-catalysts were subjected to cyclic voltammetry activation in 1 M KOH solution to obtain self-reconstruction Mo doping NiFe (oxy)hydroxide through Mo leaching. The NiFe (oxy)hydroxide reference sample was synthesized by a commonly used wet-

chemical method. Further detailed information about NiFe and MoNiFe (oxy) hydroxides synthesis can be found in Supplementary note 9, 10.

**Characterizations**. The morphologies of samples were characterized by high-resolution field emission scanning electron microscopy (SEM) (SU8010, Hitachi, Japan). The chemical composition was detected by inductively coupled plasma-optical emission spectrometry (ICP-OES) (Agilent 730 series) and X-ray photo-electron spectroscopy (XPS) (Escalab250Xi, Thermo Scientific) with Al anode. High-resolution transmission electron microscopy (HRTEM) images and energy dispersive spectroscopy (EDS) were recorded by a JEM-3200FS microscope. In-situ Raman measurements were performed on a confocal microscopic system (Lab-RAM HR Evolution, Horiba, France) equipped with a semiconductor laser (λ = 532 nm, Laser Quantum Ltd.). The laser was focused using a 50× objective lens and 600 lines/mm grating. The Raman spectra were collected continuously

with a step of 2 mV during linear sweep voltammetry measurement with a scanning rate of 0.1 mV/s. Synchrotron-based soft X-ray absorption spectroscopy (sXAS) was carried out at the BL02B02 station in Shanghai Synchrotron Radiation Facility[53].

**Electrochemical measurements**. The electrochemical measurements were performed in a three-electrode system using a CHI-660E electrochemical station. 1 M KOH aqueous solution was used as the electrolyte, and it was bubbled by $O_2$ for 30 min prior to OER measurements. The catalyst-loaded carbon cloths acted as the working electrode. The reference electrode and counter electrode were a Ag/AgCl electrode prefilled with saturated KCl aqueous solution and a Pt mesh, respectively. All electrode potentials were given versus the reversible hydrogen electrode (vs. RHE) unless otherwise mentioned. The detailed information about the electrochemical measurements can be found in Supplementary note 11.

**$^{18}$O-labeling experiment**. NiFe and MoNiFe (oxy)hydroxides were labeled with $^{18}$O-isotopes by potentiostatic reaction at 1.65 V (vs. Ag/AgCl) for 30 min in KOH solution with $H_2^{18}O$. Afterward, the $^{18}$O-labeled catalysts were rinsed with $H_2^{16}O$ for serval times to remove the remaining $H_2^{18}O$.

**DEMS measurements**. DEMS measurements were carried out using a QAS 100 device (Linglu Instruments, Shanghai). The NiFe or MoNiFe (oxy)hydroxide with $^{18}$O-labeling, a Ag/AgCl electrode prefilled with saturated KCl aqueous solution, and a Pt mesh were used as working electrode, reference electrode, and counter electrode, respectively. CV measurement was performed in KOH solution with $H_2^{16}O$ with a scan rate of 5 mV/s. In the meantime, gas products with different molecular weights were detected in real time by mass spectroscopy.

**Theoretical calculation**. Spin-polarized DFT calculations were performed using the Vienna ab initio simulation package (VASP)[54]. The generalized gradient approximation (GGA) of the Perdue-Burke-Ernzerhof (PBE) version[55] was used to describe the exchange-correlation interactions. The projector-augmented wave (PAW) method is used to model core-valence electron interactions[56]. The COHP of considered atomic pairs was calculated by the Lobster code[57–60]. The detailed information about the DFT calculation can be found in Supplementary note 12.

## Data availability

The data that support the findings of this study are available from https://figshare.com/s/489adcc0875ef42536c8. Source data are provided with this paper.

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

## Acknowledgements

This work was supported by the National Natural Science Foundation of China (11975102, Y.C.); the State Key Laboratory of Pulp and Paper Engineering (2020C01, Y.C.); the Guangdong Pearl River Talent Program (2017GC010281, Y.C.). The synchrotron experiments were carried out at Beamline 02B of the Shanghai Synchrotron Radiation Facility, which is supported by ME2 project under contract from the National Natural Science Foundation of China (11227902, N.Z.). S. Zhao acknowledges the support from the City University of Hong Kong (No. 9610425, S.Z.). The computational time provided by the Shanghai Supercomputer Center and the CityU Burgundy Supercomputer is highly acknowledged.

## Author contributions

Z.H. and Y.C. conducted the experiments and analyzed the results. J.Z. and S.Z. are responsible for the DFT calculations. Z.H., Z.G., D.Z., N.Z., and Y.C. are responsible for the testing and analysis of XAS. Z.H., H.L., W.M., and Y.C. are responsible for the in-situ Raman measurement and analysis. Z.H., J.Z., S.Z., and Y.C. planned and designed the project and wrote the manuscript.

## Competing interests

The authors declare no competing interests.
