## [Peer Review File · Nature Communications]

Activating lattice oxygen in NiFe-based (oxy)hydroxide for water electrolysisREVIEWER COMMENTS

Reviewer #1 (Remarks to the Author):

This manuscript reported a sacrificial template-directed approach to synthesize Mo-doped NiFe (oxy)hydroxide with modulated oxygen activity as an outstanding electrocatalyst towards oxygen evolution. Density functional theory calculations and spectroscopy characterizations both confirm that the Mo dopant facilitates the lattice oxygen activation and LOM pathway for OER. However, the following questions should be seriously addressed before its publication on Nature Communications.

(1) The lattice oxygen oxidation is directly related to the detection of $^{18}\text{O}^{16}\text{O}$ ($m/z=34$). Therefore, the ^{18}O isotope-labeling experiments should be measured by using differential electrochemical mass spectrometry (DEMS), which provide direct evidences of lattice oxygen oxidation (Nature Chem. 2017, 9, 457; J. Am. Chem. Soc. 2021, 143, 6482; Joule 2021, 5, 2164).

(2) The OER activity comparison in D_2O and H_2O solution is an effective way to prove the impact of proton activity. The isotopic experiments are helpful for determining whether the proton/hydrogen transfer is involved in the rate-determining step (RDS). The RDS of MoNiFe is the deprotonation of $^*\text{OOH}$ (Figure 3b), which cannot be substantiated by the different performance of MoNiFe in D_2O and H_2O solution (Figure 3e). To scientifically determine the RDS, the kinetic isotope effect (KIE) of MoNiFe and NiFe should be calculated (Angew. Chem. Int. Ed. 2021, 60, 3095).

In other hand, it may be more scientific that isotopic experiments are performed in dissolved KOD with D_2O rather than KOH with D_2O .

(3) The AEM pathway of NiFe and MoNiFe oxyhydroxide should also be calculated to fully prove the favorable kinetics of LOM pathway (Nat. Commun. 2020, 11, 4066). It should be noted that the in-situ Raman spectra have confirmed that the lattice oxygen of NiFe LDH wasn't involved in OER (Angew. Chem. Int. Ed. 2020, 59, 8072). In other words, the AEM pathway of NiFe may be more thermodynamically favorable than LOM of NiFe. Therefore, it is improper to only evaluate the energy barriers by using LOM process.

Besides, the optimized DFT models of OER process should be provided for confirming the rationality of schematic illustration of the proposed OER pathway (Figure 3a).

(4) The density of states calculations revealed that there was a upshift of O 2p band center after Mo doping. Meanwhile, the enlarged energy distance between the LHB and UHB band center leads to the downshift of LHB. Therefore, the Mo incorporation strengthened the metal-oxygen bonds and enhanced the metal-oxygen covalency. However, the COHP calculations indicated the weaker metal-oxygen bond

of MoNiFe (oxy)hydroxide comparing to that of NiFe (oxy)hydroxide. The authors should explain the contradictive conclusions for DOS and COHP calculations.

(5) The U values of NiFe and MoNiFe (oxy)hydroxide were calculated to be 6.38 eV and 7.58 eV, respectively. To validate the schematic band diagrams of NiFe and MoNiFe (oxy)hydroxide, the specific positions of LHB and UHB calculation and the details of U values calculations should be provided in this manuscript.

(6) The crystal structures of catalysts have been characterized by HRTEM and SAED. However, no X-ray diffraction (XRD) patterns of catalysts were provided in this manuscript. The authors should do the XRD measurement for the further structure characterizations.

(7) With regard to XPS data, only Ni 2p spectra of catalysts were analyzed. The Fe 2p, Mo 3d and O 1s spectra should also be measured for the elucidation of chemical states.

(8) In Figure 1c, it is unconvincing that the ultrathin (oxy)hydroxide layer uniformly covered the carbon fiber. In fact, there are evident morphology difference between MoNiFe/CC and ultrathin MoS₂ nanosheets (Figure 1b). To clearly show the ultrathin layer morphology, high-definition SEM images should be provided.

(9) In this manuscript, the Mo dopant is responsible for the activity improvement. Therefore, the performance differences among samples with various Fe contents may arise from the amount variation of Mo dopant. The authors should precisely determine the metal contents of different samples by ICP spectroscopy.

(10) It should be noted that the structures of catalysts may have considerable changes after OER. The previous reports reveal that Mo element thoroughly leached from the electrodes and the reconstructed Ni(Fe) oxyhydroxides are responsible for the electrocatalysis (ACS Nano 2021, 15, 13504; Adv. Funct. Mater. 2021, 31, 2101792; Adv. Mater. 2021, 33, 2007344). To determine the active structure, characterization of catalysts should be measured after OER, such as SEM, TEM, XRD, XPS and Raman spectra.

Reviewer #2 (Remarks to the Author):

Recommendation: It appears that publication in any form would be premature at this time.

The authors focused on Mo-doped NiFe (oxy)hydroxides, which is a strong candidate of a catalyst for the oxygen evolution reaction (OER). In their study, the Mo-doped NiFe (oxy)hydroxides were synthesized using the sacrificial template-directed approach. It is suggested that Mo-doped NiFe (oxy)hydroxides can be followed the lattice oxygen mechanism (LOM), and the authors clarified the importance of oxygen activity in the material lattice structure as an electrocatalyst for OER. They applied the various analyses to reveal the role of oxygen in atomic level and electric states by using both experimental and theoretical methods. The ideas are interesting and the findings in this work can be applied for a guideline for future catalyst design; however, I cannot provide my final recommendation, because some key information is missing. Therefore, please address the issues below and I need to read a new version to make my final recommendation.

1) Are there any previous studies on the sacrificial template-directed approach? Is this method originally developed in the authors group? Please add the information in Introduction section.

Also, are there any previous studies on Mo-doped NiFe (oxy)hydroxide or other metal-doped NiFe (oxy)hydroxide? Why did the authors choose Mo as a dopant?

Authors should mention more clearly on the purpose of the work (which part is the novel approach?) and the reason why they choose it.

2) On Fig. 4b, the meaning of "TM" is necessary in figure caption.

3) On Fig. 5 a-c of sXAS results, authors should add the reference "Rojas, T. C. et al. J. Mater. Chem. 9, 1011-1017 (1999)" in addition to Ref#33, because the corresponding discussion in Ref#33 refers the paper. If possible, the explanation of the result of Fig. 5a is needed, especially about the relationship of the sXAS result and the discussion of "an increased electron density around the oxygen ligands after Mo doping."

4) The authors should show the raw data of sXAS and the information of their background data, which is subtracted from the raw data, in Supporting Information. The background data is important for the reliability of the data, because the results in Fig. 5a-c have the clear difference only in peak intensity.

5) Do the authors have data for benchmark on the mass activity? (for example, commercial RuO₂ or IrO₂. I think commercial IrO₂ is more popularly used for a benchmarking.)

Reviewer #3 (Remarks to the Author):

The manuscript presents combined experimental and computational results on Mo-doped NiFe oxyhydroxide with modulated oxygen activity for alkaline oxygen evolution reaction (OER). The system and the findings are of potential interest to the field in general and to readers of Nature Commun. specifically, but the information provided in several key areas is insufficient to render sufficient support for the conclusions drawn. Therefore, the manuscript does not warrant publication in the present form, as discussed below.

The paper fails to discuss, let alone cite, key papers on the specific topic of lattice oxygen like Chorkendorff et al. "Impact of nanoparticle size and lattice oxygen on water oxidation on NiFeOxHy", Nat. Catal., DOI:10.1038/s41929-018-0162-x (2018). More importantly regarding the proposed role of the lattice oxygen, Fig 3. states: "and isotopic labeling experiments", but where are the oxygen isotope (^{16}O vs. ^{18}O) experiments? These results should be provided to support the proposed mechanism and discussed against relevant literature.

Similarly, the work by Bent et al. on "The Role of Aluminum in Promoting Ni-Fe-OOH Electrocatalysts for the Oxygen Evolution Reaction", ACS Appl. Energy Mater (2019), DOI: 10.1021/acsaem.9b00265 would provide a valuable comparison for the Mo-doped ultra-thin NiFe LDH system.

Further, the level of information provided regarding the DFT level calculations is insufficient. It is impossible to reproduce the calculations based on the information provided. What is the model system and how is it terminated? Where is Mo-placed? What reference is used to calculate the formation energy of the oxygen vacancy (O_2 gas or an H_2O -reference), etc? This data is needed to assess and discuss the findings.

Finally, the manuscript: "From 3D to 2D Co and Ni Oxyhydroxide Catalysts: Elucidation of the Active Site and Influence of Doping on the Oxygen Evolution Activity" (not cited either) by Vegge et al., ACS Catal. (2017), DOI: 10.1021/acscatal.7b02712. shows the variations in overpotentials resulting from the specific doping sites and whether the catalyst has bulk-like or nanosheets character. This information and discussion are also missing.

In summary, albeit the system itself and the reported performance are of high potential interest to the field, the conclusions drawn are not sufficiently supported by the presented experimental and computational results to justify publication in Nature Communication. The authors are encouraged to complete the missing investigations and resubmit.

Response to Reviewer #1

COMMENTS TO AUTHOR:

This manuscript reported a sacrificial template-directed approach to synthesize Mo-doped NiFe (oxy)hydroxide with modulated oxygen activity as an outstanding electrocatalyst towards oxygen evolution. Density functional theory calculations and spectroscopy characterizations both confirm that the Mo dopant facilitates the lattice oxygen activation and LOM pathway for OER. However, the following questions should be seriously addressed before its publication on Nature Communications.

Response: We thank the reviewer for the valuable comments and suggestions. All the concerns raised from the reviewer have been addressed in detail as follows.

(1) The lattice oxygen oxidation is directly related to the detection of $^{18}\text{O}^{16}\text{O}$ ($m/z=34$). Therefore, the ^{18}O isotope-labeling experiments should be measured by using differential electrochemical mass spectrometry (DEMS), which provide direct evidences of lattice oxygen oxidation (Nature Chem. 2017, 9, 457; J. Am. Chem. Soc. 2021, 143, 6482; Joule 2021, 5, 2164).

Response: We thank the reviewer for the valuable comments. As suggested by the reviewer, we carried out ^{18}O isotope-labeling experiments, in-situ differential electrochemical mass spectrometry (DEMS) measurements, and ex-situ Raman spectra measurement to validate the participation of lattice oxygen in OER.

First, NiFe and MoNiFe (oxy)hydroxides were labeled with ^{18}O -isotopes by potentiostatic reaction at 1.65 V (vs. Ag/AgCl) for 30 min in KOH solution with H_2^{18}O . Afterward, the ^{18}O -labeled catalysts were rinsed with H_2^{16}O for several times to remove the remaining H_2^{18}O . Cyclic voltammograms (CV) measurements were carried on the ^{18}O -labeled samples in KOH solution with H_2^{16}O . The in-situ DEMS measurements

results on the ^{18}O -labeled NiFe and MoNiFe (oxy)hydroxide showed the signals of $m/z = 32$, $m/z = 34$, and $m/z = 36$ (**Figure A1**), suggesting the presence of $^{16}\text{O}_2$, $^{16}\text{O}^{18}\text{O}$ and $^{18}\text{O}_2$ in the gas production. These results imply that both NiFe and MoNiFe (oxy)hydroxide follow the LOM mechanism. The mass spectrometric cyclic voltammograms (MSCVs) which plot the real-time gas product contents as a function of applied potential can provide direct comparison about the participation of lattice oxygen in OER process. The ^{18}O -labeled MoNiFe (oxy)hydroxide is with noticeably higher contents of $^{18}\text{O}^{16}\text{O}$ and $^{18}\text{O}_2$ in the reaction product than the ^{18}O -labeled NiFe (oxy)hydroxide (**Figure A2**), implying that the lattice oxygen of MoNiFe (oxy)hydroxide participated more actively into the OER reaction than that of NiFe (oxy)hydroxide.

Figure A1. a-b, The differential electrochemical mass spectrometry signals of $^{16}\text{O}_2$ (I_{32}), $^{16}\text{O}^{18}\text{O}$ (I_{34}), and $^{18}\text{O}_2$ (I_{36}) as a function of collecting time on NiFe (**a**) and MoNiFe (**b**) (oxy)hydroxide.

Figure A2. **a-c**, The $^{16}\text{O}_2$ (**a**), $^{16}\text{O}^{18}\text{O}$ (**b**), and $^{18}\text{O}_2$ (**c**) signals on NiFe (oxy)hydroxide. **d**, The CV curve of NiFe (oxy)hydroxide during DEMS measurement. **e-g**, The $^{16}\text{O}_2$ (**e**), $^{16}\text{O}^{18}\text{O}$ (**f**), and $^{18}\text{O}_2$ (**g**) signals on MoNiFe (oxy)hydroxide. **h**, The corresponding CV curve of MoNiFe (oxy)hydroxide during DEMS measurement. The $^{16}\text{O}_2$ signal was normalized to 1, and the $^{16}\text{O}^{18}\text{O}$ and $^{18}\text{O}_2$ signals are relative to the $^{16}\text{O}_2$ signal. All data were taken from the first cycle.

In addition to the DEMS measurement, we carried out additional Raman spectroscopy measurement to confirm the lattice oxygen activation during OER. First, the NiFe and MoNiFe (oxy)hydroxide were completely activated in KOH solution with H_2^{16}O . Then, the (oxy)hydroxide catalysts were subjected to a constant potential of 1.65 V (vs. RHE) for 30 min in KOH solution with H_2^{18}O . Raman spectra of obtained samples with ^{18}O -labeling are shown in **Figure A3**. The Raman peaks of NiFe and MoNiFe (oxy)hydroxide with ^{18}O -labeling shifted towards lower wavenumber because of the impact oxygen mass on the vibration mode (*Angewandte Chemie International Edition*, 2020, 59(21): 8072-8077; *Angewandte Chemie*, 2019, 131(30): 10401-10405.). The MoNiFe (oxy)hydroxide showed a more obvious Raman shift to lower wavenumber than NiFe (oxy)hydroxide, suggesting a more oxygen in the lattice got exchanged between the lattice oxygen and electrolytes during OER on the MoNiFe (oxy)hydroxide.

The DEMS and Raman spectra results on the ^{18}O -labeled samples consistently

suggest that Mo doping in NiFe (oxy)hydroxide promoted the lattice oxygen to exchange with electrolyte during OER, which is in accord with the conclusion we got in our previous manuscript. The newly added ^{18}O -labeling experiments and related discussion were added to the revised manuscript.

Figure A3. a-b, Raman spectra of NiFe (a) and MoNiFe (b) (oxy)hydroxide with (red) and without (black) ^{18}O -labeling.

Page 10, line 1 in manuscript, changed: “To reveal the mechanism of the high intrinsic OER activity of MoNiFe (oxy)hydroxides, DFT calculations and electrochemical experiments were carried out on NiFe-27% and MoNiFe-27% (oxy)hydroxide.” to “To reveal the mechanism of the high intrinsic OER activity of MoNiFe (oxy)hydroxides, *isotope-labeling experiments and DFT calculations were carried out on NiFe-27% and MoNiFe-27% (oxy)hydroxide.*”

Page 10, line 5 in manuscript, added: “To validate the participation of lattice oxygen in OER for our material systems, the ^{18}O isotope-labeling experiments were carried out using the procedure described in the experimental section. In-situ differential electrochemical mass spectrometry (DEMS) measurements results on the ^{18}O -labeled NiFe and MoNiFe (oxy)hydroxide showed the signals of $m/z = 32$, $m/z = 34$, and $m/z = 36$ (Supplementary Fig. S18), suggesting the presence of $^{16}\text{O}_2$, $^{16}\text{O}^{18}\text{O}$ and $^{18}\text{O}_2$ in the gas production³¹⁻³³. This result implies that both NiFe and MoNiFe (oxy)hydroxide follow the LOM mechanism^{14, 30}. The mass spectrometric cyclic voltammograms (MSCVs) which plot the real-time gas product contents as a function of applied potential can provide direct comparison about the participation of lattice oxygen in

OER process. The ^{18}O -labeled MoNiFe (oxy)hydroxide is with noticeably higher contents of $^{16}\text{O}^{18}\text{O}$ and $^{18}\text{O}_2$ in the reaction product than the ^{18}O -labeled NiFe (oxy)hydroxide (Fig. 3a-d and Supplementary Fig. S19-S20), implying the lattice oxygen of MoNiFe (oxy)hydroxide participated more actively into the OER reaction than that of NiFe (oxy)hydroxide.

In addition to the DEMS measurement, the Raman spectra were also used to confirm the participation of lattice oxygen in OER. The Raman peaks of NiFe and MoNiFe (oxy)hydroxide with ^{18}O -labeling shifted towards lower wavenumber (Fig. 3e-f) because of the impact oxygen mass on the vibration mode^{2, 34}. The MoNiFe (oxy)hydroxide shows a more obvious shift than the NiFe (oxy)hydroxide, suggesting more lattice oxygen got exchanged with the electrolytes during OER. The Raman spectra and DEMS results on the ^{18}O -labeled samples consistently suggest that Mo doping in NiFe (oxy)hydroxide effectively promoted the lattice oxygen to participate in the OER reaction.”

Page 11, line 1 in manuscript, added:

Fig 3. Evidence of lattice oxygen participating in OER provided by ^{18}O isotope-labeling experiments. a-d, Mass spectrometric cyclic voltammograms results showing different gaseous products content of OER reaction as a function of applied potential

for the ^{18}O -labeled samples: $^{16}\text{O}^{18}\text{O}$ for NiFe (oxy)hydroxide (a) and MoNiFe (oxy)hydroxide (b), and $^{18}\text{O}_2$ for NiFe (oxy)hydroxide (c) and MoNiFe (oxy)hydroxide (d). The contents of all the species were normalized by the amount of $^{16}\text{O}_2$ in the reaction products; e-f, Raman spectra of NiFe (e) and MoNiFe (f) (oxy)hydroxide with (red) and without (black) ^{18}O -labeling.

Page 26, line 11 in manuscript, added:

“ ^{18}O -labeling experiment: NiFe and MoNiFe (oxy)hydroxides were labeled with ^{18}O -isotopes by potentiostatic reaction at 1.65 V (vs. Ag/AgCl) for 30 min in KOH solution with H_2^{18}O . Afterward, the ^{18}O -labeled catalysts were rinsed with H_2^{16}O for several times to remove the remaining H_2^{18}O .

DEMS measurements: DEMS measurements were carried out using a QAS 100 device (Linglu Instruments, Shanghai). The NiFe or MoNiFe (oxy)hydroxide with ^{18}O -labeling, a Ag/AgCl electrode prefilled with saturated KCl aqueous solution, and a Pt mesh were used as working electrode, reference electrode, and counter electrode, respectively. CV measurement was performed in KOH solution with H_2^{16}O with a scan rate of 5 mV/s. In the meantime, gas products with different molecular weights were detected in real-time by mass spectroscopy.”

Page 11 in Supplementary Information, added:

Supplementary Fig. S19. a-b, The cyclic voltammograms curves of NiFe (a) and MoNiFe (b) (oxy)hydroxide during DEMS measurement.

Page 12 in Supplementary Information, added:

Supplementary Fig. S20. a-b, The DEMS signal of $m/z=32$ ($^{16}\text{O}_2$) of NiFe (a) and MoNiFe (b) (oxy)hydroxide as a function of applied potential during OER measurement.

The **Figure A1** has been added in Supplementary Information as *Supplementary Fig. S18*.

The reference recommended by the reviewer were cited in the revised manuscript.

Page 2, line 13 in manuscript, “[6] *Grimaud, A., et al. Activating lattice oxygen redox reactions in metal oxides to catalyse oxygen evolution. Nat. Chem. 9, 457-465 (2017).*”

was cited in the revised manuscript as reference [6] in “*Grimaud⁶ and Mefford⁷ reported that lattice oxygen could participate in the oxygen evolution reaction (OER) on the (La,Sr)CoO₃ surface, which was later referred to as lattice oxygen mechanism (LOM).*”

Page 10, line 11 in manuscript, “[32] *Wen, Y., et al. Stabilizing highly active Ru sites by suppressing lattice oxygen participation in acidic water oxidation. J. Am. Chem. Soc. 143, 6482-6490 (2021).* [33] *Shi, Z., et al. Confined Ir single sites with triggered lattice oxygen redox: Toward boosted and sustained water oxidation catalysis. Joule 5, 2164-2176 (2021).*” was cited in the revised manuscript as reference [32] and [33] in “*suggesting the presence of $^{16}\text{O}_2$, $^{16}\text{O}^{18}\text{O}$, and $^{18}\text{O}_2$ in the gas production³¹⁻³³.*”

(2) The OER activity comparison in D₂O and H₂O solution is an effective way to prove

the impact of proton activity. The isotopic experiments are helpful for determining whether the proton/hydrogen transfer is involved in the rate-determining step (RDS). The RDS of MoNiFe is the deprotonation of *OOH (Figure 3b), which cannot be substantiated by the different performance of MoNiFe in D₂O and H₂O solution (Figure 3e). To scientifically determine the RDS, the kinetic isotope effect (KIE) of MoNiFe and NiFe should be calculated (Angew. Chem. Int. Ed. 2021, 60, 3095). In other hand, it may be more scientific that isotopic experiments are performed in dissolved KOD with D₂O rather than KOH with D₂O.

Response: We thank the reviewer for the valuable suggestion. As suggested by the reviewer, we redid the isotopic experiment in dissolved NaOD with D₂O, and calculated the kinetic isotope effect (KIE) of MoNiFe and NiFe (oxy)hydroxide.

The linear sweep voltammetry (LSV) curves for NiFe and MoNiFe (oxy)hydroxide measured in 1 M NaOH (dissolved in H₂O) and NaOD (dissolved in D₂O) solution were shown in **Figure A4a and b**. To show the kinetic isotope effect (KIE) for NiFe and MoNiFe (oxy)hydroxide clearly, the ratio of current density obtained in NaOH and in NaOD at the given potential is plotted in **Figure A4c**. MoNiFe (oxy)hydroxide exhibited a noticeably larger KIE value in comparison to NiFe (oxy)hydroxide, suggesting a severe degradation of OER activity in NaOD. This result suggested that proton transfer had a greater impact on the OER process on MoNiFe (oxy)hydroxide than that on NiFe (oxy)hydroxide. The deuterium isotopic experiments performed in NaOH/NaOD with a different concentration of 0.5 M provided consistent results (**Figure A5**). The large isotopic effect of MoNiFe (oxy)hydroxide suggests that the proton transfer is involved in the potential determining step (PDS). This conclusion is in accord with the DFT calculation results, which show that the PDS step of OER on MoNiFe (oxy)hydroxide is the deprotonation of *OOH.

Figure A4. a-b, LSV curves for NiFe (a) and MoNiFe (b) (oxy)hydroxide measured in 1 M NaOH and 1 M NaOD solution. The LSV curves are without iR compensation. c, The kinetic isotope effect of NiFe and MoNiFe (oxy)hydroxide in 1 M NaOH/NaOD.

Figure A5. a-b, LSV curves for NiFe (a) and MoNiFe (b) (oxy)hydroxide measured in 0.5 M NaOH and 0.5 M NaOD solution. The LSV curves are without iR compensation. c, The kinetic isotope effect of MoNiFe and NiFe (oxy)hydroxide in 0.5 M NaOH/NaOD.

Page 14, line 5 in manuscript, changed from: “Therefore, to confirm such a shift of PDS by Mo doping as revealed by calculations, we evaluated the dependence of OER activity of NiFe and MoNiFe(oxy)hydroxide on proton activity by carrying out pH dependence measurements and isotopic labeling experiments.”

To “Therefore, to confirm such a shift of PDS by Mo doping as revealed by calculations, we evaluated the dependence of OER activity of NiFe and MoNiFe(oxy)hydroxide on proton activity by carrying out pH dependence measurements and *deuterium* isotopic labeling experiments.”

Page 14, line 13 in manuscript, change from: “To further prove the impact of proton

activity, ... which is consistent with the DFT calculations that the PDS shifted to a deprotonation step after Mo doping.”

To “To further prove the impact of proton activity, the OER activity of NiFe and MoNiFe (oxy)hydroxide were also evaluated in the NaOD and NaOH solution. The LSV curves for NiFe and MoNiFe (oxy)hydroxide measured in 1 M NaOH (dissolved in H₂O) and NaOD (dissolved in D₂O) solution were shown in **Fig. 4e**. To show the kinetic isotope effect (KIE) for NiFe and MoNiFe (oxy)hydroxide clearly, the ratio of current density obtained in NaOH and in NaOD at the given potential^{39, 40} is plotted in **Fig. 4f**. MoNiFe (oxy)hydroxide exhibited a noticeably larger KIE value in comparison to NiFe (oxy)hydroxide, suggesting a severe degradation of OER activity in NaOD. This result suggested that proton transfer had a greater impact on the OER process on MoNiFe (oxy)hydroxide than that on NiFe (oxy)hydroxide. The deuterium isotopic experiments performed in NaOH/NaOD with a different concentration of 0.5 M provided consistent results (**Supplementary Fig. S23**). The large isotopic effect of MoNiFe (oxy)hydroxide suggests that the proton transfer is involved in the potential determining step (PDS). This conclusion is in accord with the DFT calculation results, which show that the PDS step of OER on MoNiFe (oxy)hydroxide is the deprotonation of *OOH (**Fig. 4b**).”

The **Figure A5** has been added into Supplementary Information as **Supplementary Fig. S23**.

Correspondingly, the Fig. 3e,f in manuscript was changed to:

Fig 4. e, LSV curves for NiFe and MoNiFe (oxy)hydroxide measured in 1 M NaOH and 1 M NaOD solution. The LSV curves are without iR compensation. f, The kinetic isotope effect of MoNiFe and NiFe (oxy)hydroxide.

The reference recommended by the reviewer and the related reference were added to the revised manuscript.

Page 14, line 18 in manuscript, “[39] Bai, L., Lee, S., Hu, X. *Spectroscopic and electrokinetic evidence for a bifunctional mechanism of the oxygen evolution reaction. Angew. Chem. Int. Ed. Engl.* 60, 3095-3103 (2021). [40] Tse, E. C. M., Hoang, T. T. H., Varnell, J. A., Gewirth, A. A. *Observation of an inverse kinetic isotope effect in oxygen evolution electrochemistry. ACS Catal.* 6, 5706-5714 (2016).” were cited in the revised manuscript as reference [39] and [40] in “To show the kinetic isotope effect (KIE) for NiFe and MoNiFe (oxy)hydroxide clearly, the ratio of current density obtained in NaOH and in NaOD at the given potential^{39, 40} is plotted in **Fig. 4f**.”

(3) The AEM pathway of NiFe and MoNiFe oxyhydroxide should also be calculated to fully prove the favorable kinetics of LOM pathway (Nat. Commun. 2020, 11, 4066). It should be noted that the in-situ Raman spectra have confirmed that the lattice oxygen of NiFe LDH wasn't involved in OER (Angew. Chem. Int. Ed. 2020, 59, 8072). In other words, the AEM pathway of NiFe may be more thermodynamically favorable than

LOM of NiFe. Therefore, it is improper to only evaluate the energy barriers by using LOM process. Besides, the optimized DFT models of OER process should be provided for confirming the rationality of schematic illustration of the proposed OER pathway (Figure 3a).

Response: We thank the reviewer for the valuable suggestion. As suggested by the reviewer, we added the optimized DFT models and the calculation details of OER process following AEM mechanism in the revised manuscript.

1) Optimized DFT model of OER using for both AEM and LOM process

The slab model of NiFe (oxy)hydroxide used for both AEM and LOM reaction pathway is terminated by the (001) surface (**Figure A6**). For the AEM pathway, the metal site should be exposed to the reactants. Thus, two vacuum spaces were inserted along (001) and (010). To eliminate the interaction between periodic slabs, the thickness of vacuum spaces in both models was more than 10 Å. In addition, part of hydrogen atoms was removed because of the oxidation atmosphere.

Figure A6. The slab model of NiFe (oxy)hydroxide.

To find the stable configuration of Mo doping, we have built three slab models with different Mo sites (**Figure A7**). The relative stability of Mo replacement was determined by calculating the formation energy (ΔE_f), which was computed as:

$$\Delta E_f = E_{\text{slab}} - \sum N_i E_i,$$

where E_{slab} , E_i and N_i are total energies of the slab model, energy, and number of the i-th element, respectively. The calculated results show that the Mo atom favors

replacing the Ni site and exposes to the vacuum as this model with the most negative ΔE_f (Figure A7d).

Figure A7. Doping (Ni,Fe)(OH)₂ with Mo at different sites. a, Mo atom replaces the surface Ni and exposes to the vacuum. **b**, Mo atom replaces surface Ni atom without the exposure to vacuum. **c**, Mo atom replaces the surface Fe atom. **d**, The comparison of the formation energy (ΔE_f) of configuration a-c.

The energy barriers of OER on different surfaces were calculated based on the computational hydrogen electrode (CHE) model (*The Journal of Physical Chemistry B*, 2004, 108(46): 17886-17892.), where the electrode kinetics was determined by the adsorption Gibbs free energies from DFT calculations. The Gibbs free energies of gaseous H₂ and liquid H₂O were corrected at 298.15 K, 1.0 bar and 298.15 K, 0.035 bar from vibrational frequency calculations. The Gibbs free energy of gaseous O₂ was obtained from equation as

$$G_{O_2} = 2G_{H_2O} - 2G_{H_2} + 4.92 \text{ eV}$$

The results in **Table A1** show good agreement with references (*The Journal of Physical Chemistry B*, 2004, 108(46): 17886-17892.; *Journal of the American chemical Society*, 2013, 135(36): 13521-13530.).

Table A1 Thermal corrections to Gibbs free energies of different adsorbates.

Adsorbate	G (eV) ^a	$\Delta G_0 - 298.15 \text{ K}^a$	ZPE (eV) ^a	TS (eV) ^a	ZPE (eV) ^b	TS (eV) ^b
H ₂ (g)	-6.81	-0.05	0.27	0.40	0.27 ^b	0.41 ^b
O ₂ (g)	-9.98 ^c	-	-	-	0.10 ^b	0.64 ^b
H ₂ O (l)	-14.25	-0.03 ^c	0.57	0.70	0.56 ^b	0.67 ^b
*O _l O ^d	-	0.05 ~ 0.06	0.14	0.15 ~ 0.17	0.17 ^b	0.00 ^b
*OH	-	0.27 ~ 0.36	0.33 ~ 0.38	0.12 ~ 0.16	0.30 ^b	0.00 ^b
*O	-	0.01 ~ 0.06	0.05 ~ 0.07	0.03 ~ 0.09	0.07 ^b	0.00 ^b
*OOH	-	0.30 ~ 0.40	0.42 ~ 0.46	0.12 ~ 0.23	0.44 ^e	0.00 ^e

a: This work.

b: Data cited from reference (*The Journal of Physical Chemistry B*, 2004, 108(46): 17886-17892.).

c: Calculated from equation: $G_{O_2} = 2G_{H_2O} - 2G_{H_2} + 4.92 \text{ eV}$.

d: O_l denotes the lattice oxygen atom in the LOM pathway.

e: Data cited from reference (*Journal of the American chemical Society*, 2013, 135(36): 13521-13530.).

For the AEM pathway in an alkaline electrolyte, the four-electron reactions are:

where “*” represents the adsorption sites, which are generally the exposed metal sites.

The configurations of AEM pathway are shown in **Figure A8**. The free energy changes of each step can be calculated as:

$$\Delta G_1 = G(*OH) + 0.5 G(H_2) - G(*) - G(H_2O) - eU, \quad (5)$$

$$\Delta G_2 = G(*O) + 0.5 G(H_2) - G(*OH) - eU, \quad (6)$$

$$\Delta G_3 = G(*OOH) + 0.5 G(H_2) - G(*O) - G(H_2O) - eU, \quad (7)$$

$$\Delta G_4 = G(*) + G(O_2) + 0.5 G(H_2) - G(*OOH) - eU, \quad (8)$$

where U is the potential with respect to the normal hydrogen electrode (NHE).

The calculated overpotential (η) was then determined by:

$$\eta = \text{Max} \{ \Delta G_1, \Delta G_2, \Delta G_3, \Delta G_4 \} \quad (9)$$

Figure A8. The configuration of NiFe (a) and MoNiFe (b) (oxy)hydroxide in AEM mechanism.

The LOM pathway includes five steps, which are:

where “*” represents the vacancy sites. O_l denotes the lattice oxygen atoms.

The configurations of LOM pathway are shown in **Figure A9**. The energy barriers of LOM pathway were calculated by:

$$\Delta G_1 = G(*O_l) + 0.5 G(H_2) - G(*O_lH) - eU, \quad (15)$$

$$\Delta G_2 = G(*O_lOH) + 0.5 G(H_2) - G(H_2O) - G(*O_l) - eU, \quad (16)$$

$$\Delta G_3 = G(*O_lO) + 0.5 G(H_2) - G(*O_lOH) - eU, \quad (17)$$

$$\Delta G_4 = G(*) + G(O_2) - G(*O/O), \quad (18)$$

$$\Delta G_5 = G(*O/H) + 0.5 G(H_2) - G(H_2O) - G(*), \quad (19)$$

The overpotential of LOM is calculated by:

$$\eta = \text{Max} \{ \Delta G_1, \Delta G_2, \Delta G_3, \Delta G_4, \Delta G_5 \} \quad (20)$$

Figure A9. The configuration of NiFe (a) and MoNiFe (b) (oxy)hydroxide in LOM mechanism.

2) Determining reaction energy barriers in AEM pathway

In the AEM pathway, the OER reaction involves four subsequent proton-electron transfer steps, including *OH species adsorption, *O radical formation, *OOH transformation, and O₂ desorption (**Figure A10a**). To identify the active site in AEM pathway, both Ni site and Fe site have been considered (**Figure A10b-c**). The Fe sites were found to be the active sites with lower barrier than that on Ni sites. As shown in **Figure A10c**, the deprotonation of *OH in AEM pathway serves as potential determining step (PDS) for both NiFe and MoNiFe (oxy)hydroxide, with a barrier of 1.05 eV and 0.76 eV, respectively.

Figure A10. a, Schematic illustration of the AEM pathway. b-c, The Gibbs free energy diagrams of OER in AEM pathway on Ni site (b) and Fe site (c) in NiFe and MoNiFe (oxy)hydroxide.

(3) Comparison of AEM and LOM reaction pathway

In the LOM pathway, the (oxy)hydroxides first go through the deprotonation process to form oxyhydroxide (step 1) (**Figure A11a**). The exposed lattice oxygen then receives OH^- via nucleophilic attack to form $^*\text{OOH}$ (step 2). After the deprotonation of $^*\text{OOH}$ (step 3), gaseous O_2 releases from the lattice, and an oxygen vacancy is generated on the surface (step 4). The resulting oxygen vacancy sites are refilled by OH^- and the surface is recovered (step 5). The calculated Gibbs energy diagrams of OER on NiFe and MoNiFe (oxy)hydroxide are displayed in **Figure A11b**. For the NiFe (oxy)hydroxide, the desorption of O_2 , which was accompanied by the formation of oxygen vacancy, was found to be the PDS with a high energy barrier of 0.75 eV. In contrast, the barrier of oxygen vacancy formation became much smaller after Mo doping, which pushed the PDS on MoNiFe (oxy)hydroxide to the deprotonation of $^*\text{OOH}$ with a decreased energy barrier of 0.42 eV. It is noting that both the barriers of PDS of NiFe and MoNiFe (oxy)hydroxide in LOM pathway were much lower than that in AEM pathway, suggesting that both the NiFe and MoNiFe (oxy)hydroxide follow the LOM mechanism, which is consistent with the results of the ^{18}O isotope-labeling experiments.

Figure A11. a, Schematic illustration of the LOM pathway. b, The Gibbs free energy diagrams of OER in LOM pathway on NiFe and MoNiFe (oxy)hydroxide.

In the previous work, Lee et al. (*Angewandte Chemie International Edition*, 2020, 59(21): 8072-8077.) observed the presence of lattice oxygen exchange with electrolyte during OER when the Fe content in Ni (oxy)hydroxide was lower than 4.7% using oxygen isotope labeling and operando Raman spectroscopic experiments. However, since the active sites change from Ni sites to the Fe sites when the Fe content exceeds 4.7%, it is difficult to confirm whether the lattice oxygen participates in OER basing the shift of Raman peak of $\text{Ni}^{3+}\text{-O}$. In addition, de Araújo J. F. et al. (*Angewandte Chemie International Edition*, 2021, 60(27): 14981-14988.) used DEMS to prove the occurrence of Mars-van-Krevelen lattice oxygen evolution reaction mechanism in NiFe (oxy)hydroxide, which involves the coupling of oxygen atoms from the catalyst and the electrolyte. In our work, combining the results of ^{18}O isotope-labeling experiments and DFT calculation, we believe that the NiFe (oxy)hydroxide follows the LOM mechanism, and the Mo doping can effectively activate the oxygen activity in NiFe (oxy)hydroxide, facilitate the lattice oxygen exchange with electrolyte, and thus improve the OER performance.

Page 11, line 12 in manuscript, changed from: “As mentioned above, the OER on NiFe-based (oxy)hydroxides was reported to follow the LOM, in which lattice oxygen directly participates in the OER reactions¹⁴. ...which pushed the PDS on MoNiFe

(oxy)hydroxide to the deprotonation of *OOH with a decreased energy barrier of 0.44 eV.”

To “In additional to ^{18}O isotope-labeling experiments, DFT calculations were also carried out to identify the OER mechanism on NiFe and MoNiFe (oxy)hydroxide. Both adsorbate evolution mechanism (AEM) pathway (**Supplementary Fig. S21a**) and LOM pathway (**Fig. 4a**) of OER were considered. In the AEM pathway, the Fe sites were found to be the active sites with lower barriers than Ni sites (**Supplementary Fig. S21b-c**). The deprotonation of *OH in AEM pathway serves as the potential determining step (PDS) for both NiFe and MoNiFe (oxy)hydroxide, with a barrier of 1.05 eV and 0.76 eV, respectively. In the LOM pathway, the (oxy)hydroxides first go through the deprotonation process to form oxyhydroxide (step 1) (**Fig. 4a**). The exposed lattice oxygen then receives OH⁻ via nucleophilic attack to form *OOH (step 2). After the deprotonation of *OOH (step 3), gaseous O₂ releases from the lattice, and an oxygen vacancy is generated on the surface (step 4). The resulting oxygen vacancy sites are refilled by OH⁻ and the surface is recovered (step 5). The calculated Gibbs energy diagrams of OER on NiFe and MoNiFe (oxy)hydroxide are displayed in **Fig. 4b**. For the NiFe (oxy)hydroxide, the desorption of O₂, which was accompanied by the formation of oxygen vacancy, was found to be the PDS with a high energy barrier of 0.75 eV. In contrast, the barrier of oxygen vacancy formation became much smaller after Mo doping, which pushed the PDS on MoNiFe (oxy)hydroxide to the deprotonation of *OOH with a decreased energy barrier of 0.42 eV. It is noted that both the barriers for PDS of NiFe and MoNiFe (oxy)hydroxide in LOM pathway were much lower than that in AEM pathway, suggesting that both the NiFe and MoNiFe (oxy)hydroxide follow the LOM mechanism^{14, 30}. This result consistent with the results of the ^{18}O isotope-labeling experiments.”

The **Figure A6** has been added in Supplementary Information as **Supplementary Fig. S36**.

The **Figure A7** has been added in Supplementary Information as **Supplementary Fig. S37**.

The **Figure A8** has been added in Supplementary Information as *Supplementary Fig. S38*.

The **Figure A9** has been added in Supplementary Information as *Supplementary Fig. S39*.

The **Figure A10** has been added in Supplementary Information as *Supplementary Fig. S21*.

The **Table A1** has been added in Supplementary Information as *Table S2*.

Page 33, line 14 in Supplementary Information, added:

“1) Optimized DFT model of OER using for both AEM and LOM process:

The slab models of NiFe (oxy)hydroxide used in AEM and LOM pathway were terminated by the (001) surface (Supplementary Fig. S36). For the AEM pathway, the metal site should be exposed to the reactants. Thus, two vacuum spaces were inserted along (001) and (010). To eliminate the interaction between periodic slabs, the thickness of vacuum spaces in both models was more than 10 Å. In addition, part of hydrogen atoms was removed because of the oxidation atmosphere.

To find the stable configuration of Mo doping, we have built three slab models with different Mo sites (Supplementary Fig. S37). The relative stability of Mo replacement was determined by calculating the formation energy (ΔE_f), which was computed as:

$$\Delta E_f = E_{slab} - \sum N_i E_i,$$

where E_{slab} , E_i and N_i are total energies of the slab model, energy, and number of the i -th element, respectively. The calculated results show that the Mo atom favors replacing the Ni site and exposes to the vacuum as this model with the most negative ΔE_f (Supplementary Fig. S37d).”

Page 35, line 10 in Supplementary Information, added:

“4) Energy barriers of OER calculation:

The energy barriers of OER on different surfaces were calculated based on the computational hydrogen electrode (CHE) model²⁷, where the electrode kinetics was determined by the adsorption Gibbs free energies from DFT calculations. The Gibbs

free energies of gaseous H_2 and liquid H_2O were corrected at 298.15 K, 1.0 bar and 298.15 K, 0.035 bar from vibrational frequency calculations. The Gibbs free energy of gaseous O_2 was obtained from equation (5). The results in **Table S2** show good agreement with references^{27, 28}.

For the AEM pathway in an alkaline electrolyte, the four-electron reactions are:

where “*” represents the adsorption sites, which are generally the exposed metal sites.

The configurations of AEM pathway are shown in **Supplementary Fig. S38**. The free energy changes of each step can be calculated as:

$$\Delta G_1 = G(*OH) + 0.5 G(H_2) - G(*) - G(H_2O) - eU, \quad (10)$$

$$\Delta G_2 = G(*O) + 0.5 G(H_2) - G(*OH) - eU, \quad (11)$$

$$\Delta G_3 = G(*OOH) + 0.5 G(H_2) - G(*O) - G(H_2O) - eU, \quad (12)$$

$$\Delta G_4 = G(*) + G(O_2) + 0.5 G(H_2) - G(*OOH) - eU, \quad (13)$$

where U is the potential with respect to the normal hydrogen electrode (NHE).

The calculated overpotential (η) was then determined by:

$$\eta = \text{Max}\{\Delta G_1, \Delta G_2, \Delta G_3, \Delta G_4\} \quad (14)$$

The LOM pathway includes five steps, which are:

where “*” represents the vacancy sites. O_l denote the lattice oxygen atoms.

The configurations of LOM pathway are shown in **Supplementary Fig. S39**. The energy barriers of LOM pathway were calculated by:

$$\Delta G_1 = G(*O_l) + 0.5 G(H_2) - G(*O_lH) - eU, \quad (18)$$

$$\Delta G_2 = G(*O_1OH) + 0.5 G(H_2) - G(H_2O) - G(*O_1) - eU, \quad (19)$$

$$\Delta G_3 = G(*O_1O) + 0.5 G(H_2) - G(*O_1OH) - eU, \quad (20)$$

$$\Delta G_4 = G(*) + G(O_2) - G(*O_1O), \quad (21)$$

$$\Delta G_5 = G(*O_1H) + 0.5 G(H_2) - G(H_2O) - G(*), \quad (22)$$

The overpotential of LOM is calculated by:

$$\eta = \text{Max}\{\Delta G_1, \Delta G_2, \Delta G_3, \Delta G_4, \Delta G_5\} \quad (23)''$$

The reference recommended by the reviewer and the related reference were added to the revised manuscript.

Page 12, line 14 in manuscript, “[14] Zhang, N., et al. Lattice oxygen activation enabled by high-valence metal sites for enhanced water oxidation. *Nat. Commun.* 11, 4066 (2020). [30] Ferreira de Araujo, J., Dionigi, F., Merzdorf, T., Oh, H. S., Strasser, P. Evidence of Mars-Van-Krevelen mechanism in the electrochemical oxygen evolution on Ni-based catalysts. *Angew. Chem. Int. Ed. Engl.* 60, 14981-14988 (2021).” were cited in the revised manuscript as reference [14] and [30] in “suggesting that both the NiFe and MoNiFe (oxy)hydroxide follow the LOM mechanism^{14, 30}”

Page 10, line 23 in manuscript, “[34] Lee, S., Banjac, K., Lingenfelder, M., Hu, X. Oxygen isotope labeling experiments reveal different reaction sites for the oxygen evolution reaction on nickel and nickel iron oxides. *Angew. Chem. Int. Ed. Engl.* 58, 10295-10299 (2019). [2] Lee, S., Bai, L., Hu, X. Deciphering iron-dependent activity in oxygen evolution catalyzed by nickel-iron layered double hydroxide. *Angew. Chem. Int. Ed. Engl.* 59, 8072-8077 (2020).” were cited in the revised manuscript as reference [34] and [2] in “The Raman peaks of NiFe and MoNiFe (oxy)hydroxide with ¹⁸O-labeling shifted towards lower wavenumber (**Fig. 3e-f**) because of the impact oxygen mass on the vibration mode^{2, 34}.”

(4) The density of states calculations revealed that there was a upshift of O 2p band center after Mo doping. Meanwhile, the enlarged energy distance between the LHB and UHB band center leads to the downshift of LHB. Therefore, the Mo incorporation strengthened the metal-oxygen bonds and enhanced the metal-oxygen covalency.

However, the COHP calculations indicated the weaker metal-oxygen bond of MoNiFe (oxy)hydroxide comparing to that of NiFe (oxy)hydroxide. The authors should explain the contradictive conclusions for DOS and COHP calculations.

Response: We thank the reviewer for the insightful question. The LHB/UHB centers were determined by the total metal $3d$ -orbital distribution below/above the Fermi level (E_F) in DOS diagrams. The specific positions of LHB were calculated to be -4.36 eV and -4.67 eV for NiFe (oxy)hydroxide and MoNiFe (oxy)hydroxide, respectively. The O $2p$ band center of NiFe and MoNiFe (oxy)hydroxide were calculated to be -1.58 eV and -1.40 eV, respectively. It is noting that the LHB center located beneath the O $2p$ band center. Therefore, the downshift of LHB center and upshift of O $2p$ band center for MoNiFe (oxy)hydroxide leads to smaller overlap of metal $3d$ -orbital and oxygen $2p$ -orbital, which results in the weaker metal-oxygen bond (**Figure A12**). In addition, it is noting that despite the density of states of metal $3d$ -orbital, especially for Ni $3d$ -orbital, seems to upshift closed to Fermi level, leading to an increase in the anti-bonding states below the Fermi level (**Figure A13**). Such an effect weakens the metal-oxygen bonds, which is consistent with the COHP calculations.

Figure A12. Schematic band diagrams of NiFe (left) and MoNiFe (right) (oxy)hydroxide. The d-orbitals split into electron-filled lower Hubbard band (LHB) and empty upper Hubbard band (UHB) with an energy difference of U .

Figure A13. Projected density of states of NiFe and MoNiFe (oxy)hydroxide.

Page 16, line 6 in manuscript, added: “*The LHB/UHB center was determined by the total metal 3d-orbital distribution below/above E_F in DOS diagrams. The specific positions of LHB and UHB were calculated to be -4.36 eV and 2.01 eV for NiFe (oxy)hydroxide, and -4.67 eV and 2.90 eV for MoNiFe (oxy)hydroxide, respectively.*”

Page 16, line 14 in manuscript, changed from: “*Such an enlarged U value gives rise to the downshift of LHB, leading to a stronger interaction with the O 2p band. As a result, as anodic potential is applied, the electron removal from oxygen sites was strongly facilitated. (Fig. 4d)^{11, 14}.*”

To “*Such an enlarged U value gives rise to the downshift of LHB (Fig. 5d). As a result, as anodic potential is applied, the electron removal from oxygen sites was strongly facilitated^{11, 14}. It is noting that the LHB center located beneath the O 2p band center. Therefore, the downshift of LHB center and upshift of O 2p band center for MoNiFe (oxy)hydroxide leads to smaller overlap of metal 3d-orbital and oxygen 2p-orbital, which results in the weaker metal-oxygen bond. In addition, the density of states of metal 3d-orbital, especially for Ni 3d-orbital, upshift closed to Fermi level, leading to an increase in the anti-bonding states below the Fermi level (Fig. 5c). Such effect weakens the metal-oxygen bonds, which is consistent with the COHP calculations (Fig. 5b).*”

The Fig. 5 in manuscript was changed to:

Fig 5. Lattice oxygen activity determined by density functional theory (DFT) calculations. **a**, Crystal orbital Hamilton populations (COHP) of the Ni-O bond in NiFe and MoNiFe (oxy)hydroxide. **b**, The integrated -COHP up to Fermi level comparison of Ni-O and Fe-O in NiFe and MoNiFe (oxy)hydroxide. *TM is referred to transition metal.* **c**, Projected density of states of NiFe and MoNiFe (oxy)hydroxide. *The anti-bonding states below the Fermi level were highlighted by dash circles.* **d**, Schematic band diagrams of NiFe and MoNiFe (oxy)hydroxide. The d-orbitals split into electron-filled lower Hubbard band (LHB) and empty upper Hubbard band (UHB) with an energy difference of U . **e**, The oxygen vacancy formation energy (E_{f_vac}) of NiFe and MoNiFe (oxy)hydroxide.

(5) The U values of NiFe and MoNiFe (oxy)hydroxide were calculated to be 6.38 eV and 7.58 eV, respectively. To validate the schematic band diagrams of NiFe and

MoNiFe (oxy)hydroxide, the specific positions of LHB and UHB calculation and the details of U values calculations should be provided in this manuscript.

Response: We thank the reviewer for the valuable suggestion. The LHB was determined by the 3d-orbital distribution below E_F in DOS diagrams, while the UHB was determined by the unoccupied 3d-orbitals distribution above E_F . The centers of LHB and UHB were calculated by:

$$\bar{\epsilon}_{LHB} = \frac{\int_{-\infty}^0 n(\epsilon)\epsilon d\epsilon}{\int_{-\infty}^0 n(\epsilon)d\epsilon},$$

and

$$\bar{\epsilon}_{UHB} = \frac{\int_0^{+\infty} n(\epsilon)\epsilon d\epsilon}{\int_0^{+\infty} n(\epsilon)d\epsilon},$$

where ϵ and $n(\epsilon)$ are the energy level and number of states at this energy level, respectively.

As shown in **Figure A14**, the specific positions of LHB and UHB were calculated to be -4.36 eV and 2.01 eV for NiFe (oxy)hydroxide, and -4.67 eV and 2.90 eV for MoNiFe (oxy)hydroxide, respectively.

Figure A14. Projected density of states of NiFe and MoNiFe (oxy)hydroxide. The purple dash lines represent the specific positions of UHB or LHB center.

page 16, line 6 in manuscript, added: “*The LHB/UHB center was determined by the total metal 3d-orbital distribution below/above E_F in DOS diagrams. The specific positions of LHB and UHB was calculated to be -4.36 eV and 2.01 eV for NiFe (oxy)hydroxide, and -4.67 eV and 2.90 eV for MoNiFe (oxy)hydroxide, respectively.*”

Page 34, line 9 in Supplementary Information, added:

“(2) *Determining the LHB and UHB band center:*

The LHB was determined by the 3d-orbital distribution below E_F in DOS diagrams, while the UHB was determined by the unoccupied 3d-orbitals distribution above E_F .

The center of LHB and UHB were calculated by:

$$\bar{\epsilon}_{LHB} = \frac{\int_{-\infty}^0 n(\epsilon)\epsilon d\epsilon}{\int_{-\infty}^0 n(\epsilon)d\epsilon}, \quad (2)$$

and

$$\bar{\epsilon}_{UHB} = \frac{\int_0^{+\infty} n(\epsilon)\epsilon d\epsilon}{\int_0^{+\infty} n(\epsilon)d\epsilon}, \quad (3)$$

where ϵ and $n(\epsilon)$ are the energy level and number of states at this energy level, respectively.”

(6) The crystal structures of catalysts have been characterized by HRTEM and SAED. However, no X-ray diffraction (XRD) patterns of catalysts were provided in this manuscript. The authors should do the XRD measurement for the further structure characterizations.

Response: We thank the reviewer for the valuable suggestion. We indeed carried out X-ray diffraction (XRD) measurement on the MoNiFe (oxy)hydroxide samples which were loaded on carbon cloths, as shown in **Figure A15**. However, because of the low loading mass, we can only see the peaks of the carbon cloths substrate, and no noticeable signals of (oxy)hydroxide can be observed. Therefore, we relied on the TEM measurement to determine the crystal structure of our catalysts. To further confirm the form of MoNiFe (oxy)hydroxide, we added extra aberration-corrected high-angle annular dark-field scanning transmission electron microscope (HAADF-STEM) measurement and the results are shown in **Figure A16**. The bright points in the HAADF-STEM image represent the Mo atoms in MoNiFe (oxy)hydroxide due to the higher atomic mass of Mo than Ni and Fe atoms, which confirms the presence of Mo

doping in MoNiFe (oxy)hydroxide. However, the specific doping site cannot be observed due to the low crystallinity of MoNiFe (oxy)hydroxide and its susceptibility to be damaged under electron beam irradiation.

Figure A15. The X-ray diffraction pattern of bare carbon cloths and MoNiFe (oxy)hydroxide.

Figure A16. The HAADF-STEM image of MoNiFe (oxy)hydroxide.

Page 6, line 11 in manuscript, added: “*The presence of Mo dopant in NiFe (oxy)hydroxide was further confirmed by the aberration-corrected high-angle annular*

dark-field scanning transmission electron microscope (HAADF-STEM) (Supplementary Fig. S4).”

Page 6, line 14 in manuscript, added: “*Because of the low loading mass of MoNiFe (oxy)hydroxide, we could not determine the crystal structure of MoNiFe-27% (oxy)hydroxide using X-ray diffraction measurement (Supplementary Fig. S5). We relied on the transmission electron microscopy (TEM) measurement to confirm the formation of (oxy)hydroxide phase.*”

The **Figure A15** has been added in Supplementary Information as *Supplementary Fig. S5*.

The **Figure A16** has been added in Supplementary Information as *Supplementary Fig. S4*.

(7) With regard to XPS data, only Ni 2p spectra of catalysts were analyzed. The Fe 2p, Mo 3d and O 1s spectra should also be measured for the elucidation of chemical states.

Response: We thank the reviewer for the valuable suggestion. As suggested by the reviewer, we added the Fe 2p, Mo 3d, and O 1s XPS spectra and corresponding discussion to the revised manuscript.

As shown in **Figure A17**, the Fe 2p XPS spectra consist of two peaks located at ~710.7 eV and ~723.7 eV, which can be attributed to the spin-orbital splitting of Fe³⁺ (*Energy & Environmental Science*, 2020, 13(1): 86-95.). After Mo doping, the Fe 2p spectra shift to a higher energy level, suggesting the higher valence state of Fe in MoNiFe (oxy)hydroxide, which is consistent with the Fe L-edge XAS results (**Fig. 6c** in manuscript).

Figure A17. The Fe 2p XPS spectra of NiFe and MoNiFe (oxy)hydroxide.

Because of the low content of Mo element, we cannot detect a noticeable Mo 3d signal in the XPS measurement. Therefore, we can only determine the absolute content of Mo in the sample from ICP-OES measurement ($0.11 \mu\text{g}/\text{cm}^2$) (**Figure A18**), and cannot directly determine its valence states.

Figure A18. The content of Ni, Fe, Mo cation in MoNiFe-27% (oxy)hydroxide.

As shown in the **Figure A19a-b**, the O 1s XPS spectra can be deconvoluted into three characteristic species, including the oxygen-metal bond in the lattice (lattice O) at ~ 530.1 eV, the unsaturated oxygen with low coordination (defective O) at ~ 531.5 eV, and the adsorbed water molecules on the surface (adsorbed H_2O) at ~ 532.5 eV (*Applied Catalysis B: Environmental*, 2021, 284: 119740; *ACS Energy Letters*, 2018, 3(7): 1515-1520). To quantify the defective O in (oxy)hydroxide catalyst, the area ratio of

defective O to the total area and to the lattice O were calculated. As shown in **Figure A19c**, the MoNiFe (oxy)hydroxide shows higher defective O content than NiFe (oxy)hydroxide, suggesting that the MoNiFe (oxy)hydroxide might have more unsaturated oxygen sites, which is consistent with the higher oxygen activity as revealed by DFT calculations.

Figure A19. a-b, The O 1s XPS spectrum of NiFe (a) and MoNiFe (b) (oxy)hydroxide. **c**, Comparison of fitting results for (defective O)/total and (defective O)/(lattice O) ratio.

The **Figure A17** has been added in Supplementary Information as Supplementary Fig. S29.

The **Figure A19** has been added in Supplementary Information as Supplementary Fig. S25.

Page 29, line 3 in Supplementary Information, added: “As shown in Supplementary Fig. S29, the Fe 2p XPS spectra consist two peaks at ~710.7 eV and ~723.7 eV, which can be attributed to spin-orbital splitting of Fe³⁺.³³ After Mo doping, the Fe 2p spectra shift slightly to higher energy level, suggesting an increased in the Fe valence state. Such impact of Fe valence state by Mo doping is consistent with the Fe L-edge XAS results (Fig. 6c).”

Page 17, line 1 in manuscript, added: “This result was also confirmed by the higher content of defective oxygen in MoNiFe (oxy)hydroxide by O 1s XPS analysis (Supplementary Fig. S25, note 3).”

Page 27, line 5 in Supplementary Information, added:

“**Supplementary note 3**

As shown in the **Supplementary Fig. S25**, the O 1s XPS spectra can be deconvoluted into three characteristic species, including the oxygen-metal bond in the lattice (lattice O) at ~530.1 eV, the unsaturated oxygen with low coordination (defective O) at ~531.5 eV, and adsorbed water molecules on surface (adsorbed H₂O) at ~532.5 eV^{29,30}. To quantify the defective O in (oxy)hydroxide catalyst, the area ratio of defective O to the total area of O 1s spectra and to the lattice O were calculated. As shown in **Supplementary Fig. S25c**, the MoNiFe (oxy)hydroxide shows higher defective O content than NiFe (oxy)hydroxide, suggesting that the MoNiFe (oxy)hydroxide might with more unsaturated oxygen sites, which is consistent with the higher oxygen activity as revealed by DFT calculation.”

(8) In Figure 1c, it is unconvincing that the ultrathin (oxy)hydroxide layer uniformly covered the carbon fiber. In fact, there are evident morphology difference between MoNiFe/CC and ultrathin MoS₂ nanosheets (Figure 1b). To clearly show the ultrathin layer morphology, high-definition SEM images should be provided.

Response: We thank the reviewer for the valuable suggestion. The high-definition SEM images have been provided to show the ultrathin layer of MoNiFe (oxy)hydroxide coating on carbon cloths, as shown in **Figure A20**.

Figure A20. a-b, The SEM images of MoNiFe (oxy)hydroxide with low magnification (**a**) and high magnification (**b**).

The **Fig. 1** in manuscript was changed to:

Fig 1. Preparation and characterizations of the MoNiFe (oxy)hydroxide. a, Schematic illustration of the preparation process of the MoNiFe (oxy)hydroxide. **b,c,** Scanning electron microscopy (SEM) images of the MoS₂ nanosheet template (**b**) and MoNiFe (oxy)hydroxide (**c**). **d,** Atomic force microscopy (AFM) image of the MoNiFe (oxy)hydroxide flakes. The inset figure is the corresponding line-trace height profile across a MoNiFe (oxy)hydroxide flake. **e-h,** High resolution transmission electron microscopy (HRTEM) images with low magnification (**e**) and high magnification (**f**), selected area electron diffraction (SAED) pattern (**g**) and energy dispersive spectroscopy (EDS) mapping (**h**) for the MoNiFe (oxy)hydroxide with Ni:Fe ratio of 73 : 27.

(9) In this manuscript, the Mo dopant is responsible for the activity improvement.

Therefore, the performance differences among samples with various Fe contents may arise from the amount variation of Mo dopant. The authors should precisely determine the metal contents of different samples by ICP spectroscopy.

Response: We thank the reviewer for the valuable suggestion. The metal contents of different samples were determined by ICP measurement. As shown in **Figure A21**, the Mo contents are similar in MoNiFe (oxy)hydroxide with different Fe contents. This result suggests that the performance differences among samples with various Fe contents mainly arise from the amount variation of Fe dopant, instead of the Mo contents.

Figure A21. The metal contents determined by ICP measurement for MoNiFe-x% (oxy)hydroxide, (x=0%, 5%, 27%, 50%, 85%, 100%). The error bar represents the standard deviation of results obtained from multiple samples.

Page 6, line 26 in manuscript, added: “*All the MoNiFe-x% exhibited similar Mo content (Supplementary Fig. S7).*”

The **Figure A21** has been added in the Supplementary Information as *Supplementary Fig. S7*

(10) It should be noted that the structures of catalysts may have considerable changes after OER. The previous reports reveal that Mo element thoroughly leached from the electrodes and the reconstructed Ni(Fe) oxyhydroxides are responsible for the electrocatalysis (ACS Nano 2021, 15, 13504; Adv. Funct. Mater. 2021, 31, 2101792;

Adv. Mater. 2021, 33, 2007344). To determine the active structure, characterization of catalysts should be measured after OER, such as SEM, TEM, XRD, XPS and Raman spectra.

Response: We thank the reviewer for the valuable suggestion. As suggested by the reviewer, we carried out systematical characterizations of MoNiFe (oxy)hydroxide after chronopotentiometry (CP) measurement.

Figure A22 shows the SEM images of MoNiFe (oxy)hydroxide after CP measurement. The morphology of the MoNiFe (oxy)hydroxide showed negligible changes after CP measurement. The MoNiFe (oxy)hydroxide layer still uniformly coated on carbon cloths.

Figure A22. a-d, The SEM images of the MoNiFe (oxy)hydroxide before (a-b) and after (c-d) CP measurement at 100 mA/cm² for 65 h.

The metal contents in MoNiFe (oxy)hydroxide before and after CP measurement were determined by ICP measurement (**Figure A23**). The changes in the cation contents are within experimental error, suggesting that the Ni, Fe, and Mo contents in MoNiFe (oxy)hydroxide remained almost unchanged after CP measurement.

Figure A23. The comparison of Ni, Fe, Mo content in MoNiFe (oxy)hydroxide before and after CP measurement.

The crystal structure of MoNiFe (oxy)hydroxide after CP measurement was identified by high-resolution transmission electron microscopy (HRTEM) (**Figure A24**). The spacing between two adjacent lattice planes was quantified to be 0.21 nm (**Figure A24b**), which is assigned to the (105) plane of oxyhydroxide. Such value is slightly larger than that of the pristine MoNiFe (oxy)hydroxide (0.20 nm), which suggests the slight lattice expansion during CP measurement. The selected area electron diffraction (SAED) pattern of the MoNiFe (oxy)hydroxide after CP measurement shows clear diffraction rings of (105) and (110) plane for Ni-based oxyhydroxide (PDF-#06-0075) (**Figure A24c**). The diffraction rings for Ni-based hydroxide were not observed, indicating the complete conversion of hydroxide to oxyhydroxide during long-time CP measurement. As revealed in the SEM-EDS (**Figure A25**) and TEM-EDS (**Figure A24d**) mapping, the distribution of Mo, Ni, Fe elements in MoNiFe is uniform after CP measurement, and are the same as the pristine one.

Figure A24. a-b, High resolution transmission electron microscopy (HRTEM) images with low magnification (**a**) and high magnification (**b**) of MoNiFe (oxy)hydroxide after CP measurement at 100 mA/cm^2 for 65 h. **c**, The corresponding selected area electron diffraction (SAED) pattern. **d**, Energy dispersive spectroscopy (EDS) mapping of Ni, Fe and Mo elements.

Figure A25. a-b, The SEM-EDS mapping of the MoNiFe (oxy)hydroxide before (**a**) and after (**b**) CP measurement at 100 mA/cm^2 for 65 h.

The chemical composition of MoNiFe (oxy)hydroxide after CP measurement was identified by X-ray photoelectron spectroscopy (XPS) (**Figure A26**). Both the Fe $2p$ and Ni $2p$ remained unchanged after CP measurement comparing to the pristine one. In

the O 1s XPS spectra, the peak of adsorbed H₂O on the surface increase obviously after CP measurement while the peaks of defective O and lattice O do not show noticeable changes. All these results above demonstrate that the structure and composition of MoNiFe (oxy)hydroxide remained almost unchanged during long-time operation under OER conditions, explaining the high stability of the catalysts.

Figure A26. a-c, The Fe 2p (a), Ni 2p (b), and O 1s (c) XPS spectra of MoNiFe (oxy)hydroxide before and after CP measurement.

The **Figure A22** has been added into Supplementary Information as Supplementary Fig. S13.

The **Figure A23** has been added into Supplementary Information as Supplementary Fig. S14.

The **Figure A24** has been added into Supplementary Information as Supplementary Fig. S15.

The **Figure A25** has been added into Supplementary Information as Supplementary Fig. S16.

The **Figure A26** has been added into Supplementary Information as Supplementary Fig. S17.

Page 9, line 14 in manuscript, added: “*The structure and composition of MoNiFe (oxy)hydroxide catalyst remain unchanged after the long-time operation, as revealed by SEM, TEM, EDS, XPS, and ICP characterizations (Supplementary Fig. S13-S17, note 2).*”

“Supplementary note 2

*The characterizations of MoNiFe (oxy)hydroxide after CP measurement were carried out. **Supplementary Fig. S13** shows the SEM images of MoNiFe (oxy)hydroxide after CP measurement. The morphology of the MoNiFe (oxy)hydroxide showed negligible change after CP measurement. The MoNiFe (oxy)hydroxide layer still uniformly coated on carbon cloths.*

*The metal contents in MoNiFe (oxy)hydroxide before and after CP measurement were determined by ICP measurement (**Supplementary Fig. S14**). The changes in the cation contents are within experimental error, suggesting that the Ni, Fe, and Mo contents in MoNiFe (oxy)hydroxide remained almost unchanged after CP measurement.*

*The crystal structure of MoNiFe (oxy)hydroxide after CP measurement was identified by high-resolution transmission electron microscopy (HRTEM) (**Supplementary Fig. S15**). The spacing between two adjacent lattice planes was quantified to be 0.21 nm (**Supplementary Fig. S15b**), which is assigned to the (105) plane of oxyhydroxide. Such value is slightly larger than that of the pristine MoNiFe (oxy)hydroxide (0.20 nm), which suggests the lattice expansion during CP measurement. The selected area electron diffraction (SAED) pattern of the MoNiFe (oxy)hydroxide after CP measurement shows clear diffraction rings of (105) and (110) plane for Ni-based oxyhydroxide (PDF-#06-0075) (**Supplementary Fig. S15c**). The diffraction rings for Ni-based hydroxide were not observed, indicating the complete conversion of hydroxide to oxyhydroxide during long-time CP measurement. As revealed in the SEM-EDS (**Supplementary Fig. S16**) and TEM-EDS (**Supplementary Fig. S15d**) mapping, the distribution of Mo, Ni, Fe elements in MoNiFe is uniform after CP measurement, and are the same as the pristine one.*

*The chemical composition of MoNiFe (oxy)hydroxide after CP measurement was identified by X-ray photoelectron spectroscopy (XPS) (**Supplementary Fig. S17**). Both the Fe 2p and Ni 2p remained unchanged after CP measurement comparing to the pristine one. In the O 1s XPS spectra, the peak of adsorbed H₂O on the surface increase*

obviously after CP measurement, while the peaks of defective O and lattice O do not show noticeable changes. All these results above demonstrate that the structure and composition of MoNiFe (oxy)hydroxide remained almost unchanged during long-time operation under OER conditions, explaining the high stability of the catalysts.”

Response to Reviewer #2

COMMENTS TO AUTHOR:

The authors focused on Mo-doped NiFe (oxy)hydroxides, which is a strong candidate of a catalyst for the oxygen evolution reaction (OER). In their study, the Mo-doped NiFe (oxy)hydroxides were synthesized using the sacrificial template-directed approach. It is suggested that Mo-doped NiFe (oxy)hydroxides can be followed the lattice oxygen mechanism (LOM), and the authors clarified the importance of oxygen activity in the material lattice structure as an electrocatalyst for OER. They applied the various analyses to reveal the role of oxygen in atomic level and electric states by using both experimental and theoretical methods. The ideas are interesting and the findings in this work can be applied for a guideline for future catalyst design; however, I cannot provide my final recommendation, because some key information is missing. Therefore, please address the issues below and I need to read a new version to make my final recommendation.

Response: We thank the reviewer for the valuable comments and suggestions. All the concerns raised by the reviewer have been addressed in detail as follows.

1) Are there any previous studies on the sacrificial template-directed approach? Is this method originally developed in the authors group? Please add the information in Introduction section. Also, are there any previous studies on Mo-doped NiFe (oxy)hydroxide or other metal-doped NiFe (oxy)hydroxide? Why did the authors choose Mo as a dopant? Authors should mention more clearly on the purpose of the work (which part is the novel approach?) and the reason why they choose it.

Response: We thank the reviewer for the insightful question.

Previous to our work, several research groups reported self-reconstructed amorphous phase or (oxy)hydroxides formed during OER as the active phase on the surface of perovskite oxide (*Nature communications*, 2020, 11(1): 1-10.; *Journal of the American Chemical Society*, 2021, 143(7): 2741-2750.), nitride (*Angewandte Chemie*, 2015, 127(49): 14923-14927.), and phosphide (*Energy & Environmental Science*, 2015, 8(8): 2347-2351.), etc. (*ACS Energy Letters* 2017, 2(8): 1937-1938; *Nature Energy*, 2020, 5(11): 881-890.). Most of these works utilized a near-surface reconstruction, leading to a multicomponent core-shell structure with inadequate use of inner components. In some more recent works, the authors generated highly activity OER catalyst by a deep self-reconstruction, i.e., full conversion to another phase during OER reaction (*Matter*, 2020, 3(6): 2124-2137; *Advanced Materials*, 2021: 2007344; *Matter*, 2021, 4(9): 2850-2873). While all the pioneering works mentioned above have demonstrated that self-reconstruction of pre-catalysts during OER provide an effective way to achieve highly active catalysts, the impact of the potential doping from the pre-catalysts on the electronic structure and OER activity of final catalysts is still lack of investigation.

There have been previous works that used cation doping to modulate the electronic structure and catalytic activity of hydroxide catalysts. For instance, Jin et al. (*ACS Catalysis*, 2018, 8(3): 2359-2363.) reported Mo- and Fe modified Ni(OH)₂/NiOOH nanosheets as highly active and stable OER catalysts. The authors believe that the synergistic effect of Mo and Fe leads the Ni sites to have higher interaction strength with OER intermediates. Other dopants such as Al (*ACS Applied Energy Materials*, 2019, 2(5): 3488-3499.) and V (*Advanced Functional Materials*, 2021: 2100614.; *Nature Communications*, 2018, 9(1): 1-12; *Advanced Energy Materials*, 2018, 8(15): 1703341.) were also reported to effectively tune the electronic structure of Ni-based (oxy)hydroxide, thereby boosting the OER activity. While all these works demonstrate cation doping as effective approach for regulating the OER reaction kinetics, most of them consider the impact of cation doping from the aspect of tuning the reaction barrier of OER following the AEM mechanism. Herein, we demonstrate that cation doping not

only critically impacts the oxygen activity of the catalyst, but also impacts the OER activity by changing the reaction kinetics following a LOM reaction mechanism.

The reason we choose MoS₂ as the template is that MoS₂ could be easily oxidized during OER and be dissolved into the solution, as reported in literatures (*Nature communications*, 2020, 11(1): 1-12; *Matter*, 2020, 3(6): 2124-2137; *Cell Reports Physical Science*, 2020, 1(11): 100241.). Furthermore, it is easy to synthesize MoS₂ nanosheets with abundant surface-active sites to adsorb metal cations for the construction of MoS₂/NiFe LDH pre-catalysts. The MoS₂/NiFe LDH pre-catalysts are directly converted to NiFe (oxy)hydroxide with Mo doping by complete reconstruction. The obtained catalyst exhibited outstanding OER activity and stability.

In addition to good performance, we found that Mo doping in NiFe (oxy)hydroxide critically impacts the electronic structure and lattice oxygen activity, which determines the OER activity of (oxy)hydroxide by modulating the reaction barrier in OER mechanism. Mo doping led to higher lattice oxygen activity, which was understood by the weakened metal-oxygen bond, upshifted O 2*p* center relative to Fermi level, enlarged *U* values, and lower oxygen vacancy formation energy. Such activation of lattice oxygen shifted the potential determining step from oxygen vacancy formation for the NiFe (oxy)hydroxide to the *OOH deprotonation for the MoNiFe (oxy)hydroxide, resulting in strongly enhanced intrinsic OER activity. This new mechanism understanding can help guide the rational design of highly activity electrocatalysts based on oxygen activity regulation.

Page 3, line 8 in manuscript, changed from: “*Utilizing the self-reconstruction or material leaching effects during operation provides an alternative approach for constructing novel high-performance OER catalysts.*”

To “*While all the pioneering works mentioned above have demonstrated the self-reconstruction or material leaching effects during operation as an effective way to achieve highly active catalysts, the impacts of pre-catalysts on the activity of final catalysts are still lack of investigation*”

2) On Fig. 4b, the meaning of “TM” is necessary in figure caption

Response: We very much appreciate the reviewer’s careful reading of our manuscript. “TM” in Fig. 4b is referred to transition metal. The caption of Fig. 4 in the manuscript has been revised as follows.

Fig 5. Lattice oxygen activity determined by density functional theory (DFT) calculations. **a**, Crystal orbital Hamilton populations (COHP) of the Ni-O bond in NiFe and MoNiFe (oxy)hydroxide. **b**, The integrated -COHP up to Fermi level comparison of Ni-O and Fe-O in NiFe and MoNiFe (oxy)hydroxide. **TM is referred to transition metal.** **c**, Projected density of states of NiFe and MoNiFe (oxy)hydroxide. **The anti-bonding states below the Fermi level were highlighted by dash circles.** **d**, Schematic band diagrams of NiFe and MoNiFe (oxy)hydroxide. The d-orbitals split into electron-filled lower Hubbard band (LHB) and empty upper Hubbard band (UHB) with an energy difference of U . **e**, The oxygen vacancy formation energy ($E_{f,vac}$) of NiFe and MoNiFe (oxy)hydroxide.

3) On Fig. 5 a-c of sXAS results, authors should add the reference “Rojas, T. C. et al. *J. Mater. Chem.* 9, 1011-1017 (1999)” in addition to Ref#33, because the corresponding discussion in Ref#33 refers the paper. If possible, the explanation of the result of Fig. 5a is needed, especially about the relationship of the sXAS result and the discussion of “an increased electron density around the oxygen ligands after Mo doping.”

Response: We thank the reviewer for the valuable suggestion. The O K-edge sXAS spectra consist of two characteristic peaks at ~533.5 eV and ~540 eV, which were assigned to the O 2*p* - metal 3*d* hybridization and the O 2*p* - metal 4*sp* hybridization (*Advanced Functional Materials*, 2018, 28(44): 1803272; *Journal of Materials Chemistry*, 1999, 9(4): 1011-1017.), respectively. As shown in **Figure A27a**, the intensity of O K-edge decreased after Mo doping, indicating a decrease in unoccupied density of states (*Nature materials*, 2011, 10(10): 780-786.) and a weakening of 3*d*/4*sp*-2*p* hybridization (*Advanced Functional Materials*, 2018, 28(44): 1803272; *Chemistry of Materials* 2014, 26 (8), 2496-2501.). Such decreased intensity in O K-edge spectra, accompanied with the increased intensity of Ni L-edge and Fe L-edge peak for MoNiFe (oxy)hydroxide (**Figure A27b,c**), suggested a higher electron density at the O site and a lower electron density at the Ni/Fe sites, a higher ionic metal-oxygen bond (*Nature materials*, 2011, 10(10): 780-786; *Journal of Materials Chemistry*, 2009, 19(37): 6804-6809.). This result is consistent with the weaker metal-oxygen bond after Mo doping, as revealed by the COHP calculation (Fig. 5b in manuscript).

Figure A27. a-c, O K-edge (a), Ni L-edge (b), and Fe L-edge (c) Soft X-ray absorption spectroscopies (sXAS) of NiFe and MoNiFe (oxy)hydroxide.

Page 18, line 10 in manuscript, added: “*The O K-edge sXAS spectra consist of two characteristic peaks at ~533.5 eV and ~540 eV, which were assigned to the O 2p - metal 3d hybridization and the O 2p - metal 4sp hybridization^{44, 45}. As shown in Fig. 6a, the intensity of O K-edge decreased after Mo doping, indicating a decrease in unoccupied density of states⁴⁶ and a weakening of 3d/4sp-2p hybridization^{44, 47}. Such decreased intensity in O K-edge spectra, accompanying with the increase intensity of Ni L-edge and Fe L-edge peak for MoNiFe (oxy)hydroxide (Fig. 6b,c), suggested a higher electron density at the O site and a lower electron density at the Ni/Fe sites, a higher ionic metal-oxygen bond^{46, 48}. This result is consistent with the weaker metal-oxygen bond after Mo doping, as revealed by the COHP calculation (Fig. 5b).*”

The reference recommended by the reviewer and other related literatures have been added to the revised manuscript:

Page 18, line 15 in manuscript, “[44] Yang, J., et al. *Surface-confined fabrication of ultrathin nickel cobalt-layered double hydroxide nanosheets for high-performance supercapacitors. Adv. Funct. Mater.* 28, 1803272 (2018). [45] Teresa C. Rojas, J. n. S. n.-L. p., Mari ´ a J. Sayague ´s, Ettireddy P. Reddy, Alfonso Caballero, Asuncion Fern ´andez. *Preparation, characterization and thermal evolution of oxygen passivated nanocrystalline cobalt. J. Mater. Chem.* 9, 1011-1017 (1999).” were cited in the revised manuscript as reference [44] and [45] in “*The O K-edge sXAS spectra consist two characteristic peaks at ~533.5 eV and ~540 eV, which were assigned to the O 2p - metal 3d hybridization and the O 2p - metal 4sp hybridization^{44,45}*”

Page 18, line 17 in manuscript, “[44] Yang, J., et al. *Surface-confined fabrication of ultrathin nickel cobalt-layered double hydroxide nanosheets for high-performance supercapacitors. Adv. Funct. Mater.* 28, 1803272 (2018). [46] Liang, Y., et al. *Co₃O₄ nanocrystals on graphene as a synergistic catalyst for oxygen reduction reaction. Nat. Mater.* 10, 780-786 (2011). [47] Kwon, J.-H., et al. *Nanoscale spin-state ordering in LaCoO₃ epitaxial thin films. Chem. Mater.* 26, 2496-2501 (2014).” were cited in the revised manuscript as reference [44], [46], and [47] in “*the intensity of O k-edge*

decreased after Mo doping, indicating a decrease in unoccupied density of states⁴⁶ and a weakening of 3d/4sp-2p hybridization^{44, 47}.”

Page 18, line 21 in manuscript, “[48] Zhou, J. G., et al. *Electronic structure of TiO₂ nanotube arrays from X-ray absorption near edge structure studies. J. Mater. Chem. 19, 6804 (2009).*” was cited in the revised manuscript as reference [48] in “suggested a higher electron density at the O site and a lower electron density at the Ni/Fe sites, a higher ionic metal-oxygen bond^{46, 48}.”

4) The authors should show the raw data of sXAS and the information of their background data, which is subtracted from the raw data, in Supporting Information. The background data is important for the reliability of the data, because the results in Fig. 5a-c have the clear difference only in peak intensity.

Response: We thank the reviewer for the valuable suggestion. The raw data of sXAS spectra and the corresponding background were provided in **Figure A28-30**. For the Ni L-edge and Fe L-edge sXAS spectra, the background was subtracted, and the peak intensity was compared in **Fig. 6** in the manuscript. For the O K-edge sXAS spectra, the step jump of the background (the height between the dash lines) was normalized to be 1 (*Nature materials, 2011, 10(10): 780-786.*).

Figure A28. a-b, The raw data of the Ni L-edge of NiFe (a) and MoNiFe (b) (oxy)hydroxide. The red lines represent the background.

Figure A29. a-b, The raw data of the Fe L-edge of NiFe (a) and MoNiFe (b) (oxy)hydroxide. The red lines represent the background.

Figure A30. a-b, The raw data of the O K-edge of NiFe (a) and MoNiFe (b) (oxy)hydroxide. The dash lines represent the step jump of the background at O K-edge spectra.

The **Figure A28** have been added into Supplementary Information as *Supplementary Fig. S30*.

The **Figure A29** have been added into Supplementary Information as *Supplementary Fig. S31*.

The **Figure A30** have been added into Supplementary Information as *Supplementary Fig. S32*.

Page 20, line 4 in manuscript, added: “*The step at the background of O K-edge spectra was normalized to be 1⁴⁶. The background of Ni L-edge and Fe L-edge spectra were*

subtracted. The raw data and the background of sXAS spectra are shown in Supplementary Fig. S30-32.”

5) Do the authors have data for benchmark on the mass activity? (for example, commercial RuO₂ or IrO₂. I think commercial IrO₂ is more popularly used for a benchmarking.)

Response: We thank the reviewer for the valuable question. The OER performance of MoNiFe (oxy)hydroxide was compared with the benchmark RuO₂ and IrO₂ samples, as shown in **Figure A31a**. The MoNiFe (oxy)hydroxide delivered an overpotential of 242 mV at the current density of 10 mA/cm², which was much lower than RuO₂ (277 mV) and IrO₂ (363 mV). To reach a current density of 100 mA/cm², the MoNiFe (oxy)hydroxide required only an overpotential of 290 mV, while RuO₂ and IrO₂ needed 385 mV and 466 mV, respectively. To assess the intrinsic activity of the catalysts, the mass activity was obtained by normalizing the CV curves by loading mass (**Figure A31b**). The MoNiFe (oxy)hydroxide delivered a mass activity of 1910 A/g at the overpotential of 300 mV, which is much higher than that of 112 A/g and 5.56 A/g for RuO₂ and IrO₂, respectively (**Figure A31c**).

Figure A31. a-b, Cyclic voltammetry polarization curves normalized by geometric area **(a)** and loading mass **(b)** for MoNiFe (oxy)hydroxide, RuO₂ and IrO₂. **c**, The specific mass activity of MoNiFe (oxy)hydroxide, RuO₂ and IrO₂ at the overpotential of 300 mV.

The **Figure A31** have been added in Supplementary Information as Supplementary Fig. S12.

Page 9, line 1 in manuscript, added: *“In addition, MoNiFe-27% (oxy)hydroxide also delivered a noticeable lower overpotential and higher mass activity than the benchmark RuO₂ and IrO₂ catalysts (Supplementary Fig. S12, note 1).”*

Page 25, line 1 in Supplementary Information, added:

“Supplementary note 1

The OER performance of MoNiFe (oxy)hydroxide was compared with the benchmark RuO₂ and IrO₂ samples, as shown in Supplementary Fig. S12a. The MoNiFe (oxy)hydroxide delivered an overpotential of 242 mV at the current density of 10 mA/cm², which was much lower than RuO₂ (277 mV) and IrO₂ (363 mV). To reach a current density of 100 mA/cm², the MoNiFe (oxy)hydroxide required only an overpotential of 290 mV, while RuO₂ and IrO₂ needed 385 mV and 466 mV, respectively. To assess the intrinsic activity of the catalysts, the mass activity was obtained by normalizing the CV curves by loading mass (Supplementary Fig. S12b). The MoNiFe (oxy)hydroxide delivered a mass activity of 1910 A/g at the overpotential of 300 mV, which is much higher than that of 112 A/g and 5.56 A/g for RuO₂ and IrO₂, respectively (Supplementary Fig. S12c).”

Response to Reviewer #3

COMMENTS TO AUTHOR:

The manuscript presents combined experimental and computational results on Mo-doped NiFe oxy-hydroxide with modulated oxygen activity for alkaline oxygen evolution reaction (OER). The system and the findings are of potential interest to the field in general and to readers of Nature Commun. specifically, but the information provided in several key areas is insufficient to render sufficient support for the conclusions drawn. Therefore, the manuscript does not warrant publication in the present form, as discussed below.

Response: We thank the reviewer for the valuable comments and suggestions. All the concerns raised by the reviewer have been addressed in detail as follows.

1. The paper fails to discuss, let alone cite, key papers on the specific topic of lattice oxygen like Chorkendorff et al. “Impact of nanoparticle size and lattice oxygen on water oxidation on NiFeO_xH_y”, Nat. Catal., DOI:10.1038/s41929-018-0162-x (2018). More importantly regarding the proposed role of the lattice oxygen, Fig 3. states: “and isotopic labeling experiments”, but where are the oxygen isotope (¹⁶O vs. ¹⁸O) experiments? These results should be provided to support the proposed mechanism and discussed against relevant literature.

Response: We thank the reviewer for the valuable comments and suggestions. All the concerns raised by the reviewer have been addressed in detail as follows.

1) Adding of necessary references

The reference mentioned by the reviewer and other key papers related published recently on the topic of lattice oxygen have been added to the revised manuscript:

Page 10, line 12 in manuscript, “[31] Roy, C., et al. *Impact of nanoparticle size and lattice oxygen on water oxidation on NiFeO_xH_y*. Nat. Catal. 1, 820-829 (2018). [32] Wen, Y., et al. *Stabilizing highly active Ru sites by suppressing lattice oxygen participation in acidic water oxidation*. J. Am. Chem. Soc. 143, 6482-6490 (2021). [33] Shi, Z., et al. *Confined Ir single sites with triggered lattice oxygen redox: Toward boosted and sustained water oxidation catalysis*. Joule 5, 2164-2176 (2021).” were cited in the revised manuscript as reference [31-33] in “*suggesting the presence of ¹⁶O₂, ¹⁶O¹⁸O and ¹⁸O₂ in the gas production³¹⁻³³.*”

Page 10, line 13 in manuscript, “[14] Zhang, N., et al. *Lattice oxygen activation enabled by high-valence metal sites for enhanced water oxidation*. Nat. Commun. 11, 4066 (2020). [30] Ferreira de Araujo, J., Dionigi, F., Merzdorf, T., Oh, H. S., Strasser, P. *Evidence of Mars-Van-Krevelen mechanism in the electrochemical oxygen evolution on Ni-based catalysts*. Angew. Chem. Int. Ed. Engl. 60, 14981-14988 (2021).” were cited in the revised manuscript as reference [14] and [30] in “*This result implies that both NiFe and MoNiFe (oxy)hydroxide follow the LOM mechanism^{14, 30}.*”

2) Oxygen isotope (^{16}O vs. ^{18}O) experiments

As suggested by the reviewer, we carried out ^{18}O isotope-labeling experiments, in-situ differential electrochemical mass spectrometry (DEMS) measurements, and ex-situ Raman spectra measurement to validate the participation of lattice oxygen in OER.

First, NiFe and MoNiFe (oxy)hydroxides were labeled with ^{18}O -isotopes by potentiostatic reaction at 1.65 V (vs. Ag/AgCl) for 30 min in KOH solution with H_2^{18}O . Afterward, the ^{18}O -labeled catalysts were rinsed with H_2^{16}O for several times to remove the remaining H_2^{18}O . Cyclic voltammograms (CV) measurements were carried on the ^{18}O -labeled samples in KOH solution with H_2^{16}O . The in-situ DEMS measurements results on the ^{18}O -labeled NiFe and MoNiFe (oxy)hydroxide showed the signals of $m/z = 32$, $m/z = 34$, and $m/z = 36$ (**Figure A32**), suggesting the presence of $^{16}\text{O}_2$, $^{16}\text{O}^{18}\text{O}$ and $^{18}\text{O}_2$ in the gas production. These results imply that both NiFe and MoNiFe (oxy)hydroxide follow the LOM mechanism. The mass spectrometric cyclic voltammograms (MSCVs), which plot the real-time gas product contents as a function of applied potential, can provide direct comparison about the participation of lattice oxygen in OER process. The ^{18}O -labeled MoNiFe (oxy)hydroxide is with noticeably higher contents of $^{18}\text{O}^{16}\text{O}$ and $^{18}\text{O}_2$ in the reaction product than the ^{18}O -labeled NiFe (oxy)hydroxide (**Figure A33**), implying that the lattice oxygen of MoNiFe (oxy)hydroxide participated more actively into the OER reaction than that of NiFe (oxy)hydroxide.

Figure A32. a-b, The differential electrochemical mass spectrometry signals of $^{16}\text{O}_2$

(I₃₂), ¹⁶O¹⁸O (I₃₄), and ¹⁸O₂ (I₃₆) as a function of collecting time on NiFe (a) and MoNiFe (b) (oxy)hydroxide.

Figure A33. a-c, The ¹⁶O₂ (a), ¹⁶O¹⁸O (b), and ¹⁸O₂ (c) signals on NiFe (oxy)hydroxide. d, The CV curve of NiFe (oxy)hydroxide during DEMS measurement. e-g, The ¹⁶O₂ (e), ¹⁶O¹⁸O (f), and ¹⁸O₂ (g) signals on MoNiFe (oxy)hydroxide. h, The corresponding CV curve of MoNiFe (oxy)hydroxide during DEMS measurement. The ¹⁶O₂ signal was normalized to 1, and the ¹⁶O¹⁸O and ¹⁸O₂ signals are relative to the ¹⁶O₂ signal. All data were taken from the first cycle.

In addition to the DEMS measurement, we carried out additional Raman spectroscopy measurement to confirm the lattice oxygen activation during OER. First, the NiFe and MoNiFe (oxy)hydroxide were completely activated in KOH solution with H₂¹⁶O. Then, the (oxy)hydroxide catalysts were subjected to a constant potential of 1.65 V (vs. RHE) for 30 min in KOH solution with H₂¹⁸O. Raman spectra of obtained samples with ¹⁸O-labeling are shown in **Figure A34**. The Raman peaks of NiFe and MoNiFe (oxy)hydroxide with ¹⁸O-labeling shifted towards lower wavenumber because of the impact oxygen mass on the vibration mode (*Angewandte Chemie International Edition, 2020, 59(21): 8072-8077; Angewandte Chemie, 2019, 131(30): 10401-10405.*). The MoNiFe (oxy)hydroxide showed a more obvious Raman shift to lower wavenumber than NiFe (oxy)hydroxide, suggesting a more oxygen in the lattice got

exchanged between the lattice oxygen and electrolytes during OER on the MoNiFe (oxy)hydroxide.

The DEMS and Raman spectra results on the ^{18}O -labeled samples consistently suggest that Mo doping in NiFe (oxy)hydroxide promoted the lattice oxygen to exchange with electrolyte during OER, which is in accord with the conclusion we got in our previous manuscript. The newly added ^{18}O -labeling experiments and related discussion were added to the revised manuscript.

Figure A34. a-b, Raman spectra of NiFe (a) and MoNiFe (b) (oxy)hydroxide with (red) and without (black) ^{18}O -labeling.

Page 10, line 1 in manuscript, changed: “To reveal the mechanism of the high intrinsic OER activity of MoNiFe (oxy)hydroxides, DFT calculations and electrochemical experiments were carried out on NiFe-27% and MoNiFe-27% (oxy)hydroxide.” to “To reveal the mechanism of the high intrinsic OER activity of MoNiFe (oxy)hydroxides, *isotope-labeling experiments and DFT calculations were carried out on NiFe-27% and MoNiFe-27% (oxy)hydroxide.*”

Page 10, line 5 in manuscript, added: “To validate the participation of lattice oxygen in OER for our material systems, the ^{18}O isotope-labeling experiments were carried out using the procedure described in the experimental section. In-situ differential electrochemical mass spectrometry (DEMS) measurements results on the ^{18}O -labeled NiFe and MoNiFe (oxy)hydroxide showed the signals of $m/z = 32$, $m/z = 34$, and $m/z = 36$ (Supplementary Fig. S18), suggesting the presence of $^{16}\text{O}_2$, $^{16}\text{O}^{18}\text{O}$, and $^{18}\text{O}_2$ in the gas production³¹⁻³³. This result implies that both NiFe and MoNiFe (oxy)hydroxide

follow the LOM mechanism^{14, 30}. The mass spectrometric cyclic voltammograms (MSCVs) which plot the real time gas product contents as a function of applied potential can provide direct comparison about the participation of lattice oxygen in OER process. The ¹⁸O-labeled MoNiFe (oxy)hydroxide is with noticeably higher contents of ¹⁶O¹⁸O and ¹⁸O₂ in the reaction product than the ¹⁸O-labeled NiFe (oxy)hydroxide (Fig. 3a-d and Supplementary Fig. S19-S20), implying the lattice oxygen of MoNiFe (oxy)hydroxide participated more actively into the OER reaction than that of NiFe (oxy)hydroxide.

In addition to the DEMS measurement, the Raman spectra were also used to confirm participation of lattice oxygen in OER. The Raman peaks of NiFe and MoNiFe (oxy)hydroxide with ¹⁸O-labeling shifted towards lower wavenumber (Fig. 3e-f) because of the impact oxygen mass on the vibration mode^{2, 34}. The MoNiFe (oxy)hydroxide shows more obvious shift than the NiFe (oxy)hydroxide, suggesting more lattice oxygen got exchanged with the electrolytes during OER. The Raman spectra and DEMS results on the ¹⁸O-labeled samples consistently suggest that Mo doping in NiFe (oxy)hydroxide effectively promoted the lattice oxygen to participate in the OER reaction.”

Page 11, line 1 in manuscript, added:

Fig 3. Evidence of lattice oxygen participating in OER provided by ^{18}O isotope-labeling experiments. a-d, Mass spectrometric cyclic voltammograms results showing different gaseous products content of OER reaction as a function of applied potential for the ^{18}O -labeled samples: $^{16}\text{O}^{18}\text{O}$ for NiFe (oxy)hydroxide (a) and MoNiFe (oxy)hydroxide (b), and $^{18}\text{O}_2$ for NiFe (oxy)hydroxide (c) and MoNiFe (oxy)hydroxide (d). The contents of all the species were normalized by the amount of $^{16}\text{O}_2$ in the reaction products; e-f, Raman spectra of NiFe (e) and MoNiFe (f) (oxy)hydroxide with (red) and without (black) ^{18}O -labeling.

Page 26, line 6 in manuscript, added:

“ ^{18}O -labeling experiment: NiFe and MoNiFe (oxy)hydroxides were labeled with ^{18}O -isotopes by potentiostatic reaction at 1.65 V (vs. Ag/AgCl) for 30 min in KOH solution with H_2^{18}O . Afterward, the ^{18}O -labeled catalysts were rinsed with H_2^{16}O for several times to remove the remaining H_2^{18}O .

DEMS measurements: DEMS measurements were carried out using a QAS 100 device (Linglu Instruments, Shanghai). The NiFe or MoNiFe (oxy)hydroxide with ^{18}O -labeling, a Ag/AgCl electrode prefilled with saturated KCl aqueous solution, and a Pt mesh were used as working electrode, reference electrode, and counter electrode, respectively. CV measurement was performed in KOH solution with H_2^{16}O with a scan rate of 5 mV/s. In the meantime, gas products with different molecular weights were detected in real time by mass spectroscopy.”

Page 11 in Supplementary Information, added:

Supplementary Fig. S19. a-b, The cyclic voltammograms curves of NiFe (a) and MoNiFe (b) (oxy)hydroxide during DEMS measurement.

Page 12 in Supplementary Information, added:

Supplementary Fig. S20. a-b, The DEMS signal of $m/z=32$ ($^{16}\text{O}_2$) of NiFe (a) and MoNiFe (b) (oxy)hydroxide as a function of applied potential during OER measurement.

The **Figure A32** has been added in Supplementary Information as *Supplementary Fig. S18*.

The related references on oxygen isotope experiment were cited in the revised manuscript:

Page 10, line 7 in manuscript, “[31] Roy, C., et al. Impact of nanoparticle size and lattice oxygen on water oxidation on NiFeO_xH_y . *Nat. Catal.* 1, 820-829 (2018). [32]

Wen, Y., et al. Stabilizing highly active Ru sites by suppressing lattice oxygen participation in acidic water oxidation. *J. Am. Chem. Soc.* 143, 6482-6490 (2021). [33] Shi, Z., et al. Confined Ir single sites with triggered lattice oxygen redox: Toward boosted and sustained water oxidation catalysis. *Joule* 5, 2164-2176 (2021).” was cited in the revised manuscript as reference [31], [32] and [33] in “suggesting the presence of $^{16}\text{O}_2$, $^{16}\text{O}^{18}\text{O}$, and $^{18}\text{O}_2$ in the gas production³¹⁻³³.”

Page 10, line 8 in manuscript, “[14] Zhang, N., et al. Lattice oxygen activation enabled by high-valence metal sites for enhanced water oxidation. *Nat. Commun.* 11, 4066 (2020). [30] Ferreira de Araujo, J., Dionigi, F., Merzdorf, T., Oh, H. S., Strasser, P. Evidence of Mars-Van-Krevelen mechanism in the electrochemical oxygen evolution on Ni-based catalysts. *Angew. Chem. Int. Ed. Engl.* 60, 14981-14988 (2021).” was cited in the revised manuscript as reference [14] and [30] in “This result implies that both NiFe and MoNiFe (oxy)hydroxide follow the LOM mechanism^{14,30}.”

Page 10, line 23 in manuscript, “[34] Lee, S., Banjac, K., Lingenfelder, M., Hu, X. Oxygen isotope labeling experiments reveal different reaction sites for the oxygen evolution reaction on nickel and nickel iron oxides. *Angew. Chem. Int. Ed. Engl.* 58, 10295-10299 (2019). [2] Lee, S., Bai, L., Hu, X. Deciphering iron-dependent activity in oxygen evolution catalyzed by nickel-iron layered double hydroxide. *Angew. Chem. Int. Ed. Engl.* 59, 8072-8077 (2020).” was cited in the revised manuscript as reference [34] and [2] in “The Raman peaks of NiFe and MoNiFe (oxy)hydroxide with ^{18}O -labeling shifted towards lower wavenumber (Fig. 3e-f) because of the impact oxygen mass on the vibration mode^{2,34}.”

2. Similarly, the work by Bent et al. on “The Role of Aluminum in Promoting Ni–Fe–OOH Electrocatalysts for the Oxygen Evolution Reaction”, *ACS Appl. Energy Mater* (2019), DOI: 10.1021/acsaem.9b00265 would provide a valuable comparison for the Mo-doped ultra-thin NiFe LDH system.

Response: We thank the reviewer for the valuable suggestions. Baker et al. (*ACS Applied Energy Materials*, 2019, 2(5): 3488-3499.) demonstrated that aluminum

doping in Ni(Fe) oxyhydroxide impacts the electronic structure of the host material, Ni-OOH, through long-range interactions, giving rise to more negative charges on the oxygen (ligand) around Al and smaller gap on the O-Ni-Fe states. Such change in electronic structure is similar to what we observed in our work, that oxygen site has higher electron density as revealed by O K-edge sXPS spectra. However, Baker et al. demonstrated that the aluminum does not change the OER mechanism of NiFe oxyhydroxide, which is different from the effects of Mo doping in NiFe (oxy)hydroxide in our work. We found that Mo doping can effectively activate the lattice oxygen, which is represented by the weakened metal-oxygen bond, upshifted O 2p center relative to Fermi level, enlarged U values, and lower oxygen vacancy formation energy. Such activation of lattice oxygen can shift the potential determining step from oxygen vacancy formation for the NiFe (oxy)hydroxide to the *OOH deprotonation for the MoNiFe (oxy)hydroxide, resulting in strongly enhanced intrinsic OER activity. Therefore, the mechanism of how Al and Mo doping improve the OER activity maybe very different.

Discussion about the different mechanisms was added in the revised manuscript. The reference recommended by the reviewer was added as reference [37] in the revised manuscript:

Page 12, line 15 in manuscript, added: “*The changes in mechanism and PDS derived from Mo doping are quite different from other cation doping reported by previous work³⁵⁻³⁷.*”

Page 12, line 15 in manuscript, “[35] Baker, J. G., et al. *The role of aluminum in promoting Ni-Fe-OOH electrocatalysts for the oxygen evolution reaction. ACS Appl. Energy Mater.* 2, 3488-3499 (2019). [36] Jiang, J., et al. *Atomic-level insight into super-efficient electrocatalytic oxygen evolution on iron and vanadium co-doped nickel (oxy)hydroxide. Nat. Commun.* 9, 2885 (2018). [37] Li, P., et al. *Tuning electronic structure of NiFe layered double hydroxides with vanadium doping toward high efficient electrocatalytic water oxidation. Adv. Energy Mater.* 8, 1703341 (2018).” was cited in the revised manuscript as reference [35], [36] and [37] in “*The changes in mechanism and PDS derived from Mo doping are quite different from other cation*

doping reported by previous work³⁵⁻³⁷.”

3. Further, the level of information provided regarding the DFT level calculations is insufficient. It is impossible to reproduce the calculations based on the information provided. What is the model system and how is it terminated? Where is Mo-placed? What reference is used to calculate the formation energy of the oxygen vacancy (O₂ gas or an H₂O-reference), etc? This data is needed to assess and discuss the findings.

Response: We thank the reviewer for the valuable suggestions. The details of the DFT calculations have been provided.

The slab models of NiFe (oxy)hydroxide used for both AEM and LOM reaction pathway were terminated by the (001) surface (**Figure A35**). For the AEM pathway, the metal site should be exposed to the reactants. Thus, two vacuum spaces were inserted along (001) and (010). To eliminate the interaction between periodic slabs, the thickness of vacuum spaces in both models was more than 10 Å. In addition, part of hydrogen atoms was removed because of the oxidation atmosphere.

Figure A35. The slab model of NiFe (oxy)hydroxide.

To find the stable configuration of Mo doping, we have built three slab models with different Mo sites (**Figure A36**). The relative stability of Mo replacement was determined by calculating the formation energy (ΔE_f), which was computed as:

$$\Delta E_f = E_{\text{slab}} - \sum N_i E_i,$$

where E_{slab} , E_i and N_i are total energies of the slab model, energy, and number of

the i -th element, respectively. The calculated results show that the Mo atom favors replacing the Ni site and exposes to the vacuum as this model with the most negative ΔE_f (Figure A36d).

Figure A36. Doping (Ni,Fe)(OH)₂ with Mo at different sites. **a**, Mo atom replaces the surface Ni and exposes to the vacuum. **b**, Mo atom replaces surface Ni atom without the exposure to vacuum. **c**, Mo atom replaces the surface Fe atom. **d**, The comparison of the formation energy (ΔE_f) of configuration a-c.

The formation energy of oxygen vacancy (ΔG_{O_v}) was calculated with respect to the Gibbs free energy of O₂ at 298.15 K and 1.0 bar.

$$\Delta G_{O_v} = \frac{1}{2}G_{O_2} + G_{O_v} - G_{surface},$$

where G_{O_2} , G_{O_v} , and $G_{surface}$ are Gibbs free energies of O₂, surface with oxygen vacancy, and the clean surface, respectively. Since DFT calculations are inaccurate at describing the oxygen molecules, the Gibbs free energy of O₂ was calculated by:

$$G_{O_2} = 2G_{H_2O} - 2G_{H_2} + 4.92 \text{ eV},$$

where G_{H_2O} and G_{H_2} are Gibbs free energies of H₂O and H₂, respectively. The Gibbs free energy change of H₂O → H₂ + O₂ is 4.92 eV.

The **Figure A35** has been added in Supplementary Information as Supplementary Fig. S36.

The **Figure A36** has been added in Supplementary Information as **Supplementary Fig. S37**.

Page 33, line 14 in Supplementary Information, added:

“1) *Optimized DFT model of OER using for both AEM and LOM process:*

The slabs model of NiFe (oxy)hydroxide used in AEM and LOM pathway were terminated by the (001) surface (Supplementary Fig. S36). For the AEM pathway, the metal site should be exposed to the reactants. Thus, two vacuum spaces were inserted along (001) and (010). To eliminate the interaction between periodic slabs, the thickness of vacuum spaces in both models was more than 10 Å. In addition, part of hydrogen atoms was removed because of the oxidation atmosphere.

To find the stable configuration of Mo doping, we have built three slab models with different Mo sites (Supplementary Fig. S37). The relative stability of Mo replacement was determined by calculating the formation energy (ΔE_f), which was computed as:

$$\Delta E_f = E_{\text{slab}} - \sum N_i E_i,$$

where E_{slab} , E_i and N_i are total energies of the slab model, energy, and number of the i -th element, respectively. The calculated results show that the Mo atom favors replacing the Ni site and exposes to the vacuum as this model with the most negative ΔE_f (Supplementary Fig. S37d).”

Page 35, line 1 in Supplementary Information, added:

“3) *Oxygen vacancy formation energy calculation:*

The formation energy of oxygen vacancy (ΔG_{O_v}) was calculated with respect to the Gibbs free energy of O_2 at 298.15 K and 1.0 bar.

$$\Delta G_{O_v} = \frac{1}{2} G_{O_2} + G_{O_v} - G_{\text{surface}}, \quad (4)$$

where G_{O_2} , G_{O_v} , and G_{surface} are Gibbs free energies of O_2 , surface with oxygen vacancy, and the clean surface, respectively. Since DFT calculations are inaccurate at describing the oxygen molecules, the Gibbs free energy of O_2 was calculated by:

$$G_{O_2} = 2G_{H_2O} - 2G_{H_2} + 4.92 \text{ eV}, \quad (5)$$

where G_{H_2O} and G_{H_2} are Gibbs free energies of H_2O and H_2 , respectively. The Gibbs free energy change of $H_2O \rightarrow H_2 + O_2$ is 4.92 eV.”

4. Finally, the manuscript: “From 3D to 2D Co and Ni Oxyhydroxide Catalysts: Elucidation of the Active Site and Influence of Doping on the Oxygen Evolution Activity” (not cited either) by Vegge et al., ACS Catal. (2017), DOI: 10.1021/acscatal.7b02712. shows the variations in overpotentials resulting from the specific doping sites and whether the catalyst has bulk-like or nanosheets character. This information and discussion are also missing.

Response: We thank the reviewer for the insightful question.

First, we found that Mo doping strongly impacts the intrinsic OER activity of the NiFe (oxy)hydroxide. To rule out the impact of the morphology of catalysts on the OER activity, we assessed the intrinsic activity using mass activity, which was obtained by normalizing the CV curves by loading mass (**Figure A37**). All the Mo doping samples exhibited higher mass activity than the counterpart without Mo doping, suggesting that the Mo doping can effectively enhance the intrinsic OER activity of (oxy)hydroxide. Particularly, the MoNiFe (oxy)hydroxide exhibited the highest mass activity among all samples, with a current density of 1910 A/g at the overpotential of 300 mV. Such ultra-high mass activity of MoNiFe-27% (oxy)hydroxide is about 60 times higher than that of NiFe-27% (oxy)hydroxide (**Figure A38**).

Figure A37. a-c, Cyclic voltammetry polarization curves normalized by metal mass of catalysts of (a) Ni and MoNi, (b) Fe and MoFe, (c) NiFe-27% and MoNiFe-27% (oxy)hydroxide.

Figure A38. The comparison for the specific mass activity of Ni, MoNi, Fe, MoFe, NiFe and MoNiFe (oxy)hydroxide at the overpotential of 300 mV.

We agree with the reviewer the structure of the catalyst can critically impact its performance, and we believe the ultrathin nature of MoNiFe (oxy)hydroxide is one reason for its high stability. Chen et al. (*Advanced Materials*, 2019, 31(41): 1903909.) reported that the slow diffusion of proton acceptors within interlayer in NiFe hydroxide could lead to a local acidic environment, which results in a local etching process. The authors believed that such an etching process degrades the performance of multilayer NiFe hydroxide. In this work, the MoNiFe (oxy)hydroxide we obtained was ultrathin with an atomic thickness of 0.8 nm (mono-layer, denoted as 1L) or 1.5 nm (double-layer, denoted as 2L) as revealed by the atomic force microscopy (AFM) (**Figure A39**). Such ultra-thin nature of our MoNiFe (oxy)hydroxide can effectively prevent such local etching, and therefore is beneficial for the catalyst to remain stable during operation in alkaline solution.

Figure A39. Atomic force microscopy (AFM) image of the MoNiFe (oxy)hydroxide flakes. The inset figure is the corresponding line-trace height profile across a MoNiFe (oxy)hydroxide flake.

To identify the doping site of Mo in NiFe (oxy)hydroxide, we carried out extra aberration-corrected high-angle annular dark-field scanning transmission electron microscope (HAADF-STEM) measurement (**Figure A40**). The bright points in this Figure represent the Mo atom in MoNiFe (oxy)hydroxide due to the higher atomic mass of Mo than Ni and Fe atoms, which confirms the presence of Mo doping in MoNiFe (oxy)hydroxide. However, the specific doping site cannot be observed due to the low crystallinity of MoNiFe (oxy)hydroxide and its susceptibility to be damaged under electron beam irradiation.

Figure A40. The HAADF-STEM image of MoNiFe (oxy)hydroxide.

To reveal information about the Mo doping sites, we relied on DFT calculations to identify the possible site of Mo doping. We have built three slab models with different Mo sites (**Figure A41**). The relative stability of Mo replacement was determined by calculating the formation energy (ΔE_f), which was computed as:

$$\Delta E_f = E_{\text{slab}} - \sum N_i E_i,$$

Where E_{slab} , E_i and N_i are total energies of the slab model, energy, and number

of the i -th element, respectively. The calculated results show that the Mo atom favors replacing the Ni site and exposes to the vacuum as this model with the most negative ΔE_f (Figure A41d).

Figure A41. Doping (Ni,Fe)(OH)₂ with Mo at different sites. a, Mo atom replaces the surface Ni and exposes to the vacuum. **b**, Mo atom replaces surface Ni atom without the exposure to vacuum. **c**, Mo atom replaces the surface Fe atom. **d**, The comparison of the formation energy (ΔE_f) of configuration a-c.

The **Figure A40** has been added into Supplementary Information as *Supplementary Fig. S4*.

The **Figure A41** has been added in Supplementary Information as *Supplementary Fig. S37*.

Page 6, line 11 in manuscript, added: “*The presence of Mo dopant in NiFe (oxy)hydroxide was further confirmed by the aberration-corrected high-angle annular dark-field scanning transmission electron microscope (HAADF-STEM) (Supplementary Fig. S4).*”

Page 33, line 21 in Supplementary Information, added: “*To find the stable configuration of Mo doping, we have built three slab models with different Mo sites*

(*Supplementary Fig. S37*). The relative stability of Mo replacement was determined by calculating the formation energy (ΔE_f), which was computed as:

$$\Delta E_f = E_{slab} - \sum N_i E_i,$$

where E_{slab} , E_i and N_i are total energies of the slab model, energy, and number of the i -th element, respectively. The calculated results show that the Mo atom favors replacing the Ni site and exposes to the vacuum as this model with the most negative ΔE_f (*Supplementary Fig. S37d*).”

Page 9, line 16 in manuscript, added: “It is reported that the OER stability of (oxy)hydroxide is strongly dependent on its structural characteristics^{28, 29}.”

The reference recommended by the reviewer has been added to the revised manuscript:

Page 9, line 16 in manuscript, “[29] Tripkovic, V., Hansen, H. A., Vegge, T. From 3D to 2D Co and Ni oxyhydroxide catalysts: Elucidation of the active site and influence of doping on the oxygen evolution activity. *ACS Catal.* 7, 8558-8571 (2017).” was cited in the revised manuscript as reference [29] in “It is reported that the OER stability of (oxy)hydroxide is strongly dependent on its morphology structure^{28, 29}.”

5. In summary, albeit the system itself and the reported performance are of high potential interest to the field, the conclusions drawn are not sufficiently supported by the presented experimental and computational results to justify publication in Nature Communication. The authors are encouraged to complete the missing investigations and resubmit.

Response: We truly appreciate the reviewer for his/her valuable comments and suggestions, which enormously improved the quality and clarity of this manuscript. All of the comments and suggestions from the reviewer have been taken into account in the revised manuscript by adding additional experiments, including the ¹⁸O isotope-labeling experiments, in-situ differential electrochemical mass spectrometry (DEMS) measurements, HRTEM, and so on, as well as related discussions. We do hope the

reviewer find these changes satisfactory and are willing to further improve the manuscript if needed.

REVIEWER COMMENTS

Reviewer #1 (Remarks to the Author):

This manuscript has been revised by additional experiments, including isotope-labeling experiments and structural characterizations. Still, we have some serious concerns about this revised manuscript as follows:

(1) The electrolyte should be involved in the formation of NiOOH species when the catalysts were subjected to high potential (*J. Phys. Chem. C* 2015, 119, 7243–7254). Therefore, the Raman bands of ¹⁸O-labeled catalysts at 476 and 557 cm⁻¹ should be negatively shifted to lower wavenumbers regardless of the reaction mechanisms (*Angew. Chem. Int. Ed.* 2019, 58, 10295; *Angew. Chem. Int. Ed.* 2021, 60, 3095 – 3103). In view of this, it is unconvincing that there is no shift of Raman peaks for NiFe with ¹⁸O-labeling (Figure A3). Besides, it is difficult to determine whether there is a negative shift of Raman peaks for MoNiFe (oxy)hydroxide, due to the low signal-to-noise ratio and the evident fluctuation of baselines of Raman signals. Even if there is an evident shift of Raman bands of samples in electrolyte prepared by H₂¹⁸O, the presence of LOM still cannot be confirmed. I noticed that the above-mentioned references have been cited in this revised manuscript. However, it seems like that the authors may not fully understand the detailed procedure for determining the lattice oxygen involvement by Raman spectra. Therefore, the in-situ Raman experiments should be further performed and discussed.

(2) In this revised manuscript, the deuterium isotopic experiments and DFT calculations indicate that the potential determining step (PDS) of MoNiFe (oxy)hydroxide is the deprotonation of *OOH. Since the deprotonation of *OOH refers to a concerted proton-electron transfer (CPET) process in this manuscript, the PDS of MoNiFe (oxy)hydroxide includes the CPET. While, the previous reports indicate that the strong pH-dependence of OER activities for LOM-based catalysts arises from the potential-determining chemical deprotonation step (*Nat. Energy* 2019, 4, 329; *Nat. Commun.* 2021, 12, 3992). The authors should explain it.

(3) The DOS and COHP calculations prove that metal-oxygen bonds of MoNiFe (oxy)hydroxide are weaker than those of NiFe (oxy)hydroxide. However, the soft X-ray absorption spectroscopies, the XPS spectra and the in-situ Raman spectra (Figure 6) indicate the higher valence state of Ni and Fe in MoNiFe (oxy)hydroxide, which may lead to the improvement of covalency (*Energy Environ. Sci.* 2021, 14, 4647-4671). In other words, the theoretical calculations aren't consistent with the experimental analysis. In fact, most of previous works suggests the significance of high covalency for triggering lattice oxygen activation. The authors should explain such contradictive conclusions.

(4) The O 1s XPS spectra reveal the evident presence of oxygen defects (Figure A19). Since the oxygen defects play an important role on the reaction mechanisms, the oxygen defects should be considered when performing DFT calculations.

Reviewer #2 (Remarks to the Author):

I think the authors clearly answered the questions that I pointed out.

The authors show an inventive system and high catalytic performance in the present manuscript, and it will receive interest from the community of the field. The mechanism of the high catalytic performance was explained rationally, both from experimental analysis and theoretical calculations.

I think it is suitable to publish to Nature Communications as is.

Reviewer #3 (Remarks to the Author):

The authors have done an impressive job in addressing the comments and concerns outlined by the reviewers in the extensively revised manuscript. The manuscript has improved significantly and now warrants publication in Nature Communication. Before publication, the authors should consider the effect of correcting the known systematic errors in the use of the PBE exchange correlation functional for OER, as outlined in Christensen, et al. [10.1021/acs.jpcc.6b09141](https://doi.org/10.1021/acs.jpcc.6b09141) (Table A1, etc.).

Point-to-Point Responses to Reviewers' Comments and Suggestions

First of all, we truly appreciate the referees for their valuable comments and suggestions, which enormously improve the quality and clarity of this manuscript. All of the comments and suggestions have been taken into account in the revised manuscript as follows.

Reviewer #1 (Remarks to the Author):

COMMENTS TO AUTHOR:

This manuscript has been revised by additional experiments, including isotope-labeling experiments and structural characterizations. Still, we have some serious concerns about this revised manuscript as follows:

Response: We thank the reviewer for the valuable comments. All the concerns raised by the reviewer have been addressed in details as the following.

(1) The electrolyte should be involved in the formation of NiOOH species when the catalysts were subjected to high potential (J. Phys. Chem. C 2015, 119, 7243–7254). Therefore, the Raman bands of ^{18}O -labeled catalysts at 476 and 557 cm^{-1} should be negatively shifted to lower wavenumbers regardless of the reaction mechanisms (Angew. Chem. Int. Ed. 2019, 58, 10295; Angew. Chem. Int. Ed. 2021, 60, 3095-3103). In view of this, it is unconvincing that there is no shift of Raman peaks for NiFe with ^{18}O -labeling (Figure A3). Besides, it is difficult to determine whether there is a negative shift of Raman peaks for MoNiFe (oxy)hydroxide, due to the low signal-to-noise ratio and the evident fluctuation of baselines of Raman signals. Even if there is an evident shift of Raman bands of samples in electrolyte prepared by H_2^{18}O , the presence of LOM still cannot be confirmed. I noticed that the above-mentioned references have been cited in this revised manuscript. However, it seems like that the authors may not fully understand the detailed procedure for determining the lattice oxygen involvement by Raman spectra. Therefore, the in-situ Raman experiments should be further performed

and discussed.

Response: We thank the reviewer for the valuable comments and suggestions. We double check the description of the Raman results in previous manuscript, and find that the labels of samples in the figure maybe misleading because we skipped the detailed experimental condition in the main text. The Raman measurements in the previous manuscript were carried out in the following procedure: First, the NiFe and MoNiFe (oxy)hydroxide were completely activated in electrolyte with ^{16}O and the sample was named as NiFe or MoNiFe with ^{16}O . Then, the fully activated (oxy)hydroxide catalysts with ^{16}O were subjected to a constant potential of 1.65 V (vs. RHE) in electrolyte with ^{18}O and the obtained sample was named as NiFe or MoNiFe with ^{18}O in our previous manuscript. To be more specific, we changed the labels of the samples and the obtained Raman spectra are shown in **Figure A1**. Because the (oxy)hydroxides have been activated completely in electrolyte with ^{16}O to form ^{16}O -NiOOH species, the amount of ^{18}O in the sample after operating in electrolyte with ^{18}O only depends on the amount of oxygen in the lattice that participated in the OER reaction. The Raman peak of MoNiFe (oxy)hydroxide sample shifted to the lower wavelength suggesting a clear presence of ^{18}O in the lattice which is related to the oxygen exchange between ^{16}O in the lattice and ^{18}O in the electrolyte during OER. The reason why NiFe (oxy)hydroxide sample did not show noticeable changes is its slower oxygen exchange kinetics in comparison to MoNiFe (oxy)hydroxide. Consequently, after the same treatment, the amount of ^{18}O in the NiFe (oxy)hydroxide sample is much smaller.

Figure A1. a-b, Raman spectra of NiFe (a) and MoNiFe (b) (oxy)hydroxide after

being completely activated in electrolyte with ^{16}O (black) and then were subjected to a constant potential of 1.65 V (vs. RHE) in electrolyte with ^{18}O (red).

As suggested by the reviewer, we changed the experiment procedure to be similar as that in the references (*Angew. Chem. Int. Ed.* 2019, 58, 10295; *Angew. Chem. Int. Ed.* 2021, 60, 3095-3103). The samples were first activated in electrolyte with ^{18}O to form ^{18}O -NiOOH species, and then were subjected to a positive potential (1.65 V vs RHE) in electrolyte with ^{16}O . The Raman peaks of the samples activated in electrolyte with ^{18}O (named as ^{18}O -labelled sample) shifted to lower wavenumber compared to that of the samples activated in electrolyte with ^{16}O (named as ^{16}O -labelled sample), because of the impact of oxygen mass on the vibration mode (**Figure A2**). This result suggests that we successfully labelled both NiFe and MoNiFe samples with ^{18}O . Then, the ^{18}O -labelled (oxy)hydroxides were placed in electrolyte with ^{16}O and were treated by applying a positive potential of 1.65 V (vs. RHE) for different period of time (1 min to 20 min). The Raman spectra of obtained samples were shown in **Figure A2**. The Raman peak of ^{18}O -labelled MoNiFe (oxy)hydroxide shifted back to the position for ^{16}O -labelled MoNiFe (oxy)hydroxide within 1 min of treatment, which is much faster than that for the NiFe (oxy)hydroxide (20 min). This result suggests that, while both samples follow the LOM mechanism, the MoNiFe (oxy)hydroxide exhibited much higher rate of oxygen exchange between lattice oxygen and electrolyte. This result is in consistence with the DEMS results (Fig. 3 in the manuscript). It is needed to note the reason we carried quasi in-situ Raman spectra measurement instead of continuously measuring Raman spectra during applying potential is due to the influence of oxygen bubbles generated during OER on the Raman measurement.

Figure A2. a-b, Quasi in-situ Raman spectra of ^{18}O -labelled NiFe (a) and MoNiFe (b) (oxy)hydroxide measured at 1.65 V in 1.0 M KOH with H_2^{16}O . The Raman spectra of ^{16}O -labelled samples were shown in black dash lines for comparison.

In the revised manuscript we replace the previous results with the newly added quasi in-situ Raman measurement results.

Page 10, line 18 in manuscript, change from: “In addition to the DEMS measurement, the Raman spectra were also used to confirm the participation of lattice oxygen in OER. The Raman peaks of NiFe and MoNiFe (oxy)hydroxide with ^{18}O -labeling shifted towards lower wavenumber (Fig. 3e-f) because of the impact oxygen mass on the vibration mode^{2, 34}. The MoNiFe (oxy)hydroxide shows a more obvious shift than the NiFe (oxy)hydroxide, suggesting more lattice oxygen got exchanged with the electrolytes during OER.”

To “In addition to the DEMS measurement, the *quasi in-situ* Raman spectra were also used to confirm the participation of lattice oxygen in OER. The samples were first activated in electrolyte with ^{18}O to form ^{18}O -NiOOH species, and then were subjected to a positive potential (1.65 V vs RHE) in electrolyte with H_2^{16}O . The Raman peaks of the samples activated in electrolyte with ^{18}O (named as ^{18}O -labelled sample) shifted to lower wavenumber comparing to that of the samples activated in electrolyte with ^{16}O (named as ^{16}O -labelled sample), because of the impact of oxygen mass on the vibration mode^{2, 34}(Fig. 3e,f). This result suggests that we successfully labelled both NiFe and MoNiFe samples with ^{18}O . Then, the ^{18}O -labelled (oxy)hydroxides were placed in

electrolyte with ^{16}O and were treated by applying a positive potential of 1.65 V (vs. RHE) for different period of time (1 min to 20 min). The Raman spectra of the obtained samples were shown in Fig. 3e,f. The Raman peak of ^{18}O -labelled MoNiFe (oxy)hydroxide shifted back to the position for ^{16}O -labelled MoNiFe (oxy)hydroxide within 1 min of treatment, which is much faster than that for the NiFe (oxy)hydroxide (20 min). This result suggests that while both samples follow the LOM mechanism, the MoNiFe (oxy)hydroxide exhibited much higher rate of oxygen exchange between lattice oxygen and electrolyte.”

Correspondingly, the **Fig. 3** in manuscript was changed to:

Fig 3. Evidence of lattice oxygen participating in OER provided by ^{18}O isotope-labeling experiments. a-d, Mass spectrometric cyclic voltammograms results showing different gaseous products content of OER reaction as a function of applied potential for the ^{18}O -labeled samples: $^{16}\text{O}^{18}\text{O}$ for NiFe (oxy)hydroxide (a) and MoNiFe (oxy)hydroxide (b), and $^{18}\text{O}_2$ for NiFe (oxy)hydroxide (c) and MoNiFe (oxy)hydroxide (d). The contents of all the species were normalized by the amount of $^{16}\text{O}_2$ in the reaction products; e-f, Quasi in-situ Raman spectra of ^{18}O -labelled NiFe (e) and ^{18}O -labelled NiFe MoNiFe (f) (oxy)hydroxides after being applied a positive potential of 1.65 V (vs. RHE) in 1.0 M KOH with H_2^{16}O for different time (1 min to 20 min). The Raman spectra of ^{16}O -labelled samples were shown in black dash lines for comparison.

The reference recommended by the reviewer and the related reference have been cited in the revised manuscript:

Page 10, line 23 in manuscript, “[2] Lee, S., Bai, L., Hu, X. *Deciphering iron-dependent activity in oxygen evolution catalyzed by nickel-iron layered double hydroxide. Angew. Chem. Int. Ed. Engl.* 59, 8072-8077 (2020). [34] Lee, S., Banjac, K., Lingenfelder, M., Hu, X. *Oxygen isotope labeling experiments reveal different reaction sites for the oxygen evolution reaction on nickel and nickel iron oxides. Angew. Chem. Int. Ed. Engl.* 58, 10295-10299 (2019).” were cited in the revised manuscript as reference [2] and [34] in “*The Raman peaks of the samples activated in electrolyte with ^{18}O (named as ^{18}O -labelled sample) shifted to lower wavenumber comparing to that of the samples activated in electrolyte with ^{16}O (named as ^{16}O -labelled sample), because of the impact of oxygen mass on the vibration mode^{2, 34} (Fig. 3e,f).*”

(2) In this revised manuscript, the deuterium isotopic experiments and DFT calculations indicate that the potential determining step (PDS) of MoNiFe (oxy)hydroxide is the deprotonation of *OOH. Since the deprotonation of *OOH refers to a concerted proton-electron transfer (CPET) process in this manuscript, the PDS of MoNiFe (oxy)hydroxide includes the CPET. While, the previous reports indicate that the strong pH-dependence of OER activities for LOM-based catalysts arises from the potential-determining chemical deprotonation step (Nat. Energy 2019, 4, 329; Nat. Commun. 2021, 12, 3992). The authors should explain it.

Response: We thank the reviewer for the valuable comments. We totally agree with the reviewer that the pH-dependent activity on the RHE scale means that OER includes a non-concerted proton-electron transfer (nCPET) process. For the catalysts with the LOM mechanism, the OER reaction normally involves the nCPET process. In some literature (*Nature Communications*, 2020, 11(1): 2002.; *Nature Communications* 2021, 12, 3992), such as the ones pointed out by the reviewer, the proton and electron transfer process occurred in two separate reaction steps, in which the PDS is the step that involved proton transfer. For instance, Huang et al. reported that the PDS of OER on

$\text{Na}_x\text{Mn}_3\text{O}_7$ is the chemical deprotonation step, while the electron transfer was accompanied by the O_2 desorption step (*Nature Communications* 2021, 12, 3992). Nevertheless, there are also many previous literature which reported that although the electron and proton transfer both occurred in the same step, these two processes proceed sequentially instead of simultaneously (*Catalysis Today*, 2016, 262: 2-10.; *Advanced Materials*, 2018, 30(32): 1802912.; *EcoMat*, 2020, 2(2): e12021.). Because it is difficult to identify whether proton transfer or electron transfer occurs first, the proton transfer step and the electron transfer step generally are shown together in the schematic illustration of pathways and energy diagrams. For example, Zhou et al. (*Advanced Materials*, 2018, 30(32): 1802912.) reported a spinel oxide catalyst of $\text{ZnFe}_{0.4}\text{Co}_{1.6}\text{O}_4$ with pH-dependent OER activity, whose PDS (the formation of $^*\text{OOH}$) includes decoupled proton-electron transfer pathways (**Figure A3**). Similarly, Zhu et al. (*EcoMat*, 2020, 2(2): e12021.) observed a pH-dependent OER activity of $\text{Sr}_3(\text{Co}_{0.8}\text{Fe}_{0.1}\text{Nb}_{0.1})_2\text{O}_{7-\delta}$ catalyst and proposed a lattice oxygen mechanism with the PDS that includes both proton transfer and electron transfer (**Figure A4**).

Figure A3. a-c, The pH dependence (a), the schematic illustration of the proposed OER pathway in AEM mechanism (b), and the schematic illustration of decoupled proton-electron transfer of the potential determining step (c) for $\text{ZnFe}_{0.4}\text{Co}_{1.6}\text{O}_4$ catalyst (*Advanced Materials*, 2018, 30(32): 1802912.).

Figure A4. a-c, The pH dependence (a), the schematic illustration of the proposed OER pathway in LOM mechanism (b), and Gibbs free energy diagrams in LOM for $\text{Sr}_3(\text{Co}_{0.8}\text{Fe}_{0.1}\text{Nb}_{0.1})_2\text{O}_{7-\delta}$ catalyst. (EcoMat, 2020, 2(2): e12021.)

In our work, we indeed considered two possible pathways (LOM-1 and LOM-2) in our DFT calculations. While both proton transfer and electron transfer occur on the deprotonation of $^*\text{OOH}$ step in the LOM-1 pathway (**Figure A5a**), the proton transfer occurs on the deprotonation of $^*\text{OOH}$ step and the electron transfer occurs on the O_2 desorption step in the LOM-2 pathway (**Figure A5b**). We found that the reaction barrier for the LOM-1 pathway is significantly lower than that for the LOM-2 pathway, as shown in **Figure A5c**. This result suggests that the LOM-1 pathway is more favorable for our cases.

Figure A5. a-b, Schematic illustration of two possible LOM pathways on MoNiFe (oxy)hydroxide: LOM-1 (a) and LOM-2 (b). c, The Gibbs free energy diagrams of OER in LOM-1 and LOM-2 pathways on MoNiFe (oxy)hydroxide.

Since we observed a strong pH dependence for the MoNiFe (oxy)hydroxide, we believe that although the proton and electron transfer both occurred in the PDS, they

are actually transferred sequentially, i.e., the electron and proton transfer processes are decoupled as shown in **Figure A6**. DFT method is known to be problematic dealing with charged systems, and it is challenging to assign charge to an atom during calculations. Therefore, it is difficult to verify the sequence of proton transfer and electron transfer in our PDS step.

Figure A6. Illustration of decoupled proton-electron transfer of the potential determining step in MoNiFe (oxy)hydroxide.

In the revised manuscript, we emphasised that although the proton and electron transfer both occurred during the PDS step, i.e., the deprotonation of *OOH, the pH dependence results suggest that they actually occurred sequentially, i.e. the electron and proton transfer are decoupled.

Page 14, line 10 in manuscript, changed from: “The higher ρ^{RHE} for MoNiFe (oxy)hydroxide implied a stronger pH-dependent OER activity, which may be due to the higher degree of decoupled proton-electron transfer during the PDS step, i.e., the deprotonation of *OOH. (Supplementary Fig. S22)^{8, 9, 38}.”

To “The higher ρ^{RHE} for MoNiFe (oxy)hydroxide implied a stronger pH-dependent OER activity, which may be due to the higher degree of decoupled proton-electron transfer during the PDS step, i.e., the deprotonation of *OOH. (Supplementary note 3, Supplementary Fig. S22-23)^{8, 9, 38}.”

Page 30, line 5 in Supplementary Information, added:

“*Supplementary note 3*

The pH-dependent activity on the RHE scale means that OER includes a non-concerted proton-electron transfer (nCPET) process. For the catalysts with the LOM mechanism, the OER reaction normally involves the nCPET process. In some literature, the proton and electron transfer process occurred in two separate reaction steps, in which the PDS is the step that involved proton transfer^{29,30}. For instance, Huang et al.²⁹ reported that the PDS of OER on $\text{Na}_x\text{Mn}_3\text{O}_7$ is the chemical deprotonation step, while the electron transfer was accompanied by the O_2 desorption step. Nevertheless, there are also many previous literature which reported that although the electron and proton transfer both occurred in the same step, these two processes proceed sequentially instead of simultaneously³¹⁻³³. Because it is difficult to identify whether proton transfer or electron transfer occurs first, the proton transfer step and the electron transfer step generally are shown together in the schematic illustration of pathways and energy diagrams. For example, Zhou et al.³² reported a spinel oxide catalyst of $\text{ZnFe}_{0.4}\text{Co}_{1.6}\text{O}_4$ with pH-dependent OER activity, whose PDS (the formation of *OOH) includes decoupled proton-electron transfer pathways. Similarly, Zhu et al.³³ observed a pH-dependent OER activity of $\text{Sr}_3(\text{Co}_{0.8}\text{Fe}_{0.1}\text{Nb}_{0.1})_2\text{O}_{7-8}$ catalyst and proposed a lattice oxygen mechanism with the PDS that includes both proton transfer and electron transfer.

In our work, we indeed considered two possible pathways (LOM-1 and LOM-2) in our DFT calculations. While both proton transfer and electron transfer occur on the deprotonation of *OOH step in the LOM-1 pathway (**Supplementary Fig. S22a**), the proton transfer occurs on the deprotonation of *OOH step and the electron transfer occurs on the O_2 desorption step in the LOM-2 pathway (**Supplementary Fig. S22b**). We found that the reaction barrier for the LOM-1 pathway is significantly lower than that for the LOM-2 pathway, as shown in **Supplementary Fig. S22c**. This result suggests that the LOM-1 pathway is more favorable for our cases.

Since we observed a strong PH dependence for the MoNiFe (oxy)hydroxide, we believe that although the proton and electron transfer both occurred in the PDS, they

are actually transferred sequentially, i.e., the electron and proton transfer process are decoupled as shown in **Supplementary Fig. S23**. DFT method is known to be problematic dealing with charged systems, and it is challenging to assign charge to an atom during calculations. Therefore, it is difficult to verify the sequence of proton transfer and electron transfer in our PDS step.”

The **Figure A5** has been added in Supplementary Information as **Supplementary Fig. S22**.

The reference recommended by the reviewer and the related reference have been cited in the revised Supplementary Information:

Page 30, line 8 in Supplementary Information, “[29] Huang, Z.-F., et al. *Tuning of lattice oxygen reactivity and scaling relation to construct better oxygen evolution electrocatalyst*. *Nat. Commun.* 12, 3992 (2021). [30] Pan, Y., et al. *Direct evidence of boosted oxygen evolution over perovskite by enhanced lattice oxygen participation*. *Nat. Commun.* 11, 2002 (2020).” were cited in the revised Supplementary Information as reference [29] and [30] in “*In some literatures, the proton and electron transfer process clearly occurred in two separate reactions step, in which the PDS is the step that evolved proton transfer*^{29 30}.”

Page 30, line 12 in Supplementary Information, “[31] Giordano, L., et al. *pH dependence of OER activity of oxides: Current and future perspectives*. *Catal. Today* 262, 2-10 (2016). [32] Zhou, Y., et al. *Enlarged Co-O covalency in octahedral sites leading to highly efficient spinel oxides for oxygen evolution reaction*. *Adv. Mater.* 30, 1802912 (2018). [33] Zhu, Y., et al. *Boosting oxygen evolution reaction by activation of lattice-oxygen sites in layered Ruddlesden-Popper oxide*. *EcoMat* 2, 12021 (2020).” were cited in the revised Supplementary Information as reference [31], [32] and [33] in “*Nevertheless, there are also many previous literatures which reported that although the electron and proton transfer occurred in the same step, but they transfer sequentially instead of transfer simultaneously*³¹⁻³³.”

(3) The DOS and COHP calculations prove that metal-oxygen bonds of MoNiFe

(oxy)hydroxide are weaker than those of NiFe (oxy)hydroxide. However, the soft X-ray absorption spectroscopies, the XPS spectra and the in-situ Raman spectra (Figure 6) indicate the higher valence state of Ni and Fe in MoNiFe (oxy)hydroxide, which may lead to the improvement of covalency (Energy Environ. Sci. 2021, 14, 4647-4671). In other words, the theoretical calculations aren't consistent with the experimental analysis. In fact, most of previous works suggests the significance of high covalency for triggering lattice oxygen activation. The authors should explain such contradictive conclusions.

Response: We thank the reviewer for the valuable comments. The covalency of metal-oxygen bond was reported to be determined by the overlap of oxygen $2p$ orbitals and metal $3d$ orbitals in DOS, which can be quantified by the distance between the centers of the metal d -band and oxygen p -band ($\epsilon_{M\ 3d} - \epsilon_{O\ 2p}$) (*Nature Catalysis*, 2020, 3(7): 554-563.). In our work, we found that the MoNiFe (oxy)hydroxide showed a higher overlap between Ni $3d$ -orbital and O $2p$ -orbital. As shown in **Figure A7**, the specific positions of O $2p$ (Ni $3d$)-band center were calculated to be -1.58 eV (-2.72 eV) and -1.40 eV (-2.08 eV) for NiFe (oxy)hydroxide and MoNiFe (oxy)hydroxide, respectively. As a result, the $\epsilon_{M\ 3d} - \epsilon_{O\ 2p}$ values of NiFe and MoNiFe (oxy)hydroxide were determined to be -1.14 eV and -0.68 eV, respectively. The smaller band distance of Ni $3d$ - O $2p$ band center of MoNiFe (oxy)hydroxide suggested a higher covalency of Ni-O bond in MoNiFe (oxy)hydroxide, which is consistent with the higher valence state of Ni in MoNiFe (oxy)hydroxide. It is noted that, although the overlap between Ni $3d$ - O $2p$ band is enhanced after Mo doping, such overlap occurred more on the anti-bonding states below Fermi level as highlighted in dash circles in **Figure A7**. As a consequence, a weaker Ni-O bond was observed in the MoNiFe (oxy)hydroxide, which can facilitate oxygen vacancy formation. Consistently, we observed a lower oxygen vacancy formation energy in the MoNiFe (oxy)hydroxide than that in NiFe (oxy)hydroxide (**Figure A8**). We believe that the lower oxygen vacancy formation energy, which is related to a higher oxygen lattice activity is the reason for the low reaction barrier for the LOM pathway for MoNiFe (oxy)hydroxide.

Figure A7. Projected density of states of NiFe and MoNiFe (oxy)hydroxide. The anti-bonding states below the Fermi level were highlighted by dash circles. The dash lines indicate the position of band centers.

Figure A8. The oxygen vacancy formation energy ($E_{f,vac}$) of NiFe and MoNiFe (oxy)hydroxide.

We strongly agree with the reviewer that higher covalency of metal-oxygen bond is favorable to trigger lattice oxygen activation as reported in many previous literatures. In fact, in addition to the covalency of metal-oxygen bond (represented by the relative

energy alignment between metal 3d- and oxygen 2p-bands, $\epsilon_{M\ 3d} - \epsilon_{O\ 2p}$), oxygen activity (represented by the absolute energy level of the O 2p-band, $\epsilon_{O\ 2p}$) also exhibited a profound influence on the OER mechanism. As shown in **Figure A9**, Zhang et al. summarized the OER reaction mechanism (LOM or AEM), $\epsilon_{M\ 3d} - \epsilon_{O\ 2p}$ and $\epsilon_{O\ 2p}$ value for different catalyst (*Energy & Environmental Science*, 2021, 14: 4647-4671). For catalysts with similar $\epsilon_{M\ 3d} - \epsilon_{O\ 2p}$ value, i.e., similar metal-oxygen covalence, the ones with higher O 2p band center are more likely to follow the LOM mechanism.

Figure A9. a-b, The relationship between the OER mechanism (AEM or LOM) and electronic descriptors for perovskite oxides (**a**) and spinel oxides (**b**) electrocatalysts.

(*Energy & Environmental Science*, 2021, 14: 4647-4671)

Consistently, Sun et al. (*Nature Catalysis*, 2020, 3(7): 554-563.) found a similar phenomenon (**Figure A10**) and demonstrated that oxygen 2p-band center of spinel oxides is required to be high enough to guarantee the lattice oxygen to escape from the lattice. The reason why catalyst with high oxygen 2p band position is likely to follow the LOM mechanism (**Figure A9 and A10**) is due to the close correlation between oxygen vacancy formation energy, i.e., the stability of lattice oxygen, and O 2p band position. Catalysts with high O 2p position are generally reported to exhibit lower oxygen vacancy formation energy (*Energy & Environmental Science*, 2011, 4(10): 3966-3970.). A facilitated oxygen vacancy formation process can promote the LOM mechanism, as reported by Mefford et al. for perovskite cobaltites (*Nature*

Communications, 2016, 7(1): 11053.) In our work, we observed similar phenomena that MoNiFe (oxy)hydroxide showed higher O 2p-band center relative to Fermi level (**Figure A7**) and lower oxygen vacancy formation energy (**Figure A8**). The higher lattice oxygen activity of MoNiFe (oxy)hydroxide leads to a promoted LOM pathway.

Figure A10. The relationship between the OER mechanism (AEM or LOM) and electronic descriptors for spinel oxides. (*Nature Catalysis*, 2020, 3(7): 554-563.)

Page 16, line 2 in manuscript, change from: “*The upshift of the O 2p band resulted in deeper penetration of Fermi level into the O 2p band, which further facilitated the electron flow away from oxygen sites when an anodic potential is applied, making the lattice oxygen release from the lattice more easily^{3, 12, 43}.*”

To “*It is reported that O 2p-band center is required to be high enough to guarantee the lattice oxygen to escape from the lattice⁴³. The upshift of the O 2p band resulted in deeper penetration of Fermi level into the O 2p band, which further facilitated the electron flow away from oxygen sites when an anodic potential is applied, making the lattice oxygen release from the lattice more easily^{3, 12, 43}. As a consequence, oxygen with high O 2p band position exhibited facilitated oxygen vacancy formation process and thus promoted the LOM mechanism⁴².*”

Page 16, line 22 in manuscript, changed from: “*In addition, the density of states of metal 3d-orbital, especially for Ni 3d-orbital, upshift close to Fermi level, leading to*

an increase in the antibonding states below the Fermi level (Fig. 5c). Such an effect weakens the metal-oxygen bonds, which is consistent with the COHP calculations (Fig. 5b)."

To *"In addition, the density of states of metal 3d-orbital, especially for Ni 3d-orbital, upshift close to Fermi level. Although such upshift leads to an increased overlap between Ni 3d-orbital and O 2p-orbital show higher overlap in DOS diagrams, the overlap of O 2p - Ni 3d orbital occurs on the anti-bonding states below Fermi level as highlighted in dash circles in Fig. 5c and resulted in a weaker Ni-O bond, which is consistent with the COHP calculations (Fig. 5b)."*

The reference recommended by the reviewer and the related reference have been cited in the revised manuscript:

Page 16, line 3 in manuscript, *"[43] Sun, Y., et al. Covalency competition dominates the water oxidation structure-activity relationship on spinel oxides. Nat. Catal. 3, 554-563 (2020)."* was cited in the revised manuscript as reference [43] in *"It is reported that O 2p-band center is required to be high enough to guarantee the lattice oxygen to escape from the lattice⁴³."*

Page 16, line 6 in manuscript, *"[3] Zhang, N., Chai, Y. Lattice oxygen redox chemistry in solid-state electrocatalysts for water oxidation. Energy Environ. Sci., 4647-4671 (2021)."* was cited in the revised manuscript as reference [3] in *"The upshift of the O 2p band resulted in deeper penetration of Fermi level into the O 2p band, which further facilitated the electron flow away from oxygen sites when an anodic potential is applied, making the lattice oxygen release from the lattice more easily^{3, 12, 43}."*

Page 16, line 6 in manuscript, *"[42] Lee, Y.-L., Kleis, J., Rossmeisl, J., Shao-Horn, Y., Morgan, D. Prediction of solid oxide fuel cell cathode activity with first-principles descriptors. Energy Environ. Sci. 4, 3966-3970 (2011)."* was cited in the revised manuscript as reference [42] in *"As a consequence, oxygen with high O 2p band position exhibited facilitated oxygen vacancy formation process and thus promoted the LOM mechanism⁴²."*

(4) The O 1s XPS spectra reveal the evident presence of oxygen defects (Figure A19). Since the oxygen defects play an important role on the reaction mechanisms, the oxygen defects should be considered when performing DFT calculations.

Response: We thank the reviewer for the valuable suggestion. As suggested by the reviewer, we performed additional DFT calculations for OER reaction on NiFe and MoNiFe (oxy)hydroxide with oxygen defects.

The Gibbs free energy diagrams of OER in the AEM pathway on NiFe and MoNiFe (oxy)hydroxide with oxygen vacancy are shown in **Figure A11**. The corresponding configurations of reaction intermediate are shown in **Figure A12**. We found that the Fe sites serve as active sites in the presence of oxygen vacancy for both NiFe and MoNiFe (oxy)hydroxide, which is similar to the case without oxygen vacancy. As shown in **Figure A11c**, the deprotonation of *OH in the AEM pathway serves as PDS for both NiFe and MoNiFe (oxy)hydroxide, with a barrier of 0.87 eV and 0.90 eV, respectively. The Gibbs free energy diagrams of OER in the LOM pathway on NiFe and MoNiFe with oxygen vacancy are shown in **Figure A13**. The corresponding configurations of reaction intermediate are shown in **Figure A14**. In the LOM pathway, the formation of gaseous O₂ and the deprotonation of *OOH act as PDSs for NiFe and MoNiFe (oxy)hydroxide, with a barrier of 0.75 eV and 0.42 eV, respectively (**Figure A13b**), which is the same as the case without oxygen vacancy. These DFT results show that, after introducing oxygen vacancy on the surface, the reaction barrier of the LOM pathway is still lower than that in the AEM pathway. Therefore, the LOM pathway is still dominant for both NiFe and MoNiFe (oxy)hydroxide when surface defects were considered, and the MoNiFe (oxy)hydroxide exhibited a lower reaction barrier.

Figure A11. a, Schematic illustration of the AEM pathway. b-c, The Gibbs free

energy diagrams of OER in the AEM pathway on Ni site (b) and Fe site (c) in NiFe and MoNiFe (oxy)hydroxide with oxygen vacancy.

Figure A12. The adsorption configurations of reaction intermediates on NiFe and MoNiFe (oxy)hydroxides with oxygen vacancy in the AEM mechanism.

Figure A13. a, Schematic illustration of the LOM pathway. **b,** The Gibbs free energy diagrams of OER in the LOM pathway on NiFe and MoNiFe (oxy)hydroxides with oxygen vacancy.

Figure A14. The adsorption configurations of reaction intermediate on NiFe and MoNiFe (oxy)hydroxides with oxygen vacancy in the LOM mechanism.

Page 17, line 11 in manuscript, add: “*Our DFT calculation further shows that the LOM pathway is still dominant for both NiFe and MoNiFe (oxy)hydroxide when there is oxygen vacancy presence on the surface (Supplementary Fig. S27-S30, note 5).*”

Page 32, line 10 in Supplementary Information, added:

“**Supplementary note 5**

*The Gibbs free energy diagrams of OER in the AEM pathway on NiFe and MoNiFe with oxygen vacancy are shown in Supplementary Fig. S27. The corresponding configurations of reaction intermediate are shown in Supplementary Fig. S28. We found that the Fe sites serve as active sites in the presence of oxygen vacancy for both NiFe and MoNiFe (oxy)hydroxide, which is similar to the case without oxygen vacancy. As shown in Supplementary Fig. S27c, the deprotonation of $*OH$ in the AEM pathway serves as PDS for both NiFe and MoNiFe (oxy)hydroxide, with a barrier of 0.87 eV and 0.90 eV, respectively. The Gibbs free energy diagrams of OER in the LOM pathway on NiFe and MoNiFe with oxygen vacancy are shown in Supplementary Fig. S29. The corresponding configurations of reaction intermediate are shown in Supplementary Fig. S30. In the LOM pathway, the formation of gaseous O_2 and the deprotonation of $*OOH$ act as PDSs for NiFe and MoNiFe (oxy)hydroxide, with a barrier of 0.75 eV and 0.42 eV, respectively (Supplementary Fig. S29b), which is the same as the case without oxygen vacancy. These DFT results show that, after introducing oxygen*

vacancy on the surface, the reaction barrier in the LOM pathway is still lower than that in the AEM pathway. Therefore, the LOM pathway is still dominant for both NiFe and MoNiFe (oxy)hydroxide when surface defects were considered, and the MoNiFe (oxy)hydroxide exhibited a lower reaction barrier.”

The **Figure A11** has been added in Supplementary Information as *Supplementary Fig. S27*.

The **Figure A12** has been added in Supplementary Information as *Supplementary Fig. S28*.

The **Figure A13** has been added in Supplementary Information as *Supplementary Fig. S29*.

The **Figure A14** has been added in Supplementary Information as *Supplementary Fig. S30*.

Reviewer #2 (Remarks to the Author):

COMMENTS TO AUTHOR:

I think the authors clearly answered the questions that I pointed out.

The authors show an inventive system and high catalytic performance in the present manuscript, and it will receive interest from the community of the field. The mechanism of the high catalytic performance was explained rationally, both from experimental analysis and theoretical calculations.

I think it is suitable to publish to Nature Communications as is.

Response: We thank the reviewer for acknowledging the acceptance of our work.

Reviewer #3 (Remarks to the Author):

COMMENTS TO AUTHOR:

The authors have done an impressive job in addressing the comments and concerns outlined by the reviewers in the extensively revised manuscript. The manuscript has improved significantly and now warrants publication in Nature Communication.

Response: We thank the reviewer for acknowledging the acceptance of our work.

Before publication, the authors should consider the effect of correcting the known systematic errors in the use of the PBE exchange correlation functional for OER, as outlined in Christensen, et al. 10.1021/acs.jpcc.6b09141 (Table A1, etc.).

Response: We thank the reviewer for the valuable suggestion. We agree with the reviewer that systematic errors exist in the PBE functional. As mentioned in the reference (*The Journal of Physical Chemistry C*, 2016, 120(43): 24910-24916.), these systematic errors arise from the difficulty in describing the triplet ground state of the O-O bond using DFT. Compared with the experimental results, the PBE functional used in this work may overestimate the total energy of O-O bond by 0.20 eV with a standard deviation of 0.03 eV (*The Journal of Physical Chemistry C*, 2016, 120(43): 24910-24916.). Therefore, we corrected the systematic error in both LOM and AEM mechanisms considered in this work.

As shown in **Table A1**, the systematic errors caused by PBE functional will not affect our conclusions in this work. Specifically, the correction of PBE functional will lower the total energy of OO* and OOH* by 0.2 eV. Therefore, for those sites where the peroxide species are not involved in the PDS steps, the overpotential value remains unchanged. On the other hand, the general trends still hold if peroxide species are involved in the PDS. For example, the overpotential of OER on the surface of NiFe (oxy)hydroxide increases by 0.2 eV after the correction. In contrast, the overpotential on MoNiFe surfaces remains the same because the energies of OO* and OOH* are shifted with the same magnitude.

Table A1. Correction of errors introduced by PBE functional. All units are given in eV.

	ΔG	without correction		with correction		Correction
		NiFe	MoNiFe	NiFe	MoNiFe	
LOM	ΔG_1	0.33	-0.34	0.33	-0.34	0.00
	ΔG_2	0.45	0.12	0.25	-0.08	-0.20
	ΔG_3	0.27	0.42^a	0.27	0.42^a	0.00
	ΔG_4	0.75^a	-0.02	0.95^a	0.18	0.20
	ΔG_5	-1.81	-0.19	-1.81	-0.19	0.00
AEM on Ni site	ΔG_1	-0.01	0.27	-0.01	0.27	0.00
	ΔG_2	1.08^a	1.12^a	1.08^a	1.12^a	0.00
	ΔG_3	-0.25	-0.42	-0.45	-0.62	-0.20
	ΔG_4	-0.82	-0.97	-0.62	-0.77	0.20
AEM on Fe site	ΔG_1	-0.24	-0.26	-0.24	-0.26	0.00
	ΔG_2	1.05^a	0.76^a	1.05^a	0.76^a	0.00
	ΔG_3	-0.05	0.21	-0.25	0.01	-0.20
	ΔG_4	-0.76	-0.70	-0.56	-0.50	0.20
AEM on Ni site (with oxygen vacancy)	ΔG_1	0.30	0.18	0.30	0.18	0.00
	ΔG_2	1.14^a	0.98^a	1.14^a	0.98^a	0.00
	ΔG_3	-0.64	-0.15	-0.84	-0.35	-0.20
	ΔG_4	-0.80	-1.00	-0.60	-0.80	0.20
AEM on Fe site (with oxygen vacancy)	ΔG_1	-0.29	-0.32	-0.29	-0.32	0.00
	ΔG_2	0.87^a	0.90^a	0.87^a	0.90^a	0.00
	ΔG_3	0.02	0.03	-0.18	-0.17	-0.20
	ΔG_4	-0.60	-0.61	-0.40	-0.41	0.20

a: The potential determining step (PDS)

Page 42, line 8 in Supplementary Information, added: “*The systematic errors of the PBE functional were considered, which arise from the difficulty in describing the triplet ground state of the O-O bond using DFT⁴⁸. Compared with the experimental results, the PBE functional used in this work may overestimate the total energy of O-O bond by 0.20 eV with a standard deviation of 0.03 eV⁴⁸. As shown in Supplementary Table S3, the systematic errors caused by PBE functional will not affect our conclusions in this work. Specifically, the correction of PBE functional will lower the total energy of OO* and OOH* by 0.2 eV. Therefore, for those sites where the peroxide species are not involved in the PDS steps, the overpotential value remains unchanged. On the other hand, the general trends still hold if peroxide species are involved in the PDS. For*

example, the overpotential of OER on the surface of NiFe (oxy)hydroxide increases by 0.2 eV after the correction. In contrast, the overpotential on MoNiFe surfaces remains the same because the energies of OO and OOH* are shifted with the same magnitude.”*

Table A1 have been added in Supplementary Information as **Supplementary Table S3**.

The reference pointed by the reviewer has been added to the revised Supplementary Information:

Page 42, line 8 in Supplementary Information, “[48] *Christensen, R., Hansen, H. A., Dickens, C. F., Nørskov, J. K., Vegge, T. Functional Independent Scaling Relation for ORR/OER Catalysts. The Journal of Physical Chemistry C 120, 24910-24916 (2016).*”

was cited in the revised manuscript as reference [48] in “*The systematic errors of the PBE functional were considered, which arise from the difficulty in describing the triplet ground state of the O-O bond using DFT⁴⁸.*”

REVIEWERS' COMMENTS

Reviewer #1 (Remarks to the Author):

I thank the authors for their careful consideration of the revision requests. They have added new data as well as clarifying explanations that address the questions that were raised on previous versions of the manuscript. I now recommend this manuscript for publication.

Point-to-Point Responses to Reviewers' Comments and Suggestions

Reviewer #1 (Remarks to the Author):

COMMENTS TO AUTHOR:

I thank the authors for their careful consideration of the revision requests. They have added new data as well as clarifying explanations that address the questions that were raised on previous versions of the manuscript. I now recommend this manuscript for publication.

Response: We thank the reviewer for acknowledging the acceptance of our work. We truly appreciate the reviewer for his/her valuable comments and suggestions, which enormously improve the quality and clarity of this manuscript.